# Remasking Discrete Diffusion Models with Inference-Time Scaling

**Guanghan Wang**[*][†]   **Yair Schiff**[†]   **Subham Sekhar Sahoo**   **Volodymyr Kuleshov**

Cornell Tech, Cornell University

{gw354,yzs2,sss284,vk379}@cornell.edu

[*] denotes corresponding author   [†] denotes equal contribution

## Abstract

Part of the success of diffusion models stems from their ability to perform iterative refinement, i.e., repeatedly correcting outputs during generation. However, modern masked discrete diffusion lacks this capability: when a token is generated, it cannot be updated again, even when it introduces an error. Here, we address this limitation by introducing the remasking diffusion model (ReMDM) sampler, a method that can be applied to pretrained masked diffusion models in a principled way and that is derived from a discrete diffusion model with a custom remasking backward process. Most interestingly, ReMDM endows discrete diffusion with a form of inference-time compute scaling. By increasing the number of sampling steps, ReMDM generates natural language outputs that approach the quality of autoregressive models, whereas when the computation budget is limited, ReMDM better maintains quality. ReMDM also improves sample quality of masked diffusion models for discretized images, and in scientific domains such as molecule design, ReMDM facilitates diffusion guidance and pushes the Pareto frontier of controllability relative to classical masking and uniform noise diffusion. We provide the code along with a blog post on the project page: https://remdm.github.io

## 1   Introduction

Diffusion models have gained significant traction as algorithms for generating high-quality images and videos [50, 52, 18, 42]. Part of the success of diffusion stems from its ability to perform iterative refinement—repeatedly modifying outputs and fixing errors over multiple steps of generation—which makes diffusion models especially effective at guided generation [9, 17] and fast sampling [51, 44, 54] and supports inference-time scaling to improve sample quality [30, 49, 66].

Discrete counterparts of diffusion models [2] have also been steadily improving in quality on tasks such as language modeling and biological sequence design [29, 41, 46, 1, 43]. However, modern discrete diffusion—especially the most performant kind that relies on masking [32, 41, 48]—lacks the fundamental ability to iteratively refine outputs. Once a discrete token is unmasked, it cannot be updated again, even if it introduces an error. The lack of error correction in turn sets limits on controllable generation, sampling speed, and sample quality.

Here, we address the inability of masked diffusion models to perform iterative refinement by introducing a new sampler that supports remasking during generation. Our method, the remasking diffusion model (ReMDM) sampler, is simple and allows users to directly specify the probability of remasking a token at each time step. We augment the sampler with components that range from nucleus sampling to remasking schedules, significantly boosting performance.

Our approach for deriving the sampler is rooted in probabilistic modeling. We show that our method corresponds to ancestral sampling in a discrete diffusion model (also called ReMDM) characterized

39th Conference on Neural Information Processing Systems (NeurIPS 2025).

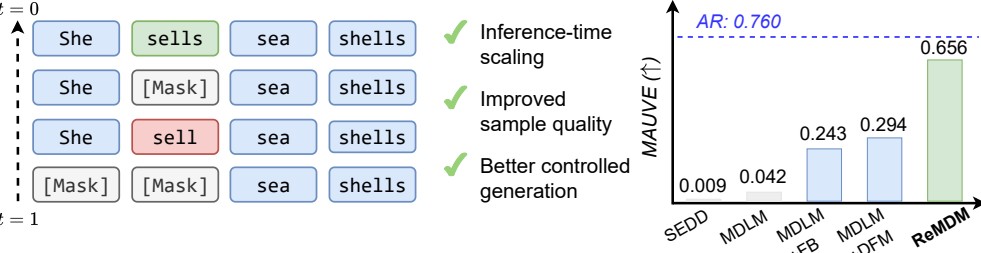

Figure 1: Our family of masked diffusion processes allow for more flexible generation with remasking of already decoded tokens. This improves sample quality and further closes the gap to AR models. *(Left)* An illustrative example of errors fixed by ReMDM. The first two tokens can be "They sell" or "She sells", but due to the independence of the parallelized decoding processes, "She sell" is decoded. Such mistakes can be corrected by remasking samplers. *(Right)* MAUVE scores on OpenWebText. MDLM is from Sahoo et al. [41]. FB and DFM denote the forward-backward [5] and discrete flow matching [13] correctors, respectively.

by a remasking backward process. The ReMDM model admits an objective that is a rescaled version of the MDLM objective [41]. This suggests using the ReMDM sampler on top of pretrained MDLMs, which we find to work well. We complement our analysis of the sampler by also showing that it can be interpreted as a predictor-corrector technique [53, 5]. Notably, we demonstrate that ReMDM is theoretically more general than previous corrector samplers for masked diffusion models [5, 13].

Most interestingly, the ReMDM sampler endows discrete diffusion with a form of inference-time compute scaling. By increasing the number of denoising steps on language modeling tasks, ReMDM generates samples with significantly higher sample quality metrics than any previous diffusion model and almost matches the performance of an autoregressive (AR) model with the same architecture. Conversely, when we reduce the sampling steps to increase speed, ReMDM's performance degrades less than other samplers. On image generation, ReMDM scales better with inference-time compute than vanilla masked diffusion models like MaskGiT [7]. Additionally, ReMDM models are more amenable to guidance [46]—in molecule design experiments, ReMDM pushes the Pareto frontier of novelty and the desired molecular property relative to alternatives relying on masking or uniform noise diffusion.

**Contributions** We summarize our contributions as follows:

1. We introduce the remasking diffusion model (ReMDM) sampler that yields performant iterative refinement via remasking to masked diffusion models.

2. We show that our method is a form of ancestral sampling in a probabilistic model whose ELBO is similar to that of classical masked diffusion. This analysis suggests using our sampler on top of pretrained models, which we find to work well.

3. Across the domains of natural language, discretized images, and molecule string representations, we demonstrate empirically that ReMDM endows masked diffusion with inference-time scaling that improves sample quality with more computation and enhances controlled generation and downstream performance.

## 2 Background

**Discrete Diffusion Models** Diffusion models [50, 18] are characterized by parametric models $p_\theta$ trained to reverse a fixed noising process $q$ that maps clean data $\mathbf{x}$ to increasingly noisy latent variables $\mathbf{z}_t$, for $t \in [0, 1]$. Austin et al. [2], Sahoo et al. [41], Shi et al. [48] extend this framework to discrete signals. Formally, we define $\mathbf{x} \in \{0, 1\}^{|V|} \subset \Delta^{|V|}$ as a one-hot vector representing a token, where $|V|$ is the vocabulary size and $\Delta^{|V|}$ is the simplex over this vocabulary. For finite-time processes, we let $T$ be the number of time steps. For each time step $t(i) = \frac{i}{T} \in [0, 1]$ (i.e., for $i = 0, 1, \ldots, T$), each intermediate latent variable $\mathbf{z}_{t(i)} \in \{0, 1\}^{|V|}$ has the marginal:

$$q(\mathbf{z}_t \mid \mathbf{x}) = \text{Cat}(\mathbf{z}_t; \alpha_t \mathbf{x} + (1 - \alpha_t)\boldsymbol{\pi}) \in \Delta^{|V|}, \tag{1}$$

where $\boldsymbol{\pi}$ is a user-specified prior and where we have dropped the explicit dependence of $t$ on $i$. The predefined schedule $\alpha_t \in [0, 1]$ is monotonically decreasing in $t$. For sequences of $L$ tokens, we denote clean / noisy sequences as $\mathbf{x}^{(1:L)}$ / $\mathbf{z}_t^{(1:L)}$ and each token $\ell \in \{1, \dots, L\}$ as $\mathbf{x}^{(\ell)}$ / $\mathbf{z}_t^{(\ell)}$.

**Masked Diffusion Language Models** A special case of this formulation, known as 'masked' or 'absorbing state' diffusion, sets the limiting distribution $\boldsymbol{\pi} = \boldsymbol{m}$, a one-hot representation of a special [MASK] token. With $s(i) = \frac{i-1}{T}$ as the time step directly preceding $t$, these posteriors take the form:

$$q(\mathbf{z}_s \mid \mathbf{z}_t, \mathbf{x}) = \mathrm{Cat}\Big(\mathbf{z}_s; \frac{\alpha_s - \alpha_t}{1 - \alpha_t}\mathbf{x} + \frac{1 - \alpha_s}{1 - \alpha_t}\mathbf{z}_t\Big). \tag{2}$$

Following Sahoo et al. [41], we omit the subscript $(i)$ in $s(i)$ and $t(i)$ in the rest of this paper. Importantly, despite the recent success of MDLM [41] and other absorbing state discrete diffusion models, these models suffer from a key drawback, which we call the **failure to remask** property. Namely, the posterior formula in (2) implies that any unmasked token $\mathbf{z}_t \neq \boldsymbol{m}$ must remain unchanged throughout the entire denoising process. Formally, in masked diffusion models, equation 1 ensures that any token $\mathbf{x}$ only possesses two possible values: unmasked, i.e., keeping unchanged, and masked, i.e., $\boldsymbol{m}$, which means that in the reverse process, for any token $\mathbf{z}_t \neq \boldsymbol{m}$, for all $s \leq t$, plugging $\mathbf{z}_t = \mathbf{x}$ into equation 2 leads to $q(\mathbf{z}_s \mid \mathbf{z}_t, \mathbf{x}) = \mathrm{Cat}(\mathbf{z}_s; \mathbf{z}_t)$. Intuitively, once a token is decoded, this prediction is 'locked-in', a limitation that these diffusion models share with AR models.

As in Sahoo et al. [41], a denoising neural network $\mathbf{x}_\theta$ is used for parameterizing $p_\theta(\mathbf{z}_s|\mathbf{z}_t) = q(\mathbf{z}_s|\mathbf{z}_t, \mathbf{x}_\theta(\mathbf{z}_t))$ and trained using the following variational objective (see the derivation in Sahoo et al. [41]):

$$\mathcal{L} = \mathbb{E}_{\mathbf{z}_0 \sim q(\mathbf{z}_0|\mathbf{x})}\Big[ -\log p_\theta(\mathbf{x} \mid \mathbf{z}_0)\Big] + \mathbb{E}_{t \in \{\frac{1}{T}, \dots, 1\}} \mathbb{E}_{\mathbf{z}_t \sim q(\mathbf{z}_t|\mathbf{x})} T\Big[ \frac{\alpha_t - \alpha_s}{1 - \alpha_t} \log(\mathbf{x}_\theta(\mathbf{z}_t)^\top \mathbf{x})\Big] \tag{3}$$

## 3 Remasking Diffusion Models

In this work, we alleviate the failure to remask limitation that affects state-of-the-art masked diffusion models. Our strategy is to design a probabilistic model similar to MDLM [41], but whose denoising posterior generalizes (2) and supports remasking.

Specifically, we will define a discrete diffusion model with posterior for $\mathbf{z}_t \neq \boldsymbol{m}$ given by:

$$q(\mathbf{z}_s \mid \mathbf{z}_t = \mathbf{x}, \mathbf{x}) = (1 - \sigma_t)\mathbf{x} + \sigma_t \boldsymbol{m}. \tag{4}$$

The parameter $\sigma_t$ gives us the flexibility of remasking a decoded token during generation. Because of this property, we dub our method **ReM**asking **D**iffusion **M**odels (**ReMDM**).

### 3.1 ReMDM Probability Decomposition

We define the probability decomposition of ReMDM as:

$$q_\sigma(\mathbf{z}_{0:1} \mid \mathbf{x}) = q_\sigma(\mathbf{z}_1 \mid \mathbf{x}) \prod_{i=1}^{T} q_\sigma(\mathbf{z}_s \mid \mathbf{z}_t, \mathbf{x}) \tag{5}$$

where $q_\sigma(\mathbf{z}_1 \mid \mathbf{x}) = q(\mathbf{z}_1) = \boldsymbol{m}$, as in masked diffusion. We construct the posteriors as:

$$q_\sigma(\mathbf{z}_s \mid \mathbf{z}_t, \mathbf{x}) = \begin{cases} \mathrm{Cat}(\mathbf{z}_s; (1 - \sigma_t)\mathbf{x} + \sigma_t \boldsymbol{m}), & \mathbf{z}_t \neq \boldsymbol{m} \\ \mathrm{Cat}(\mathbf{z}_s; \frac{\alpha_s - (1 - \sigma_t)\alpha_t}{1 - \alpha_t}\mathbf{x} + \frac{1 - \alpha_s - \sigma_t \alpha_t}{1 - \alpha_t}\boldsymbol{m}), & \mathbf{z}_t = \boldsymbol{m}. \end{cases} \tag{6}$$

Observe how (6) maintains the remasking property laid out in (4). Additionally, our chosen form for $q(\mathbf{z}_s \mid \mathbf{z}_t = \boldsymbol{m}, \mathbf{x})$ ensures that the marginals for $q_\sigma(\mathbf{z}_t \mid \mathbf{x})$ are the same as in classical masked diffusion (e.g., Sahoo et al. [41]). This property will yield an ELBO for ReMDM that is very similar to that of MDLM, and will help us argue for using the ReMDM sampling procedure on top of a pretrained MDLM model. Formally, we establish the following result (proof in Appendix A.1):

**Theorem 3.1.** *Given the posterior in (6), the marginal distribution $q_\sigma(\mathbf{z}_t \mid \mathbf{x})$ is the same as in (1).*

In Appendix A.2, we demonstrate that this ability to remask necessitates that ReMDM is a non-Markovian process (as in e.g., DDIM [51]). Intuitively, this is necessary because the latents $\mathbf{z}_t$ can transition from the masked state $m$ back to $\mathbf{x}$, and therefore the transition kernel needs to be conditioned on $\mathbf{x}$.

**Understanding the Role of** $\sigma_t$   To ensure that the posterior in (6) is a valid probability distribution, we place following constraints on $\sigma_t$ (see derivation in Appendix A.3):

$$0 \leq \sigma_t \leq \min\left\{1, \frac{1-\alpha_s}{\alpha_t}\right\} =: \sigma_t^{max}. \tag{7}$$

Further examining (6), we see that as $\sigma_t$ increases, we move mass away from $\mathbf{z}_t$. Importantly, this means that when $\mathbf{z}_t$ is unmasked, $\sigma_t$ directly controls the probability of remasking. When $\sigma_t = 0$, we recover the posterior from MDLM in (2). Thus, ReMDM can be seen as a more general formulation that admits MDLM as a special case.

### 3.2   Negative Evidence Lower Bound

We define the joint distribution of the parameterized forward process $p_\theta$ as follows:

$$p_\theta(\mathbf{x})p_\theta(\mathbf{z}_{0:1} \mid \mathbf{x}) = p_\theta(\mathbf{z}_1)p_\theta(\mathbf{x} \mid \mathbf{z}_0) \prod_{i=1}^{T} p_\theta(\mathbf{z}_s \mid \mathbf{z}_t). \tag{8}$$

We define $p_\theta(\mathbf{z}_1) = \boldsymbol{\pi}$ and parameterize $p_\theta(\mathbf{z}_s \mid \mathbf{z}_t) = q(\mathbf{z}_s \mid \mathbf{z}_t, \mathbf{x}_\theta(\mathbf{z}_t))$, where $\mathbf{x}_\theta(\mathbf{z}_t)$ is a denoising model.

With this parameterization, the NELBO for ReMDM is (see Appendix A.4 for details):

$$\mathcal{L}^\sigma = \mathbb{E}_{\mathbf{z}_0 \sim q(\mathbf{z}_0|\mathbf{x})}\left[-\log p_\theta(\mathbf{x} \mid \mathbf{z}_0)\right] + \mathbb{E}_{t \in \{\frac{1}{T}, \dots, 1\}}\mathbb{E}_{\mathbf{z}_t \sim q(\mathbf{z}_t|\mathbf{x})}T\left[\frac{(1-\sigma_t)\alpha_t - \alpha_s}{1-\alpha_t}\log(\mathbf{x}_\theta^\top \mathbf{x})\right]. \tag{9}$$

Since $\alpha_t \leq \alpha_s$, each term in the second expectation increases monotonically in $\sigma_t$. When $\sigma_t = 0$, for all $t$, we recover the MDLM objective from (3). This implies that the MDLM NELBO produces a tighter bound compared to ReMDM. Furthermore, since the diffusion loss term in (9) is simply a reweighted version of that in (3), one can presumably reuse weights from a model trained with (3), i.e., using different $\sigma_t$ at training and inference. Indeed, we find this to be a performant strategy in practice. Moreover, we find the performance (test perplexity) between models trained with the MDLM and the ReMDM objectives to be comparable (see Appendix E.1), further justifying our reuse of pretrained weights, with the added benefit of more flexible sampling unlocked by ReMDM.

## 4   ReMDM Samplers

In practice, we recommend using ReMDM with a different $\sigma_t$ at training and inference, just as one might switch to a different noise schedule during sampling. In particular, using $\sigma_t = 0$ for training is equivalent to combining the ReMDM sampler with a pretrained MDLM model $\mathbf{x}_\theta$, which works well and does not require re-training $\mathbf{x}_\theta$.

Using ReMDM for sampling opens a large design space for choosing $\sigma_t$. In the following, we explore several practical design strategies for $\sigma_t$ that significantly improve performance, and in Section 5 we describe further add-ons to the sampler (e.g., nucleus sampling) that also improve quality. The high-level pseudocode for the ReMDM sampler is provided in Algorithm 1, with more detailed algorithms for implementing the schedules below deferred to Appendix C.

### 4.1   Design Strategies for the Remasking Schedule $\sigma_t$

Here, we list several strategies for setting $\sigma_t \in [0, \sigma_t^{max}]$, which can either be employed separately or in conjunction with the strategies introduced in Section 4.2. For sequences, each token can have a different $\sigma_t^{(\ell)}$. When we omit this superscript, this implies that the same $\sigma_t$ is used for all tokens.

**Max-Capped Schedule**   We can potentially reduce the maximum probability of remasking to a constant $\eta_{cap} \in [0, 1]$. Concretely, we let $\sigma_t = \min\{\eta_{cap}, \frac{1-\alpha_s}{\alpha_t}\}$, for all $t \in [0, 1]$. We denote this schedule as "ReMDM-cap."

**Algorithm 1** Sampling with ReMDM.

---

// Differences to standard MDLM sampling noted in brown.
**Input:** pre-trained denoising network $\mathbf{x}_\theta$ (e.g., MDLM), number of timesteps $T$, noise schedule $\alpha_t$, remasking schedule $\sigma_t$.
Initialize $\mathbf{z}_t = \boldsymbol{m}$.
**for** $i = T$ **to** $1$ **do**
   $t = i/T, s = (i-1)/T$.
   Set $\alpha_t, \alpha_s$ according to noise schedule.
   Set $\sigma_t \in [0, \sigma_t^{max}]$ according to remasking schedule.
   Compute approximate posterior:

$$p_\theta(\mathbf{z}_s \mid \mathbf{z}_t) = q_\sigma(\mathbf{z}_s \mid \mathbf{z}_t, \mathbf{x} = \mathbf{x}_\theta(\mathbf{z}_t)) = \begin{cases} \mathrm{Cat}(\mathbf{z}_s; (1 - \sigma_t)\mathbf{x}_\theta + \sigma_t \boldsymbol{m}), & \mathbf{z}_t \neq \boldsymbol{m} \\ \mathrm{Cat}(\mathbf{z}_s; \frac{\alpha_s - (1-\sigma_t)\alpha_t}{1 - \alpha_t}\mathbf{x}_\theta + \frac{1 - \alpha_s - \sigma_t \alpha_t}{1 - \alpha_t}\boldsymbol{m}), & \mathbf{z}_t = \boldsymbol{m} \end{cases}$$

   Sample $\mathbf{z}_s \sim p_\theta$.
   Set $\mathbf{z}_t = \mathbf{z}_s$.
**end for**
**Output:** $\mathbf{z}_t$.

---

**Rescaled Schedule** Alternatively, we can temper the chances of remasking by setting $\sigma_t = \eta_{rescale} \cdot \sigma_t^{max}$, with $\eta_{rescale} \in [0, 1]$ as a hyperparameter that controls this rescaling. We denote this schedule as "ReMDM-rescale."

**Confidence-Based Schedule** In conjunction with the two strategies above, we explore a further reweighing of $\sigma_t$ which is based on the intuition that tokens of which the denoising model is less confident should be assigned a larger probability of remasking. Consider the $\ell$-th token in a sequence of $L$ latents at time $t$. For each $\ell \in \{1, \ldots, L\}$, we store its decoding probability at the time $\tau$ at which it was last unmasked. Concretely, if $\mathbf{z}_t^{(\ell)} \neq \boldsymbol{m}$, then we define $\psi_t^{(\ell)} := \mathbf{x}_{\theta,\tau}^{(\ell)\top} \mathbf{z}_\tau^{(\ell)}$. If $\mathbf{z}_t = \boldsymbol{m}$, then $\psi_t^{(\ell)} := \infty$. Thus, $\psi_t^{(\ell)}$ serves as a 'confidence score' for unmasked tokens. We then compute

$$\sigma_t^{(\ell)} = \eta_{conf}^{(\ell)} \cdot \sigma_t \ , \text{ where } \ \eta_{conf}^{(\ell)} = \frac{\exp(-\psi_t^{(\ell)})}{\sum_{l=1}^{L} \exp(-\psi_t^{(\ell')})}.$$

With this schedule, masked tokens are decoded using the approximate posterior from MDLM, and the unmasked tokens are remasked negatively proportional to their confidence. We denote this schedule as "ReMDM-conf."

## 4.2 Design Strategies for 'Turning On' ReMDM

There may be certain periods of the generation process where remasking is more valuable and some when it is detrimental / can slow down sampling, e.g., at the beginning of sampling, one may wish to generate some tokens in a sequence using standard MDLM decoding, and only after generating a reasonable starting candidate, then spend some computational budget 'fixing mistakes' via the remasking posterior. We, therefore, propose two methods for optionally 'turning on/off' ReMDM sampling, which amount to the following modification of the $\sigma_t$ schedules above:

$$\tilde{\sigma}_t = \begin{cases} \sigma_t, & \text{if } t \in [t_{on}, t_{off}), \text{with } t_{on} > t_{off} \\ 0, & \text{otherwise.} \end{cases} \tag{10}$$

**Switch** We choose some $t_{switch} \in (0, 1]$ and in (10) we have $[t_{on}, t_{off}) = [t_{switch}, 0)$. We denote this strategy as "ReMDM-switch."

**Loop** In this strategy, we set both $t_{on}, t_{off} \in (0, 1]$. Furthermore, in the range when ReMDM is activated, we modify the noise schedule to be constant, such that $\alpha_t = \alpha(t_{on})$. As shown in Figure 2, this divides the sampling process into three phases. In the first phase, the model generates tokens without remasking ($\sigma_t = 0$, i.e., using MDLM). In the second phase, we hold $\alpha$ constant (i.e., $\alpha_s = \alpha_t$), and the model is asked to remask and predict a proportion of the generated tokens in a loop. In particular, one can use any ReMDM strategy introduced in Section 4.1 in this phase. Intuitively, if

a 'bad' token is remasked, it will likely be replaced with a 'good' token due to the abundant context. Even if a 'good' token happens to be remasked, since the signal-to-noise ratio is designed to be high in this portion of the generation process, it will likely be re-decoded as other (if not the same) 'good' tokens. In this way, ReMDM-loop corrects the 'bad' tokens and maintains the 'good' tokens, i.e., fixes mistakes. Finally, in the third phase, we let the model predict any remaining unmasked tokens using the MDLM posterior. We denote this strategy as "ReMDM-loop."

### 4.3 Comparison with Discrete Predictor-Corrector Samplers

Previous works, e.g., the forward-backward (FB; Campbell et al. [6]) and discrete flow matching (DFM; Gat et al. [13]) correctors, propose to tackle the failure to remask property with discrete predictor-corrector samplers, a special type of discrete diffusion samplers that decompose a single sampling step into one predictor step followed by a certain number of corrector steps that remediate possible mistakes without changing the marginals. Here, we demonstrate that these methods are special cases of ReMDM sampling by first noting the following result (see Appendix A.5 for the proof):

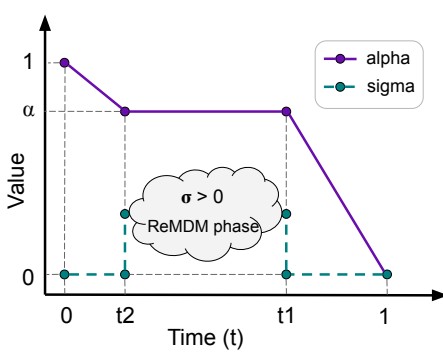

Figure 2: ReMDM-loop illustration.

**Theorem 4.1.** $q_\sigma(\mathbf{z}_s \mid \mathbf{z}_t, \mathbf{x})$ from (6) is equivalent to an MDLM predictor step $q_{predictor}(\mathbf{z}_{s_p} \mid \mathbf{z}_t, \mathbf{x})$ followed by a corrector step $q_{corrector}(\mathbf{z}_{s_c} \mid \mathbf{z}_{s_p}, \mathbf{x})$ in the form of

$$\begin{cases} \mathrm{Cat}(\mathbf{z}_{s_c}; (1 - \sigma_t)\mathbf{x} + \sigma_t \boldsymbol{m}), & \mathbf{z}_{s_p} \neq \boldsymbol{m} \\ \mathrm{Cat}(\mathbf{z}_{s_c}; \frac{\sigma_t \alpha_s}{1 - \alpha_s}\mathbf{x} + \frac{1 - (1 + \sigma_t)\alpha_s}{1 - \alpha_s}\boldsymbol{m}), & \mathbf{z}_{s_p} = \boldsymbol{m}. \end{cases} \quad (11)$$

We now formalize the generality of ReMDM (proofs provided in Appendices A.6, A.7, and A.8).

**Proposition 4.2.** *The FB corrector [5] on MDLM is a special case of ReMDM where $\sigma_t = \frac{\alpha_s - \alpha_t}{\alpha_t}$.*

**Proposition 4.3.** *A DFM corrector [13] on MDLM can be converted to a ReMDM sampler where $\sigma_t = \frac{\beta_t(\alpha_s - \alpha_t)}{\alpha_t}$. $\beta_t \in \mathbb{R}$ denotes the DFM corrector schedule.*

**Proposition 4.4.** *The ReMDM sampler is more general than DFM corrector, since only the former can accommodate noise schedules with constant $\alpha_t$ for some range $[t, t - \Delta t]$ where $\Delta t > 0$.*

## 5 Experiments

### 5.1 ReMDM Improves Sample Quality

#### 5.1.1 Text Generation

**Experimental Setup** We test the text generation capability of ReMDM with unconditional generation from models trained on OpenWebText (OWT; Gokaslan & Cohen [14]). The OWT dataset was tokenized using the `gpt-2` tokenizer [38] and sequences were wrapped to a max length of $L = 1024$. Our baselines include AR, SEDD [29], MDLM, and MDLM in conjunction with the FB [5] and DFM [13]

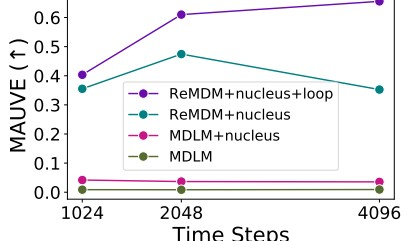

Figure 3: ReMDM ablation for OWT generation quality.

correctors during inference. For all the experiments, we reuse the checkpoints trained by Sahoo et al. [41] for fair comparison. Following Pillutla [36], we generate 5,000 samples from each model/sampler and compute the MAUVE score [28, 37], as this metric balances sample quality and diversity.

**Generative Perplexity vs. MAUVE** It is common practice to report generative perplexity (Gen PPL.) as a quality metric for text generation. However, for corrector samplers such as ReMDM, we find this metric can be uninformative. In line with previous work [68], we observe that one can get arbitrarily low Gen PPL. by tuning corrector schedule hyperparameters (see Appendix E.2.2) to achieve low Gen PPL. while sacrificing sentence entropy/diversity, with an extreme case of generating a sequence of repetitive words like "love love love..." and getting a Gen PPL. of 1.5 (see Appendix F.1). Given this

phenomenon, we choose MAUVE as the primary quality metric since MAUVE measures the balance between generative perplexity and diversity (see Pillutla et al. [37] for a more detailed discussion).

**Floating-Point Precision**  As indicated in Zheng et al. [68], previous masked diffusion models [29, 41] report sampling with 32-bit floating point precision, which was found to significantly curb the diversity of generated samples. We therefore use 64-bit floating point for all text-sampling experiments.

**Nucleus Sampling**  We observe that nucleus sampling [19] is critical to generating high-quality text sequences. We therefore use this strategy (with $top\text{-}p = 0.9$) for all models (except SEDD, as it does not output logits).

**Results**  In Table 1, we report results for inference-time scaling ($T \geq L$) and faster sampling ($T < L$). Note that we report Gen PPL. and entropy for completeness and that lower Gen PPL. does not always correlate with better quality, as noted above. For inference-time scaling, we use the max-capped schedule ($\eta_{cap} = 0.02$) in conjunction with the loop strategy ($t_{on} = 0.55, t_{off} = 0.05$, and $\alpha(t_{on}) = 0.9$). For faster sampling, we use the max-capped schedule ($\eta_{cap} = 0.04$) on its own.

As shown in Table 1, ReMDM scales favorably with inference-time compute, achieving a $15.62\times$ MAUVE score compared to masked diffusion models and a $2.23\times$ MAUVE score compared to MDLM with corrector samplers. In contrast, masked diffusion models scale poorly with $T$, and corrector sampler methods saturate when $T$ is large. For faster sampling where a more limited computational budget is available, ReMDM is able to maintain sample quality better than baselines.

**Ablation: Function of Each Component**  In Figure 3, we sequentially ablate each building block of ReMDM and report the MAUVE score of samples. Starting with MDLM, we see that each of our proposed sampling improvements increases the MAUVE score, with the largest improvement coming from remasking ability of ReMDM.

Table 1: ReMDM improves sample quality in inference-time scaling and faster sampling on OWT. ReMDM outperforms state-of-the-art masked diffusion models (SEDD; [29], MDLM; [41]) and masked diffusion models with corrector samplers such as Forward-Backward (FB; [5]) and Discrete Flow Matching (DFM; [13]) corrector samplers. $^\dagger$ indicates nucleus sampling. For each $T$, the best diffusion MAUVE score is **bolded**.

| Method | MAUVE (↑) | | | Gen PPL. (↓) | | | Entropy (↑) | | |
|---|---|---|---|---|---|---|---|---|---|
| Data | 1.00 | | | 14.8 | | | 5.44 | | |
| AR *(T=1024)* $^\dagger$ | 0.760 | | | 12.1 | | | 5.22 | | |
| | *T=1024* | *T=2048* | *T=4096* | *T=1024* | *T=2048* | *T=4096* | *T=1024* | *T=2048* | *T=4096* |
| SEDD (absorb) | 0.008 | 0.008 | 0.009 | 104.7 | 103.2 | 102.5 | 5.62 | 5.61 | 5.61 |
| MDLM$^\dagger$ | 0.042 | 0.037 | 0.035 | 51.3 | 51.3 | 50.9 | 5.46 | 5.46 | 5.45 |
| MDLM+FB$^\dagger$ | 0.133 | 0.197 | 0.243 | 33.8 | 28.6 | 22.8 | 5.35 | 5.28 | 5.18 |
| MDLM+DFM$^\dagger$ | 0.254 | 0.294 | 0.269 | 21.7 | 21.0 | 20.7 | 5.20 | 5.19 | 5.17 |
| ReMDM$^\dagger$ | **0.403** | **0.610** | **0.656** | 28.6 | 22.8 | 17.6 | 5.38 | 5.30 | 5.20 |
| | *T=128* | *T=256* | *T=512* | *T=128* | *T=256* | *T=512* | *T=128* | *T=256* | *T=512* |
| SEDD (absorb) | 0.007 | 0.007 | 0.008 | 119.2 | 110.1 | 107.2 | 5.65 | 5.63 | 5.62 |
| MDLM$^\dagger$ | 0.015 | 0.023 | 0.031 | 61.5 | 55.8 | 53.0 | 5.52 | 5.49 | 5.48 |
| MDLM+FB$^\dagger$ | **0.064** | 0.084 | 0.100 | 42.8 | 39.6 | 37.1 | 5.44 | 5.41 | 5.38 |
| MDLM+DFM$^\dagger$ | 0.041 | 0.144 | 0.211 | 37.9 | 26.5 | 23.3 | 5.31 | 5.26 | 5.23 |
| ReMDM$^\dagger$ | 0.057 | **0.216** | **0.350** | 42.5 | 30.5 | 21.1 | 5.43 | 5.34 | 5.21 |

### 5.1.2 Image Generation

**Experimental Setup**  We test ReMDM's class-conditioned image generation ability. Concretely, we use a pretrained MaskGiT [7] that was trained on ImageNet [8] samples with $256 \times 256$ pixels. The images were patchified and flattened to a sequence of $L = 256$ and encoded according to a codebook with 1024 tokens [10]. See Chang et al. [7] for more details about the model and training settings. We compare ReMDM's sampler to the original MaskGiT sampling and to MDLM. Note that MaskGiT does not follow an ancestral sampling method but rather relies on an effective heuristic where masked tokens are decoded proportional to model confidence. However, MaskGiT is still restricted by the failure to remask property. For each sampler, we conditionally generate 50,000 images, with class

labels randomly chosen from the 1,000 ImageNet categories, and we measure sample quality using Fréchet Inception Distance (FID; [16]) and Inception Score (IS; [45]).

**Results** We report the best results for each method in Table 2 (temperature=1.0 for MaskGiT, and temperature=0.8 for MDLM and ReMDM). We see that ReMDM has the best scaling, producing the highest quality images of the three methods at $T = 64$.

**Ablation: Max-capped vs. Rescaled schedules** In Appendix E.3, we present the full ReMDM parameter search results. We find that ReMDM-rescale slightly outperforms ReMDM-cap, and that for both strategies, increasing their corresponding $\eta$ parameters leads to improved results.

Table 2: ReMDM produces the highest quality images on ImageNet conditional generation. For each metric and $T$, the best value is **bolded**.

|  | Sampler | $T = 16$ | $T = 32$ | $T = 64$ |
|---|---|---|---|---|
| FID ($\downarrow$) | MaskGiT | **6.74** | **4.92** | 4.85 |
|  | MDLM | 7.88 | 5.37 | 4.69 |
|  | ReMDM | 7.40 | **4.92** | **4.45** |
| IS ($\uparrow$) | MaskGiT | **155.32** | 181.57 | 196.38 |
|  | MDLM | 140.97 | 169.79 | 187.93 |
|  | ReMDM | 145.27 | **182.05** | **209.45** |

## 5.2 ReMDM Improves Guidance

**Experimental Setup** We follow the setup from Schiff et al. [46] to explore controlled small molecule generation. Specifically, we use the QM9 dataset [40, 39] of ~133k molecules and their character-based SMILES string representations [59]. Sequences were tokenized using a regular expression method [47] and padded to a maximum length of $L = 32$. We use the D-CFG and D-CBG methods defined in Schiff et al. [46] to conditionally generate molecules with higher ring counts (greater than $90^{\text{th}}$ percentile in the original dataset). We vary the guidance strength $\gamma \in \{1, 2, 3, 4, 5\}$ and scale inference steps $T \in \{32, 64, 128\}$. Our baselines consist of an AR model guided using either CFG or the popular classifier-based FUDGE method [62], MDLM, and the uniform diffusion language model (UDLM) proposed in Schiff et al. [46], which was also introduced to alleviate the no remasking limitation of masked diffusion. For ReMDM sampling, we use pretrained MDLM models. We explore the various strategies defined in Section 4 (see Appendix E.4 for full search results). After generating 1,024 sequences, we use the RDKit library [23] to determine whether sequences are valid and compute novelty of the generated sequences (number of unique valid sequences that do not appear in the original QM9 dataset) and mean ring count of novel sequences.

**Results** In Figure 4, we display the trade-off that comes from increasing guidance strength $\gamma$. We only visualize results for samples that had at least 50 novel sequences. For both forms of guidance, CFG and CBG, ReMDM outperforms AR and diffusion approaches, pushing the novelty-property maximization frontier beyond that of the baseline methods. Additionally, ReMDM scales favorably with more inference-time compute, seen by the curves for larger $T$ dominating those for smaller $T$. For ReMDM, we found the best strategy to be a combination of the ReMDM-rescale (with $\eta_{rescale} = 0.9$) and ReMDM-conf schedules. For D-CFG, we use the loop strategy, with $t_{on} = 0.25, t_{off} = 0.125$. In addition to these findings, in Appendix E.4.1, we present experimental results for maximizing a different property of interest, drug-likeness (QED; [3]).

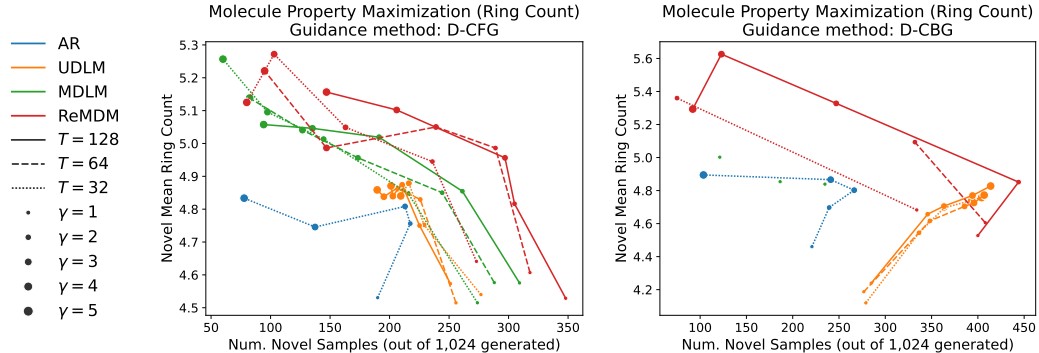

Figure 4: ReMDM improves steer-ability by extending the novelty-property maximization frontier. Lines show controlled generation for ring count maximization on QM9 dataset, varying compute $T$ and guidance strength $\gamma$. Curves in the upper right region are better. *(Left)* Discrete classifier-free guidance (D-CFG). *(Right)* Discrete classifier-based guidance (D-CBG) and FUDGE for AR.

**Ablation: Importance of using Confidence-based Schedule** Our parameter tuning revealed that using ReMDM-rescale in conjunction with the confidence-based scheduler led to improved results (see Appendix E.4.3).

### 5.3 ReMDM Improves Benchmark Performance

**Experimental Setup** Recent diffusion large language models (dLLMs) such as LLaDA 8B [31] and Dream 7B [64] have demonstrated on par downstream task performance with autoregressive language models of the same parameter scale. However, these dLLMs still suffer from the failure to remask property. To test the effect of remasking sampling on dLLLMs, we take LLaDA 8B Instruct and apply DFM and ReMDM samplers (see Appendix D.4 for sampler details). Specifically, we use Countdown (bidirectional reasoning) [63] and TruthfulQA (factual knowledge grasp) [27] as benchmarking tasks. In Countdown, the model is given four integers and asked to develop equations where first three integers produce the fourth using addition, subtraction, multiplication and division operations. In TruthfulQA, the model is asked 817 questions on factual knowledge. We calculate the ROUGE [26] scores between the model and correct answers, as well as the model and plausibly wrong answers. The difference between the right and wrong ROUGE scores is reported. For both of the tasks, we use max-length $L = 32$ and no semi-autoregressive sampling [31]. See Appendix D.4 & F.3 for more setting details.

**Results** We repeat experiments with multiple random seeds to report mean values and 95% confidence intervals. In Table 3, LLaDA with ReMDM consistently performs the best and exceeds the original LLaDA with statistical significance.

## 6 Discussion and Related Work

**Denoising Diffusion Implicit Models** An analogy can be drawn between our work and DDIM [51], where ReMDM is to MDLM for discrete signals as DDIM is to DDPM [18] for continuous data. Both our work and DDIM propose alternative forward and backward processes that maintain marginals while

Table 3: LLaDA with ReMDM performs the best on downstream tasks, and consistently exceeds the original LLaDA with statistical significance. For each metric, the best value is **bolded.**

| Metric | Countdown pass@1% | TruthfulQA $\Delta$ ROUGE 1/2/L |
|---|---|---|
| LLaDA | $45.2_{\pm 0.2}$ | $27.1_{\pm 0.4}$ / $30.1_{\pm 0.4}$ / $27.2_{\pm 0.4}$ |
| LLaDA-DFM | $44.8_{\pm 0.2}$ | $28.2_{\pm 0.4}$ / $31.1_{\pm 0.4}$ / $28.3_{\pm 0.4}$ |
| LLaDA-ReMDM | $\mathbf{46.1}_{\pm 0.2}$ | $\mathbf{29.5}_{\pm 0.4}$ / $\mathbf{31.8}_{\pm 0.4}$ / $\mathbf{29.5}_{\pm 0.3}$ |

enabling more flexible generation. Of note, Song et al. [51] also present a way of adapting processes to discrete domains; however, they focus on uniform categorical as the limiting distribution. In Appendix B, we demonstrate how this can be adapted for absorbing state diffusion and derive an equivalence to our method.

**Discrete Predictor-Corrector Samplers** Continuous time Markov chain (CTMC) theory provides a framework for samplers that correct errors in the reverse process, extending the original predictor-corrector formulation for continuous data [53] to discrete domains. In addition to the FB and DFM correctors discussed above, other correctors include: the Stein operator [67], and DPC [24] which improves the sample quality of MaskGiT by corrector sampling. Unlike the plug-and-play methods of FB, DFM, and ReMDM, the Stein operator and DPC correctors require additional training.

## 7 Conclusion

In this work, we have presented a novel family of absorbing state discrete diffusion samplers. Our method leverages the strong language modeling performance of this class of models by enabling the use of pretrained weights with the added benefit of more flexible sampling strategies that allow for remasking of predicted tokens. We demonstrate empirically that this leads to improved sample quality for both unconditional and conditional generation and leads to better controlled generation and downstream performance. Our approach also unlocks an important inference-time compute scaling axis that is more limited for existing masked diffusion models.

# Acknowledgments

This work was partially funded by the National Science Foundation under awards DGE-1922551, CAREER awards 2046760 and 2145577, and by the National Institute of Health under award MIRA R35GM151243. We gratefully acknowledge use of the research computing resources of the Empire AI Consortium, Inc, with support from Empire State Development of the State of New York, the Simons Foundation, and the Secunda Family Foundation. This research was also supported in part through the use of computational resources provided by Lambda (`lambda.ai`) in partnership with Open Athena AI Foundation, Inc. We gratefully acknowledge their generous GPU infrastructure grants that helped make this work possible.

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

# Contents

## A  Theoretical Results

### A.1  Proof of Theorem 3.1

*Proof.* We proceed by induction. The starting point $q(\mathbf{z}_1 \mid \mathbf{x}) = \text{Cat}(\mathbf{z}_1; \alpha_1 \mathbf{x} + (1 - \alpha_1)\boldsymbol{m})$ is true by design. For the induction hypothesis, assume that for any $t < 1$, $q(\mathbf{z}_t \mid \mathbf{x}) = \text{Cat}(\mathbf{z}_t; \alpha_t \mathbf{x} + (1 - \alpha_t)\boldsymbol{m})$. Recall that in absorbing state processes, given $\mathbf{x}$, for all $t \in [0, 1]$, $\mathbf{z}_t \in \{\mathbf{x}, \boldsymbol{m}\}$. Then for the previous time step $s$, we have:

$$q(\mathbf{z}_s = \mathbf{x} \mid \mathbf{x}) = q(\mathbf{z}_s = \mathbf{x} \mid \mathbf{z}_t = \mathbf{x}, \mathbf{x})q(\mathbf{z}_t = \mathbf{x} \mid \mathbf{x}) + q(\mathbf{z}_s = \mathbf{x} \mid \mathbf{z}_t = \boldsymbol{m}, \mathbf{x})q(\mathbf{z}_t = \boldsymbol{m} \mid \mathbf{x})$$

$$= (1 - \sigma_t)\alpha_t + \frac{\alpha_s - (1 - \sigma_t)\alpha_t}{1 - \alpha_t}(1 - \alpha_t) = \alpha_s, \tag{12}$$

and

$$q(\mathbf{z}_s = \boldsymbol{m} \mid \mathbf{x}) = q(\mathbf{z}_s = \boldsymbol{m} \mid \mathbf{z}_t = \mathbf{x}, \mathbf{x})q(\mathbf{z}_t = \mathbf{x} \mid \mathbf{x}) + q(\mathbf{z}_s = \boldsymbol{m} \mid \mathbf{z}_t = \boldsymbol{m}, \mathbf{x})q(\mathbf{z}_t = \boldsymbol{m} \mid \mathbf{x})$$

$$= \sigma_t \alpha_t + \frac{1 - \alpha_s - \sigma_t \alpha_t}{1 - \alpha_t}(1 - \alpha_t) = 1 - \alpha_s. \tag{13}$$

Combining (12) and (13) yields $q(\mathbf{z}_s \mid \mathbf{x}) = \text{Cat}(\mathbf{z}_s; \alpha_s \mathbf{x} + (1 - \alpha_s)\boldsymbol{m})$.    □

### A.2  Deriving the non-Markovian forward step $q_\sigma(\mathbf{z}_t \mid \mathbf{z}_s, \mathbf{x})$

By Bayes' rule we have:

$$q_\sigma(\mathbf{z}_t \mid \mathbf{z}_s, \mathbf{x}) = \frac{q_\sigma(\mathbf{z}_s \mid \mathbf{z}_t, \mathbf{x})q_\sigma(\mathbf{z}_t \mid \mathbf{x})}{q_\sigma(\mathbf{z}_s \mid \mathbf{x})}.$$

From Theorem 3.1, we can replace the marginals $q_\sigma(\mathbf{z}_t \mid \mathbf{x})$ and $q_\sigma(\mathbf{z}_s \mid \mathbf{x})$ with those from (1).

To derive the form for $q_\sigma(\mathbf{z}_t \mid \mathbf{z}_s, \mathbf{x})$, we now look at the two possible values of $\mathbf{z}_s \neq \boldsymbol{m}$ and $\mathbf{z}_s = \boldsymbol{m}$.

**Case 1a: $\mathbf{z}_s \neq \boldsymbol{m}, \mathbf{z}_t \neq \boldsymbol{m}$**

$$q_\sigma(\mathbf{z}_t = \mathbf{x} \mid \mathbf{z}_s = \mathbf{x}, \mathbf{x}) = \frac{q_\sigma(\mathbf{z}_s = \mathbf{x} \mid \mathbf{z}_t = \mathbf{x}, \mathbf{x})q_\sigma(\mathbf{z}_t = \mathbf{x} \mid \mathbf{x})}{q_\sigma(\mathbf{z}_s = \mathbf{x} \mid \mathbf{x})} = \frac{(1 - \sigma_t)\alpha_t}{\alpha_s}. \tag{14}$$

**Case 1b: $\mathbf{z}_s \neq \boldsymbol{m}, \mathbf{z}_t = \boldsymbol{m}$**

$$q_\sigma(\mathbf{z}_t = \boldsymbol{m} \mid \mathbf{z}_s = \mathbf{x}, \mathbf{x}) = \frac{q_\sigma(\mathbf{z}_s = \mathbf{x} \mid \mathbf{z}_t = \boldsymbol{m}, \mathbf{x})q_\sigma(\mathbf{z}_t = \boldsymbol{m} \mid \mathbf{x})}{q_\sigma(\mathbf{z}_s = \mathbf{x} \mid \mathbf{x})}$$

$$= \frac{\left(\frac{\alpha_s - (1 - \sigma_t)\alpha_t}{1 - \alpha_t}\right)(1 - \alpha_t)}{\alpha_s}$$

$$= \frac{\alpha_s - (1 - \sigma_t)\alpha_t}{\alpha_s}. \tag{15}$$

**Case 2a:** $\mathbf{z}_s = \boldsymbol{m}, \mathbf{z}_t \neq \boldsymbol{m}$

$$q_\sigma(\mathbf{z}_t = \mathbf{x} \mid \mathbf{z}_s = \boldsymbol{m}, \mathbf{x}) = \frac{q_\sigma(\mathbf{z}_s = \boldsymbol{m} \mid \mathbf{z}_t = \mathbf{x}, \mathbf{x})q_\sigma(\mathbf{z}_t = \mathbf{x} \mid \mathbf{x})}{q_\sigma(\mathbf{z}_s = \boldsymbol{m} \mid \mathbf{x})} = \frac{\sigma_t \alpha_t}{1 - \alpha_s}. \tag{16}$$

**Case 2b:** $\mathbf{z}_s = \boldsymbol{m}, \mathbf{z}_t = \boldsymbol{m}$

$$\begin{aligned} q_\sigma(\mathbf{z}_t = \boldsymbol{m} \mid \mathbf{z}_s = \boldsymbol{m}, \mathbf{x}) &= \frac{q_\sigma(\mathbf{z}_s = \boldsymbol{m} \mid \mathbf{z}_t = \boldsymbol{m}, \mathbf{x})q_\sigma(\mathbf{z}_t = \mathbf{x} \mid \mathbf{x})}{q_\sigma(\mathbf{z}_s = \boldsymbol{m} \mid \mathbf{x})} \\ &= \frac{\left(\frac{1 - \alpha_s - \sigma_t \alpha_t}{1 - \alpha_t}\right)(1 - \alpha_t)}{1 - \alpha_s} \\ &= \frac{1 - \alpha_s - \sigma_t \alpha_t}{1 - \alpha_s}. \end{aligned} \tag{17}$$

Combining (14), (15), (16), and (17) yields the non-Markovian forward process:

$$q_\sigma(\mathbf{z}_t \mid \mathbf{z}_s, \mathbf{x}) = \begin{cases} \mathrm{Cat}(\mathbf{z}_t; \frac{(1 - \sigma_t)\alpha_t}{\alpha_s}\mathbf{x} + \frac{\alpha_s - (1 - \sigma_t)\alpha_t}{\alpha_s}\boldsymbol{m}), & \mathbf{z}_s \neq \boldsymbol{m} \\ \mathrm{Cat}(\mathbf{z}_t; \frac{\sigma_t \alpha_t}{1 - \alpha_s}\mathbf{x} + \frac{1 - \alpha_s - \sigma_t \alpha_t}{1 - \alpha_s}\boldsymbol{m}), & \mathbf{z}_s = \boldsymbol{m} \end{cases}. \tag{18}$$

## A.3 Deriving Bounds for $\sigma_t$

To ensure that we have defined a valid probability, in both cases of the posterior in (6), since $\mathbf{x}^\top \boldsymbol{m} = 0$, we require that the coefficients on both terms be within the range $[0, 1]$. Using this, we can derive the bounds for $\sigma_t$ given in (7).

For case where $\mathbf{z}_t \neq \boldsymbol{m}$, both coefficients must satisfy:

$$0 \leq \sigma_t \leq 1. \tag{19}$$

For the case where $\mathbf{z}_t = \boldsymbol{m}$, ensuring that the coefficients are in the range $[0, 1]$ leads to the restriction that:

$$\frac{\alpha_t - \alpha_s}{\alpha_t} \leq \sigma_t \leq \frac{1 - \alpha_s}{\alpha_t}. \tag{20}$$

Since the noise schedule is monotonically decreasing in $t$, the lower bound in (20) is $\leq 0$. Hence, combining (19) and (20) produces the bounds in (7).

## A.4 Deriving the ReMDM NELBO

The variational objective or negative evidence lower bound (NELBO) for diffusion models has the following form [50]:

$$\mathcal{L} = \mathbb{E}_{\mathbf{z}_{0:T} \sim q(\mathbf{z}_{0:1}|\mathbf{x})} \big[ \log(q(\mathbf{z}_{0:1} \mid \mathbf{x}) \,/\, p_\theta(\mathbf{x})p_\theta(\mathbf{z}_{0:1} \mid \mathbf{x})) \big] \tag{21}$$

Introducing the joint probability decomposition defined in equation 5 and equation 8, we have

$$\mathcal{L} = \mathbb{E}_{\mathbf{z}_{0:1} \sim q(\mathbf{z}_{0:1}|\mathbf{x})} \Bigg[ \underbrace{-\log p_\theta(\mathbf{x}|\mathbf{z}_0)}_{\mathcal{L}_{recon}} + \underbrace{D_{\mathrm{KL}}[q(\mathbf{z}_1|\mathbf{x})\|p_\theta(\mathbf{z}_1)]}_{\mathcal{L}_{prior}} + \underbrace{\sum_{i=1}^{T} D_{\mathrm{KL}}[q(\mathbf{z}_s \mid \mathbf{z}_t, \mathbf{x})\|p_\theta(\mathbf{z}_s \mid \mathbf{z}_t)]}_{\mathcal{L}_{diffusion}} \Bigg], \tag{22}$$

where $D_{\mathrm{KL}}$ denotes the Kullback–Leibler divergence.

Starting from (21), we note that, in practice, we set $p_\theta(\mathbf{z}_1) = q(\mathbf{z}_1 \mid x) = q(\mathbf{z}_1) = \boldsymbol{\pi}$ which ensures $\mathcal{L}_{prior} = 0$. Additionally, the reconstruction loss $\mathcal{L}_{recon}$ is equivalent in both (21) and (9). We therefore turn our attention to the ReMDM diffusion loss term

$$\mathcal{L}_{diffusion}^\sigma = \sum_{j=1}^{T} D_{\mathrm{KL}}[q_\sigma(\mathbf{z}_s \mid \mathbf{z}_t, \mathbf{x})\|p_\theta(\mathbf{z}_s \mid \mathbf{z}_t)] \tag{23}$$

Additionally, we note that in Sahoo et al. [41], MDLM is parameterized using the denoising $\mathbf{x}_\theta$ that is restricted in two ways. First the denoising model places zero probability mass on the [MASK] token, i.e., $\mathbf{x}_\theta^\top \boldsymbol{m} = 0$. Second the model 'carries-over' unmasked tokens, i.e., $\mathbf{x}_\theta^\top \mathbf{z}_t = 1$, if $\mathbf{z}_t \neq \boldsymbol{m}$. Below we assume that our denoising network follows a similar parameterization.

We break down each term in this summation by the two possible values $\mathbf{z}_t \neq \boldsymbol{m}$ and $\mathbf{z}_t = \boldsymbol{m}$.

**Case 1: $\mathbf{z}_t \neq \boldsymbol{m}$**

$$D_{\mathrm{KL}}[q_\sigma(\mathbf{z}_s \mid \mathbf{z}_t = \mathbf{x}, \mathbf{x}) \| p_\theta(\mathbf{z}_s \mid \mathbf{z}_t = \mathbf{x})]$$

$$=q_\sigma(\mathbf{z}_s = \mathbf{x} \mid \mathbf{z}_t = \mathbf{x}, \mathbf{x}) \log \frac{q_\sigma(\mathbf{z}_s = \mathbf{x} \mid \mathbf{z}_t = \mathbf{x}, \mathbf{x})}{p_\theta(\mathbf{z}_s = \mathbf{x} \mid \mathbf{z}_t = \mathbf{x})} + q_\sigma(\mathbf{z}_s = \boldsymbol{m} \mid \mathbf{z}_t = \mathbf{x}, \mathbf{x}) \log \frac{q_\sigma(\mathbf{z}_s = \boldsymbol{m} \mid \mathbf{z}_t = \mathbf{x}, \mathbf{x})}{p_\theta(\mathbf{z}_s = \boldsymbol{m} \mid \mathbf{z}_t = \mathbf{x})}$$

$$=(1 - \sigma_t) \log \frac{1 - \sigma_t}{1 - \sigma_t} + \sigma_t \log \frac{\sigma_t}{\sigma_t} = 0. \tag{24}$$

**Case 2: $\mathbf{z}_t = \boldsymbol{m}$**

$$D_{\mathrm{KL}}[q_\sigma(\mathbf{z}_s \mid \mathbf{z}_t = \boldsymbol{m}, \mathbf{x}) \| p_\theta(\mathbf{z}_s \mid \mathbf{z}_t = \boldsymbol{m})]$$

$$=q_\sigma(\mathbf{z}_s = \mathbf{x} \mid \mathbf{z}_t = \boldsymbol{m}, \mathbf{x}) \log \frac{q_\sigma(\mathbf{z}_s = \mathbf{x} \mid \mathbf{z}_t = \boldsymbol{m}, \mathbf{x})}{p_\theta(\mathbf{z}_s = \mathbf{x} \mid \mathbf{z}_t = \boldsymbol{m})} + q_\sigma(\mathbf{z}_s = \boldsymbol{m} \mid \mathbf{z}_t = \boldsymbol{m}, \mathbf{x}) \log \frac{q_\sigma(\mathbf{z}_s = \boldsymbol{m} \mid \mathbf{z}_t = \boldsymbol{m}, \mathbf{x})}{p_\theta(\mathbf{z}_s = \boldsymbol{m} \mid \mathbf{z}_t = \boldsymbol{m})}$$

$$=\frac{\alpha_s - (1 - \sigma_t)\alpha_t}{1 - \alpha_t} \log \frac{\frac{\alpha_s - (1 - \sigma_t)\alpha_t}{1 - \alpha_t}}{\frac{\alpha_s - (1 - \sigma_t)\alpha_t}{1 - \alpha_t}\mathbf{x}^\top \mathbf{x}_\theta} + \frac{1 - \alpha_s - \sigma_t \alpha_t}{1 - \alpha_t} \log \frac{\frac{1 - \alpha_s - \sigma_t \alpha_t}{1 - \alpha_t}}{\frac{1 - \alpha_s - \sigma_t \alpha_t}{1 - \alpha_t}}$$

$$=\frac{(1 - \sigma_t)\alpha_t - \alpha_s}{1 - \alpha_t} \log(\mathbf{x}^\top \mathbf{x}_\theta) \tag{25}$$

The carry-over unmasked tokens property of $\mathbf{x}_\theta$ implies that $\log(\mathbf{x}^\top \mathbf{x}_\theta) = \log(1) = 0$, which lets us write the two cases from (24) and (25) as a single expression:

$$D_{\mathrm{KL}}[q_\sigma(\mathbf{z}_s \mid \mathbf{z}_t, \mathbf{x}) \| p_\theta(\mathbf{z}_s \mid \mathbf{z}_t)] = \frac{(1 - \sigma_t)\alpha_t - \alpha_s}{1 - \alpha_t} \log(\mathbf{x}^\top \mathbf{x}_\theta) \tag{26}$$

Then rewriting the summation in (23) as an expectation with $t$ sampled uniformly from $\{\frac{1}{T}, \ldots, 1\}$ and plugging in (26), we recover the diffusion loss term in (9).

### A.5 Proof of Theorem 4.1

*Proof.* Recall that the MDLM *predictor* step amounts to drawing a sample $\mathbf{z}_{s_p}$ from the posterior given in (2), which we can reformulate as:

$$q(\mathbf{z}_{s_p} \mid \mathbf{z}_t, \mathbf{x}) = \begin{cases} \mathrm{Cat}(\mathbf{z}_{s_p}; \mathbf{z}_t), & \mathbf{z}_t \neq \boldsymbol{m}, \\ \mathrm{Cat}(\mathbf{z}_{s_p}; \frac{\alpha_s - \alpha_t}{1 - \alpha_t}\mathbf{x} + \frac{1 - \alpha_s}{1 - \alpha_t}\boldsymbol{m}), & \mathbf{z}_t = \boldsymbol{m}. \end{cases}$$

To prove the theorem statement, we must show that $q(\mathbf{z}_{s_c} \mid \mathbf{z}_t, \mathbf{x})$ is equivalent to the ReMDM posterior from (6). We begin by noting that conditioned on the predictor sample $\mathbf{z}_{s_p}$, the corrector sample $\mathbf{z}_{s_c}$ is independent of $\mathbf{z}_t$. That is

$$q(\mathbf{z}_{s_c} \mid \mathbf{z}_{s_p}, \mathbf{z}_t, \mathbf{x}) = q(\mathbf{z}_{s_c} \mid \mathbf{z}_{s_p}, \mathbf{x}) \tag{27}$$

We now look at the two cases of $\mathbf{z}_t \neq \boldsymbol{m}$ and $\mathbf{z}_t = \boldsymbol{m}$:

**Case 1a: $\mathbf{z}_t \neq \boldsymbol{m}, \mathbf{z}_{s_c} = \mathbf{x}$**

$$q(\mathbf{z}_{s_c} = \mathbf{x} \mid \mathbf{z}_t = \mathbf{x}, \mathbf{x}) = \sum_{\mathbf{z}' \in \{\mathbf{x}, \boldsymbol{m}\}} q(\mathbf{z}_{s_c} = \mathbf{x} \mid \mathbf{z}_{s_p} = \mathbf{z}', \mathbf{z}_t = \mathbf{x}, \mathbf{x}) q(\mathbf{z}_{s_p} = \mathbf{z}' \mid \mathbf{z}_t = \mathbf{x}, \mathbf{x})$$

$$=q(\mathbf{z}_{s_c} = \mathbf{x} \mid \mathbf{z}_{s_p} = \mathbf{x}, \mathbf{x}) q(\mathbf{z}_{s_p} = \mathbf{x} \mid \mathbf{z}_t = \mathbf{x}, \mathbf{x}) + q(\mathbf{z}_{s_c} = \mathbf{x} \mid \mathbf{z}_{s_p} = \boldsymbol{m}, \mathbf{x}) q(\mathbf{z}_{s_p} = \boldsymbol{m} \mid \mathbf{z}_t = \mathbf{x}, \mathbf{x})$$

$$=q(\mathbf{z}_{s_c} = \mathbf{x} \mid \mathbf{z}_{s_p} = \mathbf{x}, \mathbf{x}) = (1 - \sigma_t). \tag{28}$$

**Case 1b: $\mathbf{z}_t \neq \boldsymbol{m}, \mathbf{z}_{s_c} = \boldsymbol{m}$** Using the same argument as in Case 1a, we have that

$$q(\mathbf{z}_{s_c} = \boldsymbol{m} \mid \mathbf{z}_t = \mathbf{x}, \mathbf{x}) = q(\mathbf{z}_{s_c} = \boldsymbol{m} \mid \mathbf{z}_{s_p} = \mathbf{x}, \mathbf{x}) = \sigma_t. \tag{29}$$

**Case 2a: $\mathbf{z}_t = \boldsymbol{m}, \mathbf{z}_{s_c} = \mathbf{x}$**

$$q(\mathbf{z}_{s_c} = \mathbf{x} \mid \mathbf{z}_t = \boldsymbol{m}, \mathbf{x}) = \sum_{\mathbf{z}' \in \{\mathbf{x}, \boldsymbol{m}\}} q(\mathbf{z}_{s_c} = \mathbf{x} \mid \mathbf{z}_{s_p} = \mathbf{z}', \mathbf{z}_t = \boldsymbol{m}, \mathbf{x}) q(\mathbf{z}_{s_p} = \mathbf{z}' \mid \mathbf{z}_t = \boldsymbol{m}, \mathbf{x})$$

$$= q(\mathbf{z}_{s_c} = \mathbf{x} \mid \mathbf{z}_{s_p} = \mathbf{x}, \mathbf{x}) q(\mathbf{z}_{s_p} = \mathbf{x} \mid \mathbf{z}_t = \boldsymbol{m}, \mathbf{x}) + q(\mathbf{z}_{s_c} = \mathbf{x} \mid \mathbf{z}_{s_p} = \boldsymbol{m}, \mathbf{x}) q(\mathbf{z}_{s_p} = \boldsymbol{m} \mid \mathbf{z}_t = \boldsymbol{m}, \mathbf{x})$$

$$= (1 - \sigma_t) \left( \frac{\alpha_s - \alpha_t}{1 - \alpha_t} \right) + \left( \frac{\sigma_t \alpha_s}{1 - \alpha_s} \right) \left( \frac{1 - \alpha_s}{1 - \alpha_t} \right)$$

$$= \frac{\alpha_s - (1 - \sigma_t)\alpha_t}{1 - \alpha_t}. \tag{30}$$

**Case 2b: $\mathbf{z}_t = \boldsymbol{m}, \mathbf{z}_{s_c} = \boldsymbol{m}$**

$$q(\mathbf{z}_{s_c} = \boldsymbol{m} \mid \mathbf{z}_t = \boldsymbol{m}, \mathbf{x}) = \sum_{\mathbf{z}' \in \{\mathbf{x}, \boldsymbol{m}\}} q(\mathbf{z}_{s_c} = \boldsymbol{m} \mid \mathbf{z}_{s_p} = \mathbf{z}', \mathbf{z}_t = \boldsymbol{m}, \mathbf{x}) q(\mathbf{z}_{s_p} = \mathbf{z}' \mid \mathbf{z}_t = \boldsymbol{m}, \mathbf{x})$$

$$= q(\mathbf{z}_{s_c} = \boldsymbol{m} \mid \mathbf{z}_{s_p} = \mathbf{x}, \mathbf{x}) q(\mathbf{z}_{s_p} = \mathbf{x} \mid \mathbf{z}_t = \boldsymbol{m}, \mathbf{x}) + q(\mathbf{z}_{s_c} = \boldsymbol{m} \mid \mathbf{z}_{s_p} = \boldsymbol{m}, \mathbf{x}) q(\mathbf{z}_{s_p} = \boldsymbol{m} \mid \mathbf{z}_t = \boldsymbol{m}, \mathbf{x})$$

$$= \sigma_t \left( \frac{\alpha_s - \alpha_t}{1 - \alpha_t} \right) + \left( \frac{1 - (1 + \sigma_t)\alpha_s}{1 - \alpha_s} \right) \left( \frac{1 - \alpha_s}{1 - \alpha_t} \right)$$

$$= \frac{1 - \alpha_s - \sigma_t \alpha_t}{1 - \alpha_t}. \tag{31}$$

Combining (28), (29), (30), and (31) yields the desired result. $\square$

### A.6   Proof of Proposition 4.2

*Proof.* The forward-backward corrector sampler [5] is derived using continuous time Markov chain theory. In particular, the rate matrix for the corrector sampler step $\mathbf{R}_{corrector}$ is the sum of the forward rate matrix $\mathbf{R}_t$ and the backward rate matrix $\tilde{\mathbf{R}}_t$, i.e. $\mathbf{R}_{corrector} = \mathbf{R}_t + \tilde{\mathbf{R}}_t$.

We first extract the forward rate matrix $\mathbf{R}_t$ of MDLM using the following discretization formula.

$$q(\mathbf{z}_t = \mathbf{y}' \mid \mathbf{z}_s = \mathbf{y}) = \delta_{\mathbf{y}', \mathbf{y}} + \mathbf{R}_t(\mathbf{y}, \mathbf{y}')\Delta t + o(\Delta t) \tag{32}$$

From Sahoo et al. [41], we know that the forward step of MDLM $q(\mathbf{z}_t \mid \mathbf{z}_s) = \text{Cat}(\mathbf{z}_t; \frac{\alpha_t}{\alpha_s}\mathbf{z}_s + (1 - \frac{\alpha_t}{\alpha_s})\boldsymbol{m})$. Let $\mathcal{V}$ denote the vocabulary set.

**Case 1a: $\mathbf{z}_s = \mathbf{x} \in \mathcal{V} \setminus \boldsymbol{m}, \mathbf{z}_t = \mathbf{x}$**

$$1 + \mathbf{R}_t(\mathbf{x}, \mathbf{x})\Delta t + o(\Delta t) = q(\mathbf{z}_t = \mathbf{x} \mid \mathbf{z}_s = \mathbf{x}) = \frac{\alpha_t}{\alpha_s}$$

$$\Rightarrow \mathbf{R}_t(\mathbf{x}, \mathbf{x}) = \lim_{\Delta t \to 0} \frac{1}{\Delta t} \left( \frac{\alpha_t}{\alpha_s} - 1 \right) = \frac{\alpha'_t}{\alpha_t} \tag{33}$$

**Case 1b: $\mathbf{z}_s = \mathbf{x} \in \mathcal{V} \setminus \boldsymbol{m}, \mathbf{z}_t = \boldsymbol{m}$**

$$\mathbf{R}_t(\mathbf{x}, \boldsymbol{m})\Delta t + o(\Delta t) = q(\mathbf{z}_t = \boldsymbol{m} \mid \mathbf{z}_s = \mathbf{x}) = 1 - \frac{\alpha_t}{\alpha_s}$$

$$\Rightarrow \mathbf{R}_t(\mathbf{x}, \boldsymbol{m}) = \lim_{\Delta t \to 0} \frac{1}{\Delta t} \left( 1 - \frac{\alpha_t}{\alpha_s} \right) = -\frac{\alpha'_t}{\alpha_t} \tag{34}$$

**Case 1c: $\mathbf{z}_s = \mathbf{x} \in \mathcal{V} \setminus \boldsymbol{m}, \mathbf{z}_t = \mathbf{y} \in \mathcal{V} \setminus \{\mathbf{x}, \boldsymbol{m}\}$**

$$\mathbf{R}_t(\mathbf{x}, \mathbf{y})\Delta t + o(\Delta t) = q(\mathbf{z}_t = \mathbf{y} \mid \mathbf{z}_s = \mathbf{x}) = 0 \Rightarrow \mathbf{R}_t(\mathbf{x}, \mathbf{y}) = 0 \tag{35}$$

**Case 1d: $\mathbf{z}_s = \boldsymbol{m}, \mathbf{z}_t = \boldsymbol{m}$**

$$1 + \mathbf{R}_t(\boldsymbol{m}, \boldsymbol{m})\Delta t + o(\Delta t) = q(\mathbf{z}_t = \boldsymbol{m} \mid \mathbf{z}_s = \boldsymbol{m}) = 1 \Rightarrow \mathbf{R}_t(\boldsymbol{m}, \boldsymbol{m}) = 0 \tag{36}$$

**Case 1e: $\mathbf{z}_s = \boldsymbol{m}, \mathbf{z}_t = \mathbf{x} \in \mathcal{V} \setminus \boldsymbol{m}$**

$$\mathbf{R}_t(\boldsymbol{m}, \mathbf{x})\Delta t + o(\Delta t) = q(\mathbf{z}_t = \mathbf{x} \mid \mathbf{z}_s = \boldsymbol{m}) = 0 \Rightarrow \mathbf{R}_t(\boldsymbol{m}, \mathbf{x}) = 0 \tag{37}$$

Similarly, we can derive the backward rate matrix $\tilde{\mathbf{R}}_t$ using the following formula and MDLM posterior (2).

$$q(\mathbf{z}_s = \mathbf{y}' \mid \mathbf{z}_t = \mathbf{y}, \mathbf{x}) = \delta_{\mathbf{y}',\mathbf{y}} + \mathbf{R}_t(\mathbf{y}, \mathbf{y}')\Delta t + o(\Delta t) \tag{38}$$

**Case 2a:** $\mathbf{z}_t = \mathbf{x} \in \mathcal{V} \setminus m, \mathbf{z}_s = \mathbf{x}$

$$1 + \tilde{\mathbf{R}}_t(\mathbf{x}, \mathbf{x})\Delta t + o(\Delta t) = q(\mathbf{z}_s = \mathbf{x} \mid \mathbf{z}_t = \mathbf{x}) = 1 \Rightarrow \tilde{\mathbf{R}}_t(\mathbf{x}, \mathbf{x}) = 0 \tag{39}$$

**Case 2b:** $\mathbf{z}_t = \mathbf{x} \in \mathcal{V} \setminus m, \mathbf{z}_s = \mathbf{y} \in \mathcal{V} \setminus \mathbf{x}$

$$\tilde{\mathbf{R}}_t(\mathbf{x}, \mathbf{y})\Delta t + o(\Delta t) = q(\mathbf{z}_s = \mathbf{y} \mid \mathbf{z}_t = \mathbf{x}) = 0 \Rightarrow \tilde{\mathbf{R}}_t(\mathbf{x}, \mathbf{y}) = 0 \tag{40}$$

**Case 2c:** $\mathbf{z}_t = m, \mathbf{z}_s = \mathbf{x}$

$$\tilde{\mathbf{R}}_t(m, \mathbf{x})\Delta t + o(\Delta t) = q(\mathbf{z}_s = \mathbf{x} \mid \mathbf{z}_t = m) = \frac{\alpha_s - \alpha_t}{1 - \alpha_t}$$

$$\Rightarrow \tilde{\mathbf{R}}_t(m, \mathbf{x}) = \lim_{\Delta t \to 0} \frac{1}{\Delta t} \frac{\alpha_s - \alpha_t}{1 - \alpha_t} = -\frac{\alpha_t'}{1 - \alpha_t} \tag{41}$$

**Case 2d:** $\mathbf{z}_t = m, \mathbf{z}_s = m$

$$1 + \tilde{\mathbf{R}}_t(m, m)\Delta t + o(\Delta t) = q(\mathbf{z}_s = m \mid \mathbf{z}_t = m) = \frac{1 - \alpha_s}{1 - \alpha_t}$$

$$\Rightarrow \tilde{\mathbf{R}}_t(m, m) = \lim_{\Delta t \to 0} \frac{1}{\Delta t}\left(\frac{1 - \alpha_s}{1 - \alpha_t} - 1\right) = \frac{\alpha_t'}{1 - \alpha_t} \tag{42}$$

**Case 2e:** $\mathbf{z}_t = m, \mathbf{z}_s = \mathbf{y} \in \mathcal{V} \setminus \{\mathbf{x}, m\}$

$$\tilde{\mathbf{R}}_t(m, \mathbf{y})\Delta t + o(\Delta t) = q(\mathbf{z}_s = m \mid \mathbf{z}_t = \mathbf{y}) = 0 \Rightarrow \tilde{\mathbf{R}}_t(m, m) = 0 \tag{43}$$

By adding $\mathbf{R}_t$ and $\tilde{\mathbf{R}}_t$, we get the rate matrix of the forward-backward corrector step. The next step is to discretize it. Concretely, we apply (38) to the forward-backward rate matrix.

$$q_{FB}(\mathbf{z}_{s_c} \mid \mathbf{z}_{s_p}, \mathbf{x}) = \begin{cases} \mathrm{Cat}(\mathbf{z}_{s_c}; \frac{2\alpha_t - \alpha_s}{\alpha_t}\mathbf{x} + \frac{\alpha_s - \alpha_t}{\alpha_t}m), & \mathbf{z}_{s_p} \neq m \\ \mathrm{Cat}(\mathbf{z}_{s_c}; \frac{\alpha_s - \alpha_t}{1 - \alpha_t}\mathbf{x} + \frac{1 - \alpha_s}{1 - \alpha_t}m), & \mathbf{z}_{s_p} = m. \end{cases} \tag{44}$$

Note that (44) keeps the marginal at time $t$ unchanged while (11) keeps the marginal at time $s$ unchanged. In order to conduct a fair comparison, we rewrite the time $t$ version of (11) as follows.

$$q_{ReMDM}^{corrector}(\mathbf{z}_{s_c} \mid \mathbf{z}_{s_p}, \mathbf{x}) = \begin{cases} \mathrm{Cat}(\mathbf{z}_{s_c}; (1 - \sigma_t)\mathbf{x} + \sigma_t m), & \mathbf{z}_{s_p} \neq m \\ \mathrm{Cat}(\mathbf{z}_{s_c}; \frac{\sigma_t \alpha_t}{1 - \alpha_t}\mathbf{x} + \frac{1 - (1 + \sigma_t)\alpha_t}{1 - \alpha_t}m), & \mathbf{z}_{s_p} = m. \end{cases} \tag{45}$$

Comparing (44) and (45), we can find that (44) is a special case where $\sigma_t = \frac{\alpha_s - \alpha_t}{\alpha_t}$. $\qquad\square$

## A.7 Proof of Proposition 4.3

*Proof.* The DFM corrector sampler [13] is derived from a generating velocity $u_t^{corr}$ using the following transformation equation.

$$\mathbf{z}_{s_c} \sim \delta_{\mathbf{z}_t}(\cdot) + u_t^{corr}(\cdot, \mathbf{z}_t)\Delta t \tag{46}$$

The corrector generating velocity is defined as a weighted sum of the forward sampling generating velocity $\hat{u}_t$ and the backward sampling generating velocity $\breve{u}_t$.

$$u_t^{corr}(\cdot, \mathbf{z}_t) = (1 + \beta_t)\hat{u}_t(\cdot, \mathbf{z}_t) - \beta_t \breve{u}_t(\cdot, \mathbf{z}_t) \tag{47}$$

The weighting coefficient $\beta_t \in \mathbb{R}$ is referred to as the corrector schedule and can be user-specified.

The forward and backward sampling generating velocities take the following forms:

$$\hat{u}_t(\cdot, \mathbf{z}_t) = -\frac{\alpha'_t}{1 - \alpha_t}\big[p_{0|t}(\cdot \mid \mathbf{z}_t) - \delta_{\mathbf{z}_t}(\cdot)\big] \tag{48}$$

$$\check{u}_t(\cdot, \mathbf{z}_t) = -\frac{\alpha'_t}{\alpha_t}\big[\delta_{\mathbf{z}_t}(\cdot) - p_{1|t}(\cdot \mid \mathbf{z}_t)\big] \tag{49}$$

Note that our formulation is slightly different from the original presentation in Gat et al. [13], since in our notation as $t$ goes from 1 to 0, we move from noise to the target distribution, whereas in the flow matching literature this direction is reversed. Plugging (48) and (49) into (47), we derive the following form for $u_t^{corr}$:

$$u_t^{corr}(\cdot, \mathbf{z}_t) = \Big[\frac{(1+\beta_t)\alpha'_t}{1-\alpha_t} + \frac{\beta_t \alpha'_t}{\alpha_t}\Big]\delta_{\mathbf{z}_t}(\cdot) - \frac{(1+\beta_t)\alpha'_t}{1-\alpha_t}p_{0|t}(\cdot \mid \mathbf{z}_t) - \frac{\beta_t \alpha'_t}{\alpha_t}p_{1|t}(\cdot \mid \mathbf{z}_t) \tag{50}$$

By plugging (50) into (46), we have that

$$\mathbf{z}_{s_c} \sim \Big[1 + \frac{(1+\beta_t)\alpha'_t \Delta t}{1-\alpha_t} + \frac{\beta_t \alpha'_t \Delta t}{\alpha_t}\Big]\delta_{\mathbf{z}_t}(\cdot) - \frac{(1+\beta_t)\alpha'_t \Delta t}{1-\alpha_t}p_{0|t}(\cdot \mid \mathbf{z}_t) - \frac{\beta_t \alpha'_t \Delta t}{\alpha_t}p_{1|t}(\cdot \mid \mathbf{z}_t)$$

$$\sim \Big[1 + \frac{(1+\beta_t)(\alpha_t - \alpha_s)}{1-\alpha_t} + \frac{\beta_t(\alpha_t - \alpha_s)}{\alpha_t}\Big]\delta_{\mathbf{z}_t}(\cdot) - \frac{(1+\beta_t)(\alpha_t - \alpha_s)}{1-\alpha_t}p_{0|t}(\cdot \mid \mathbf{z}_t) - \frac{\beta_t(\alpha_t - \alpha_s)}{\alpha_t}p_{1|t}(\cdot \mid \mathbf{z}_t) \tag{51}$$

We can rewrite this as

$$q_{DFM}(\mathbf{z}_{s_c} \mid \mathbf{z}_t, \mathbf{x}) = \begin{cases} \mathrm{Cat}(\mathbf{z}_{s_c}; (1 + \frac{\beta_t(\alpha_t - \alpha_s)}{\alpha_t})\mathbf{x} + \frac{\beta_t(\alpha_s - \alpha_t)}{\alpha_t}\boldsymbol{m}), & \mathbf{z}_t \neq \boldsymbol{m} \\ \mathrm{Cat}(\mathbf{z}_{s_c}; \frac{(1+\beta_t)(\alpha_s - \alpha_t)}{1-\alpha_t}\mathbf{x} + (1 + \frac{(1+\beta_t)(\alpha_t - \alpha_s)}{1-\alpha_t})\boldsymbol{m}), & \mathbf{z}_t = \boldsymbol{m}. \end{cases} \tag{52}$$

Comparing (52) and (6), we see that (52) is equivalent to (6) where $\sigma_t = \frac{\beta_t(\alpha_s - \alpha_t)}{\alpha_t}$.

$\square$

## A.8  Proof of Proposition 4.4

*Proof.* Recall that the ReMDM posterior is defined as:

$$q_\sigma(\mathbf{z}_s \mid \mathbf{z}_t, \mathbf{x}) = \begin{cases} \mathrm{Cat}(\mathbf{z}_s; (1 - \sigma_t)\mathbf{x} + \sigma_t \boldsymbol{m}), & \mathbf{z}_t \neq \boldsymbol{m} \\ \mathrm{Cat}\left(\mathbf{z}_s; \frac{\alpha_s - (1-\sigma_t)\alpha_t}{1-\alpha_t}\mathbf{x} + \frac{1-\alpha_s - \sigma_t \alpha_t}{1-\alpha_t}\boldsymbol{m}\right), & \mathbf{z}_t = \boldsymbol{m}. \end{cases} \tag{53}$$

When $\alpha_s = \alpha_t = \alpha$, the posterior simplifies to:

$$q_\sigma(\mathbf{z}_s \mid \mathbf{z}_t, \mathbf{x}) = \begin{cases} \mathrm{Cat}(\mathbf{z}_s; (1 - \sigma_t)\mathbf{x} + \sigma_t \boldsymbol{m}), & \mathbf{z}_t \neq \boldsymbol{m} \\ \mathrm{Cat}\left(\mathbf{z}_s; \frac{\sigma_t \alpha}{1-\alpha}\mathbf{x} + \frac{1-(1+\sigma_t)\alpha}{1-\alpha}\boldsymbol{m}\right), & \mathbf{z}_t = \boldsymbol{m}. \end{cases} \tag{54}$$

For $0 \leq \sigma_t \leq \min\left\{1, \frac{1-\alpha}{\alpha}\right\}$, (54) defines a valid probability distribution.

In contrast, the DFM sampler applies the following transformation:

$$\mathbf{z}_{s_c} \sim \delta_{\mathbf{z}_t}(\cdot) + u_t^{corr}(\cdot, \mathbf{z}_t)\Delta t \tag{55}$$

where the corrector velocity $u_t^{corr}$ is a weighted combination of the forward velocity $\hat{u}_t$ and backward velocity $\check{u}_t$:

$$u_t^{\text{corr}}(\cdot, \mathbf{z}_t) = (1 + \beta_t)\hat{u}_t(\cdot, \mathbf{z}_t) - \beta_t\check{u}_t(\cdot, \mathbf{z}_t) \tag{56}$$

with

$$\hat{u}_t(\cdot, \mathbf{z}_t) = -\frac{\alpha_t'}{1 - \alpha_t}\left[p_{0|t}(\cdot \mid \mathbf{z}_t) - \delta_{\mathbf{z}_t}(\cdot)\right], \tag{57}$$

$$\check{u}_t(\cdot, \mathbf{z}_t) = -\frac{\alpha_t'}{\alpha_t}\left[\delta_{\mathbf{z}_t}(\cdot) - p_{1|t}(\cdot \mid \mathbf{z}_t)\right]. \tag{58}$$

If there exists a time interval $[t, t - \Delta t]$ with $\Delta t > 0$ during which $\alpha_t$ remains constant, then $\alpha_t' = 0$ throughout this interval. Consequently, both $\hat{u}_t$ and $\check{u}_t$ vanish, leading to $u_t^{\text{corr}} = 0$ and hence $\mathbf{z}_{s_c} \sim \delta_{\mathbf{z}_t}(\cdot)$. In other words, the DFM corrector becomes a degenerate sampler that merely copies the existing token without making any changes.

In conclusion, Proposition 4.3 implies that the DFM corrector is a special case of the more general ReMDM sampler. The above analysis further establishes that this inclusion is strict. Thus, ReMDM strictly generalizes the DFM corrector.

$\square$

## B  Comparison to DDIM Non-Markovian Processes

In DDIM [51], the authors present non-Markovian forward processes for both continuous and discrete signals. For discrete data, although in DDIM the proposed method assumes a uniform categorical as the limiting distribution, here we demonstrate how one can adapt the proposed method in DDIM for absorbing state diffusion and derive an equivalence between ReMDM and this new DDIM process under a reparameterization of $\sigma_t$. For clarity, we use $\sigma_t^{DDIM}$ to denote the parameter from DDIM and $\sigma_t^{ReMDM}$ to denote the parameter used in our work.

In DDIM, the processes assuming a uniform categorical distribution as the limiting distribution is defined to have the following posteriors:

$$q(\mathbf{z}_s \mid \mathbf{z}_t, \mathbf{x}) = (\alpha_s - \sigma_t\alpha_t)\mathbf{x} + \sigma_t^{DDIM}\mathbf{z}_t + (1 - \alpha_s - (1 - \alpha_t)\sigma_t^{DDIM})|V|^{-1}\mathbf{1}, \tag{59}$$

where $\mathbf{1}$ is a column vector of ones and $|V|$ is the vocabulary size. We can apply the methodology from DDIM to absorbing state diffusion by replacing the limiting distribution $\mathbf{1}/|V|$ in (59) with $\boldsymbol{m}$, which produces:

$$\begin{aligned}
q(\mathbf{z}_s \mid \mathbf{z}_t, \mathbf{x}) &= (\alpha_s - \sigma_t\alpha_t)\mathbf{x} + \sigma_t^{DDIM}\mathbf{z}_t + (1 - \alpha_s - (1 - \alpha_t)\sigma_t)\boldsymbol{m} \\
&= \begin{cases} (\alpha_s + \sigma_t^{DDIM}(1 - \alpha_t))\mathbf{x} + ((1 - \alpha_s) - (1 - \alpha_t)\sigma_t^{DDIM})\boldsymbol{m} & \mathbf{z}_t \neq \boldsymbol{m} \\ (\alpha_s - \sigma_t^{DDIM}\alpha_t)\mathbf{x} + (1 - \alpha_s + \alpha_t\sigma_t^{DDIM})\boldsymbol{m} & \mathbf{z}_t = \boldsymbol{m} \end{cases}
\end{aligned} \tag{60}$$

Comparing the $\mathbf{z}_t \neq \boldsymbol{m}$ case in (60) to that in the ReMDM posterior in (6), we can derive an equivalence between our proposed method and that from DDIM with the following reparameterization:

$$\sigma_t^{ReMDM} = 1 - \alpha_s - (1 - \alpha_t)\sigma_t^{DDIM}. \tag{61}$$

Plugging this reparameterization into the $\mathbf{z}_t = \boldsymbol{m}$ case in (6) yields an equivalence to the $\mathbf{z}_t = \boldsymbol{m}$ case in (60) as well.

In addition to extending the discrete formulation from DDIM to absorbing state processes, we believe that our formulation is easier to analyze relative to the one when reparameterizing to use $\sigma_t^{DDIM}$, e.g., deriving bounds on $\sigma_t$ is more straightforward for our work and it is more natural to explore the various design decisions defined in Section 4 using our formulation.

## C  ReMDM Sampler Algorithms

Below we present algorithms for ReMDM-switch (Algorithm 2) and ReMDM-loop (Algorithm 3) both of which can optionally combine with the various strategies for setting $\sigma_t$ described in Section 4.1.

---

**Algorithm 2** Sampling with ReMDM-switch.

---

**Input:** pre-trained denoising network $\mathbf{x}_\theta$ (e.g., MDLM), number of timesteps $T$, number of tokens in a sequence $L$, noising schedule $\alpha_t$, maximum value for remasking $\eta_{cap} \in [0, 1]$, rescale value for remasking $\eta_{rescale} \in [0, 1]$, boolean value for whether to use confidence strategy $\texttt{use\_conf}$, time for 'switching on' ReMDM $t_{switch} \in (0, 1)$.

Initialize $\mathbf{z}_t^{(1:L)} = \{\boldsymbol{m}\}^L$.

Initialize $\psi_t^{(1:L)} = \{-\infty\}^L$.

**for** $i = T$ **to** 1 **do**
$\quad t = i/T, s = (i-1)/T$.
$\quad$ Set $\alpha_t, \alpha_s$ according to noise schedule.
$\quad$ **if** $t \leq t_{switch}$ **then**
$\quad\quad \sigma_t^{(\ell)} = \eta_{rescale} \cdot \min\{\eta_{cap}, (1 - \alpha_s)/\alpha_t\}$ for all $\ell = \{1, \ldots, L\}$.
$\quad\quad$ **if** $\texttt{use\_conf}$ **then**
$\quad\quad\quad$ Compute $\eta_{conf}^{(\ell)} = \frac{\exp(-\psi^{(\ell)})}{\sum_{l'} \exp(-\psi^{(\ell')})}$ for all $\ell \in \{1, \ldots, L\}$.
$\quad\quad\quad \sigma_t^{(\ell)} = \eta_{conf}^{(\ell)} \cdot \sigma_t^{(\ell)}$ for all $\ell \in \{1, \ldots, L\}$.
$\quad\quad$ **end if**
$\quad$ **else**
$\quad\quad \sigma_t^{(1:L)} = \{0\}^L$.
$\quad$ **end if**
$\quad$ For all $\ell \in \{1, \ldots, L\}$, compute approximate posterior:

$$
\begin{aligned}
p_\theta(\mathbf{z}_s^{(\ell)} \mid \mathbf{z}_t^{(1:L)}) &= q_\sigma(\mathbf{z}_s^{(\ell)} \mid \mathbf{z}_t^{(1:L)}, \mathbf{x} = \mathbf{x}_\theta^{(\ell)}(\mathbf{z}_t^{(1:L)})) \\
&= \begin{cases} \mathrm{Cat}(\mathbf{z}_s; (1 - \sigma_t^{(\ell)})\mathbf{x}_\theta^{(\ell)} + \sigma_t^{(\ell)}\boldsymbol{m}), & \mathbf{z}_t \neq \boldsymbol{m} \\ \mathrm{Cat}(\mathbf{z}_s; \frac{\alpha_s - (1-\sigma_t^{(\ell)})\alpha_t}{1 - \alpha_t}\mathbf{x}_\theta^{(\ell)} + \frac{1 - \alpha_s - \sigma_t^{(\ell)}\alpha_t}{1 - \alpha_t}\boldsymbol{m}), & \mathbf{z}_t = \boldsymbol{m} \end{cases}
\end{aligned}
$$

$\quad$ Sample $\mathbf{z}_s^{(\ell)} \sim p_\theta$ for all $\ell \in \{1, \ldots, L\}$.
$\quad$ **if** $\texttt{use\_conf}$ **then**
$\quad\quad$ Store confidence scores $\psi_t^{(\ell)} = (\mathbf{z}_s^{(\ell)})^\top \mathbf{x}_\theta(\mathbf{z}_t^{(1:L)})^{(\ell)}$, for all newly decoded $\mathbf{z}_s^{(\ell)} \neq \mathbf{z}_t^{(\ell)} \neq \boldsymbol{m}$.
$\quad$ **end if**
$\quad$ Set $\mathbf{z}_t^{(1:L)} = \mathbf{z}_s^{(1:L)}$.
**end for**
**output** $\mathbf{z}_t^{(1:L)}$.

---

# D  Additional Experimental Details

## D.1  OpenWebText

In this experiment, we reuse the pretrained AR, SEDD, and MDLM checkpoints released by [41] where the diffusion models are trained using a log-linear schedule, i.e., $\alpha_t = 1 - t$. AR, SEDD, and MDLM share the same architecture: a Transformer-based model [56] that augments the diffusion transformer [35] with rotary embeddings [55] and consists of 169M parameters. The neural network is comprised of 12 layers, 12 attention heads, and 768 hidden dimensions. Please see Sahoo et al. [41] for the full model architecture and training details.

As in Sahoo et al. [41], we use $\texttt{gpt-2}$ tokenizer for our experiments. We use the same train-validation split as in Sahoo et al. [41] (where the last 100k documents of OWT were designated as the validation set) and randomly select 5,000 samples from the validation set to serve as the 'reference' for MAUVE score computation. For the discrete flow matching (DFM) corrector sampler, we follow Gat et al. [13] and set the corrector schedule $\beta(t) = At^{0.25}(1 - t)^{0.25}$ where $A = 10$ (see Appendix A.7 for the definition of corrector schedule). We empirically find that nucleus sampling performs better than the temperature technique proposed in Gat et al. [13]. We also use non-fixed-width time steps as in Gat et al. [13].

For evaluation metrics, we report MAUVE scores, generative perplexity, and entropy. For MAUVE, we generate 5,000 samples for each model/sampler. We use the $\texttt{gpt-2}$ tokenizer, GPT-2 Large [38] as

---

**Algorithm 3** Sampling with ReMDM-loop.

---

**Input:** pre-trained denoising network $\mathbf{x}_\theta$ (e.g., MDLM), number of timesteps $T$, number of tokens in a sequence $L$, noising schedule $\alpha_t$, maximum value for remasking $\eta_{cap} \in [0, 1]$, rescale value for remasking $\eta_{rescale} \in [0, 1]$, boolean value for whether to use confidence strategy use_conf, start time for ReMDM loop $t_{on} \in [0, 1)$, number of discrete time steps to spend prior to ReMDM loop $n_{phase_1} \in (0, T)$, number of discrete time steps to spend in ReMDM loop $n_{phase_2} \in (0, T - n_{phase_1})$.

Initialize $\mathbf{z}_t^{(1:L)} = \{\boldsymbol{m}\}^L$.

Initialize $\psi_t^{(1:t)} = \{-\infty\}^L$.

**for** $i = T$ **to** $1$ **do**

   $t = i/T, s = (i - 1)/T$.

   **if** $t > (T - n_{phase_1})/T$ **then**

      // Phase 1

      Rescale and shift time: $t = (t \cdot (1 - t_{on}) \cdot T/n_{phase_1}) + (T \cdot (t_{on} - 1)/n_{phase_1}) + 1$.

      Rescale and shift time: $s = (s \cdot (1 - t_{on}) \cdot T/n_{phase_1}) + (T \cdot (t_{on} - 1)/n_{phase_1}) + 1$.

      Set $\alpha_t, \alpha_s$ according to noise schedule.

      $\sigma_t^{(1:L)} = \{0\}^L$.

   **else if** $t \geq (T - n_{phase_1} - n_{phase_2})/T$ **then**

      // Phase 2: ReMDM loop

      Set $\alpha_t = \alpha(t_{on}), \alpha_s = \alpha(t_{on})$.

      $\sigma_t^{(\ell)} = \eta_{rescale} \cdot \min\{\eta_{cap}, (1 - \alpha_s)/\alpha_t\}$ for all $\ell = \{1, \ldots, L\}$.

      **if** use_conf **then**

         Compute $\eta_{conf}^{(\ell)} = \frac{\exp(-\psi^{(\ell)})}{\sum_{l'} \exp(-\psi^{(\ell')})}$ for all $\ell \in \{1, \ldots, L\}$.

         $\sigma_t^{(\ell)} = \eta_{conf}^{(\ell)} \cdot \sigma_t^{(\ell)}$ for all $\ell \in \{1, \ldots, L\}$.

      **end if**

   **else**

      // Phase 3

      Rescale time: $t = t \cdot t_{on} \cdot T/(T - n_{phase_1} - n_{phase_2})$.

      Rescale time: $s = s \cdot t_{on} \cdot T/(T - n_{phase_1} - n_{phase_2})$.

      Set $\alpha_t, \alpha_s$ according to noise schedule.

      $\sigma_t^{(1:L)} = \{0\}^L$.

   **end if**

   For all $\ell \in \{1, \ldots, L\}$, compute approximate posterior:

$$
p_\theta(\mathbf{z}_s^{(\ell)} \mid \mathbf{z}_t^{(1:L)}) = q_\sigma(\mathbf{z}_s^{(\ell)} \mid \mathbf{z}_t^{(1:L)}, \mathbf{x} = \mathbf{x}_\theta^{(\ell)}(\mathbf{z}_t^{(1:L)}))
$$
$$
= \begin{cases} \mathrm{Cat}(\mathbf{z}_s; (1 - \sigma_t^{(\ell)})\mathbf{x}_\theta^{(\ell)} + \sigma_t^{(\ell)}\boldsymbol{m}), & \mathbf{z}_t \neq \boldsymbol{m} \\ \mathrm{Cat}(\mathbf{z}_s; \frac{\alpha_s - (1 - \sigma_t^{(\ell)})\alpha_t}{1 - \alpha_t}\mathbf{x}_\theta^{(\ell)} + \frac{1 - \alpha_s - \sigma_t^{(\ell)}\alpha_t}{1 - \alpha_t}\boldsymbol{m}), & \mathbf{z}_t = \boldsymbol{m} \end{cases}
$$

   Sample $\mathbf{z}_s^{(\ell)} \sim p_\theta$ for all $\ell \in \{1, \ldots, L\}$.

   **if** use_conf **then**

      Store confidence scores $\psi_t^{(\ell)} = (\mathbf{z}_s^{(\ell)})^\top \mathbf{x}_\theta(\mathbf{z}_t^{(1:L)})^{(\ell)}$, for all newly decoded $\mathbf{z}_s^{(\ell)} \neq \mathbf{z}_t^{(\ell)} \neq \boldsymbol{m}$.

   **end if**

   Set $\mathbf{z}_t^{(1:L)} = \mathbf{z}_s^{(1:L)}$.

**end for**

**output** $\mathbf{z}_t^{(1:L)}$.

---

the embedding model, and the MAUVE scaling hyperparameter is set to 5. For generative perplexity, we use GPT-2 Large as the external model. As in [68], we also report the average sequence entropy as a diversity metric. Specifically, we compute the entropy for the number of tokens for each sequence before it is decoded by the tokenizer and then report the mean entropy value of 5,000 generated sequences.

## D.2 ImageNet

In this experiment, we reuse the pre-trained MaskGiT model for all samplers [7]. The MaskGiT architecture is a Transformer model [56] that consists of 24 layers, 8 attention heads, 768 embedding dimensions, and 3,072 hidden dimensions. Similar to MDLM, this model implements the carry-over unmasked tokens property in the parameterization of $x_\theta$. The full model architecture and training setup details are available in Chang et al. [7].

In Table 2, the MaskGiT sampler refers to the heuristic confidence-based decoding described in Chang et al. [7]. The MDLM sampler refers to using the outputs of the pre-trained MaskGiT denoising model, then applying the zero-mask parameterization and plugging $x_\theta$ into the posterior from (2). The ReMDM sampler refers to using the same $x_\theta$ as that used with the MDLM sampler, but with the posterior from (6). For ReMDM, we explore the max-capped (with $\eta_{cap} \in \{0.01, 0.02, 0.05\}$), rescaled (with $\eta \in \{0.01, 0.02, 0.05\}$), and confidence-based schedules described in Section 4.1. We do not combine these strategies, but rather test each separately. For all three samplers, we perform a sweep over softmax temperature scaling using a scale parameter $\tau \in \{0.6, 0.8, 1.0\}$. For a vector $\mathbf{y}$, with $\mathbf{y}_i$ denoting its $i^{\text{th}}$ component, softmax temperature scaling is implemented as follows:

$$\frac{\exp(\mathbf{y}_i/\tau)}{\sum_j \exp(\mathbf{y}_j/\tau)}.$$

For all samplers we use a log-linear schedule for $\alpha_t$, i.e., $\alpha(t) = 1 - t$.

We use the code from Dhariwal & Nichol [9] to compute FID and IS metrics.

## D.3 QM9

For the guidance experiments on QM9, we follow the setup used in Schiff et al. [46]. The dataset was tokenized using a regular expression-based tokenizer [47] with padded max sequence lengths of $L = 32$. Including special tokens, the tokenizer vocabulary size is $|V| = 40$. The AR, MDLM, and UDLM models used for discrete CBG and CFG were taken from the implementation provide in Schiff et al. [46]. These models are based on a Transformer architecture (causally masked for AR) known as DiT [35]. See Schiff et al. [46] for the full model and experimental setup details. Note that Schiff et al. [46] provide a first-order approximation for D-CBG that avoids the required $\mathcal{O}(|V| \cdot L \cdot T)$ calls to the classifier $p_\phi$, which comes from needing to evaluate every possible token replacement from a vocabulary of size $|V|$ at every position in a sequence of length $L$ for each step of the sampling process of duration $T$. However, given the relatively shorter sequences and smaller vocabulary size used in the QM9 experiments and that Schiff et al. [46] found generally higher quality results *without* the first order approximation, we omit the first-order approximation from our experiments and instead use the full, un-approximated D-CBG implementation.

In the main text, we report results for experiments performed to maximize the ring count value of molecules. In Appendix E.4, we also present results for maximizing the property of drug-likeness (QED), which was also explored in Schiff et al. [46]. Guidance training and inference were performed on the molecular property of interest (ring count or QED) using a binarized label, where molecules with property value $\geq$ the $90^{\text{th}}$ percentile in the dataset were labeled $y = 1$. The QED and ring count properties were extracted for each molecule in the QM9 dataset using the RDKit library [23].

For all the models and guidance mechanisms, we report results with varying guidance strength $\gamma \in \{1, 2, 3, 4, 5\}$ and timesteps $T \in \{32, 64, 128\}$. For the diffusion models, we generate using a log-linear schedule for $\alpha_t$, i.e., $\alpha(t) = 1 - t$.

For evaluation, we generated 1,024 sequences with each model / guidance mechanism. We used RDKit to parse the generated sequences. Sequences that could not be parsed were deemed 'invalid'. Of the valid molecule strings, we remove duplicates and report the number of 'novel' sequences, which we define as valid and unique generated sequences that do not appear in the full QM9 dataset. We then use RDKit to compute QED / ring count of the novel generated sequences and report the mean value.

For ReMDM samplers, we reuse the pre-trained MDLM weights. We perform an extensive grid search for both properties, QED and ring count, and guidance mechanisms, D-CFG and D-CBG. Namely we explore the ReMDM-rescale schedule with $\eta_{rescale} \in \{0.1, 0.5, 0.9, 1.0\}$. We test sampling with and without the confidence-based schedule being used in conjunction with

ReMDM-rescale. For each combination, we also try both the switch and loop strategies. For switch, we use $t_{switch} \in \{0.1, 0.5, 0.9\}$. For loop, we sweep over the tuples $(t_{on}, t_{off}) \in \{(0.5, 0.25), (0.25, 0.125), (0.1, 0.05)\}$. We use non-fixed width steps in ReMDM-loop, so that for each $T$, the loop starts at discrete step $i = T/2$ and ends at step $i = \lfloor (1 - t_{off}) \cdot T \rfloor$, and we 'rescale' time accordingly before and after the loop phases, as described in Algorithm 3.

## D.4 LLaDA

We reuse the open-source LLaDA 8B Instruct model from Nie et al. [31] in this experiment. Since LLaDA utilizes MaskGiT-style samplers, we modify ReMDM-loop to similar samplers with confidence heuristics. For a maximum mask length equal to 32, we first use the LLaDA sampler to generate 28 tokens. Then we step into the ReMDM-loop phase, where we remask a fixed number (1 for Countdown and 4 for TruthfulQA) of tokens with the smallest decoding confidence and unmask the same number of tokens each time step. The ReMDM-loop phase lasts 32 steps, and after the loop phase, we use the LLaDA sampler to fill in the remaining gaps. For DFM, since it does not support looping (see Proposition 4.4), we remove the loop phase and let it remask one token and unmask two tokens each time step instead. Different from LLaDA, which utilizes greedy sampling, we enable random sampling to compute error bars.

We use the widely adopted evaluation harness from Gao et al. [12], and for both Countdown and TruthfulQA, we use a maximum length of 32 tokens without semi-autoregressive sampling introduced in [31], that is, the block size equals 32 due to relatively short answer lengths. For Countdown, we use 4-shot evaluation, and for TruthfulQA, we use 6-shot evaluation (see Appendix F.3 for the chat templates we use). For Countdown, we evaluate on 256 synthetically generated countdown questions. We repeat the sampling process for each question 10 times and calculate pass@1 as $\frac{c}{10}$ where $c$ is the number of correct trials. We then report the average pass@1 for the 256 questions. For TruthfulQA, we let the model answer 817 questions on factual knowledge and calculate the ROUGE scores between the model's answers and correct answers, as well as the model and plausibly wrong answers. The difference between the correct and incorrect ROUGE scores is reported. For both tasks, we repeat the experiments 20 times with 20 different random seeds (0-19) and report the mean values and 95% confidence intervals.

# E  Additional Experimental Results

## E.1  Training with MDLM v.s. ReMDM NELBO

In Table 4, we report QM9 dataset validation set perplexities for models trained with either the MDLM NELBO from (3) or the ReMDM NELBO from (9), with $\sigma_t = \min\{\eta_{cap}, (1 - \alpha_s)/\alpha_t\}$, for some $\eta_{cap} \in (0, 1]$. We follow the same model architecture and training settings defined in Schiff et al. [46] for this dataset. Overall, we find that validation perplexities for this dataset are fairly consistent across models trained with either objective.

Table 4: Training with ReMDM and MDLM NELBO objectives leads to similar validation set perplexity values (for the QM9 dataset). In the top row, the model is trained using the continuous time formulation of the MDLM objective from Sahoo et al. [41]. In the bottom two rows, models are trained with discrete-time objectives. The first column, $\sigma_t = 0$, corresponds to the MDLM objective from (3) and the columns with varying $\eta_{cap}$ correspond to training with the ReMDM objective from (9) with $\sigma_t = \min\{\eta_{cap}, (1 - \alpha_s)/\alpha_t\}$. We find that results are comparable across training settings.

|  | $\sigma_t = 0$ | $\eta_{cap} = 0.5$ | $\eta_{cap} = 1.0$ |
|---|---|---|---|
| $T = \infty$ | 2.088 | – | – |
| $T = 4096$ | 2.094 | 2.107 | 2.119 |
| $T = 8192$ | 2.092 | 2.097 | 2.101 |

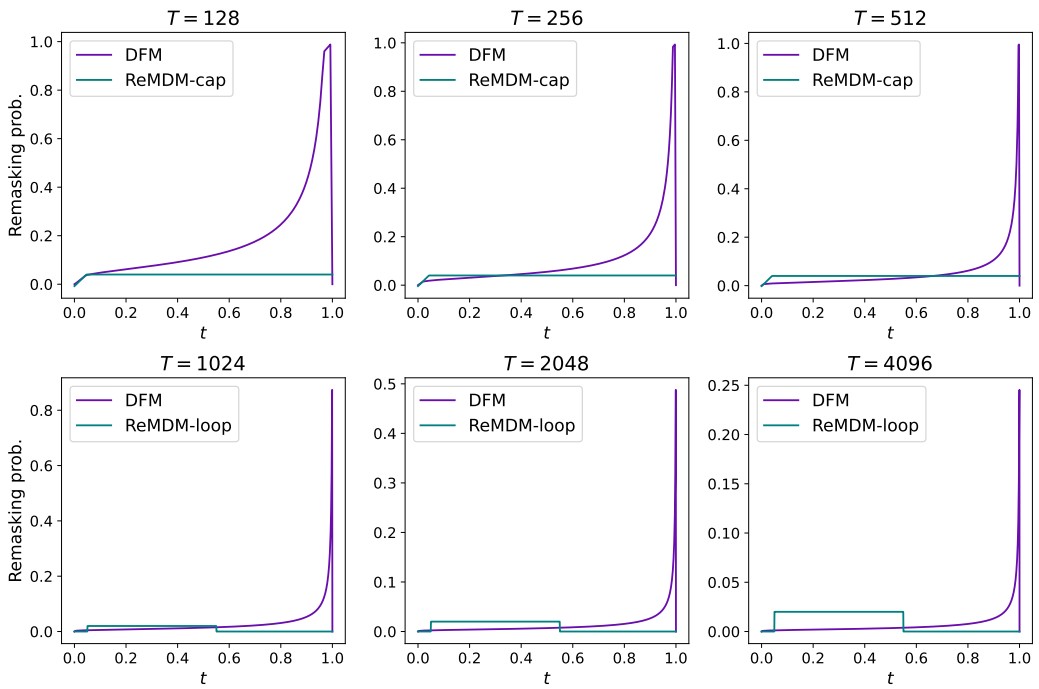

Figure 5: Remasking probability schedule of ReMDM and DFM corrector on OWT. Schedules used in experiments from Table 1 are plotted.

## E.2 OpenWebText

### E.2.1 ReMDM $\sigma_t$ v.s. DFM $\sigma_t$

In Figure 5, we plot the remasking probability schedules of ReMDM and DFM correctors in the experiments reported in Table 1. The DFM schedules show a spike at first and a long tail afterward. Here, the best performing $\beta_t = At^{0.25}(1-t)^{0.25}$ schedule reported in Gat et al. [13] is used. We also adjust the width of the time steps accordingly, as reported in Gat et al. [13].

### E.2.2 Hacking Generative Perplexity

In Table 5, we present the MAUVE, Gen PPL., and entropy values of ReMDM-cap on OWT with different ReMDM hyperparameters $\eta$. As $\eta$ increases, Gen PPL. decreases drastically at the sacrifice of entropy/diversity. In other words, Gen PPL. is hacked by tuning hyperparameters. In contrast, MAUVE cannot be hacked in this way since it measures the balance between generative perplexity and diversity by definition.

Table 5: As $\eta$ increases, we are able to get arbitrarily good Gen PPL. at the sacrifice of entropy/diversity. In contrast, MAUVE cannot be hacked in this way. ReMDM-cap on OWT with $T = 1024$ and $L = 1024$ is used and for each experiment, we generate 5000 samples. For each metric, the best value is **bolded**.

| Metric | MAUVE ($\uparrow$) | Gen PPL. ($\downarrow$) | Entropy ($\uparrow$) |
|---|---|---|---|
| $\eta_{cap} = 0.008$ | **0.36** | 27.7 | **5.32** |
| $\eta_{cap} = 0.05$ | 0.25 | 12.9 | 4.93 |
| $\eta_{cap} = 0.2$ | 0.02 | **3.1** | 2.43 |

### E.2.3 Comparing Different ReMDM Strtegies

In Table 6, we report the OpenWebText unconditional generation results for various ReMDM schedules. In the case of inference-time scaling ($T \geq L$), ReMDM-loop performs the best while in the case of faster sampling ($T < L$), ReMDM-cap is found to perform better than others. In both scenarios, ReMDM-rescale does not perform as well as the other two strategies.

Table 6: Comparing sample quality on OWT for various ReMDM strategies. $^{\dagger}$ indicates nucleus sampling. For each $T$, the best MAUVE score is **bolded**.

| Method | MAUVE (↑) | | | Gen PPL. (↓) | | | Entropy (↑) | | |
|---|---|---|---|---|---|---|---|---|---|
| | $T=1024$ | $T=2048$ | $T=4096$ | $T=1024$ | $T=2048$ | $T=4096$ | $T=1024$ | $T=2048$ | $T=4096$ |
| ReMDM-cap$^{\dagger}$ | 0.355 | 0.474 | 0.353 | 27.7 | 20.2 | 14.4 | 5.32 | 5.21 | 5.06 |
| ReMDM-loop$^{\dagger}$ | **0.403** | **0.610** | **0.656** | 28.6 | 22.8 | 17.6 | 5.38 | 5.30 | 5.20 |
| ReMDM-rescale$^{\dagger}$ | 0.253 | 0.289 | 0.208 | 27.4 | 20.4 | 14.6 | 5.28 | 5.15 | 4.94 |
| | $T=128$ | $T=256$ | $T=512$ | $T=128$ | $T=256$ | $T=512$ | $T=128$ | $T=256$ | $T=512$ |
| ReMDM-cap$^{\dagger}$ | **0.057** | **0.216** | **0.350** | 42.5 | 30.5 | 21.1 | 5.43 | 5.34 | 5.21 |
| ReMDM-loop$^{\dagger}$ | 0.041 | 0.159 | 0.328 | 51.5 | 38.1 | 29.3 | 5.52 | 5.45 | 5.38 |
| ReMDM-rescale$^{\dagger}$ | 0.040 | 0.107 | 0.218 | 46.2 | 34.8 | 25.6 | 5.44 | 5.36 | 5.24 |

### E.2.4 Tuning $\eta_{cap}$ / $\eta_{rescale}$

In Figure 6, we plot MAUVE scores when using the ReMDM-cap schedule with different $\eta_{cap}$ values and varying number of decoding steps $T$. As shown in Figure 6a, in the case of inference-time scaling ($T \geq L$), the MAUVE scores at $T = 4096$ tend to decrease as $\eta_{cap}$ increases while the MAUVE scores at $T = 1024$ tend to increase with larger $\eta_{cap}$. We attribute this to the perplexity-entropy trade-off. In particular, when $\eta_{cap}$ increases, the probability of remasking increases, improving perplexity, but also harming diversity. In the case of $T = 4096$, the models can generate higher quality sentences and the reduction in diversity outweighs marginal improvements in generative perplexity. For $T = 1024$, the samples are of relatively poorer quality and the improvement in perplexity plays a major role in driving up the MAUVE score. To achieve the best balance, we choose $\eta_{cap} = 0.008$ for the setting of inference-time scaling.

For faster sampling ($T < L$), in Figure 6b, with the exception of $\eta = 0.045$ we see a positive trend between increasing $\eta_{cap}$ and improving MAUVE scores. In Tables 1 and 6, we report results using $\eta = 0.040$.

We also performed similar studies for schedules that combine ReMDM-cap with the ReMDM-loop strategy (Figure 7) and for ReMDM-rescale schedules (Figure 8). For both of these settings, we observe trends for increasing $\eta_{cap}$ / $\eta_{rescale}$ that are similar to those described above. In Table 6, we report the following choice of hyperparameter for each of these settings. For ReMDM-loop combined with ReMDM-cap, in the inference-time scaling experiments ($T \geq L$), we report results using $\eta_{cap} = 0.020$. For the faster sampling experiments ($T < L$), we report results for $\eta_{cap} = 0.055$. For the ReMDM-rescale inference-time scaling experiments, we report results with $\eta_{rescale} = 0.015$ and for the faster sampling experiments, we use $\eta_{rescale} = 0.045$ in Table 1.

### E.3 ImageNet

In Table 7, we present the full results for varying the sampling temperature $\tau$ and the various ReMDM schedules and the their corresponding hyperparameters ($\eta_{cap}$ for ReMDM-cap. and $\eta_{rescale}$ for ReMDM-rescale). As noted in Section 5.1.2, while both MDLM and ReMDM benefit from using a softmax temperature of $\tau = 0.8$, the MaskGiT sampler produces the best results with no temperature scaling (i.e., $\tau = 1$). Of note, we use FID to determine which setting constitutes the 'best' results for each sampler. With reduced temperature, the entropy of the softmax is reduced leading to less diverse samples. This is reflected in the trade-off of the FID vs. IS metrics across sampling temperatures, with IS penalizing a lack of diversity less than FID [16].

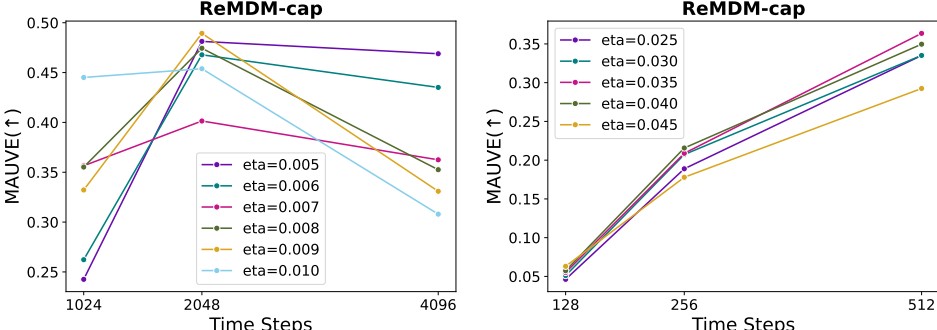

(a) MAUVE scores of ReMDM-cap inference-time scaling

(b) MAUVE scores of ReMDM-cap faster sampling

Figure 6: Impact of $\sigma_t$ on ReMDM-cap's unconditional text generation quality on OpenWebText.

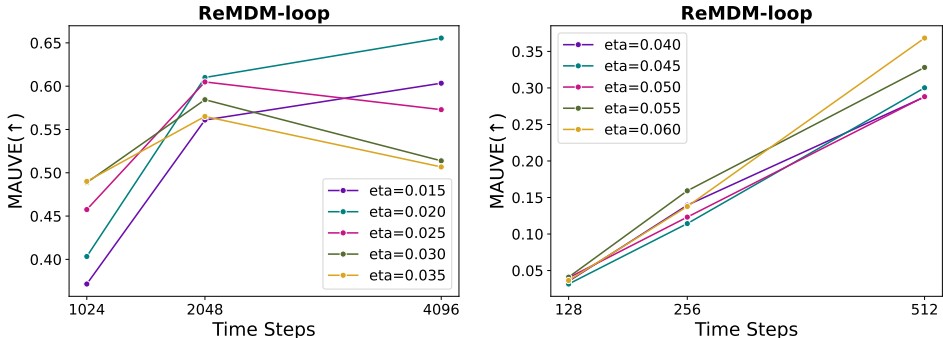

(a) MAUVE scores of ReMDM-loop inference-time scaling.

(b) MAUVE scores of ReMDM-loop faster sampling.

Figure 7: Impact of $\eta_{cap}$ on unconditional text generation quality on OpenWebText using the ReMDM-loop strategy with ReMDM-cap.

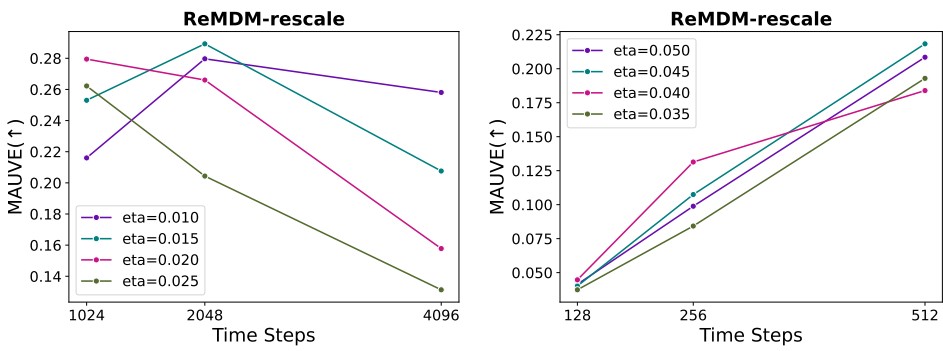

(a) MAUVE scores of ReMDM-rescale inference-time scaling.

(b) MAUVE scores of ReMDM-rescale faster sampling.

Figure 8: Impact of $\eta_{rescale}$ on unconditional text generation quality on OpenWebText using the ReMDM-rescale schedule.

Table 7: Discretized ImageNet conditional generation grid search over softmax temperature $\tau$ and ReMDM hyperparameters. Values reflect FID / IS for varying $T$. For each sampler, the row corresponding to the hyperparameter setup that is reported in the main Table 2 is **bolded**.

| Strategy | $\tau$ | FID ($\downarrow$) | | | IS ($\uparrow$) | | |
| | | $T = 16$ | $T = 32$ | $T = 64$ | $T = 16$ | $T = 32$ | $T = 64$ |
|---|---|---|---|---|---|---|---|
| MaskGiT | 0.6 | 10.06 | 11.03 | 11.72 | 236.90 | 244.79 | 243.64 |
| | 0.8 | 6.66 | 7.09 | 7.75 | 205.91 | 225.08 | 233.87 |
| | **1.0** | **6.74** | **4.92** | **4.85** | **155.32** | **181.57** | **196.38** |
| MDLM | 0.6 | 6.99 | 7.67 | 8.43 | 214.55 | 233.59 | 242.28 |
| | **0.8** | **7.88** | **5.37** | **4.69** | **140.97** | **169.79** | **187.93** |
| | 1.0 | 26.02 | 18.18 | 13.96 | 64.01 | 82.54 | 96.30 |
| ReMDM-cap, $\eta_{cap} = 0.01$ | 0.6 | 7.07 | 7.81 | 8.75 | 216.00 | 238.35 | 243.82 |
| | 0.8 | 7.71 | 5.20 | 4.56 | 141.31 | 172.66 | 194.77 |
| | 1.0 | 26.25 | 18.78 | 14.72 | 63.97 | 81.25 | 93.69 |
| ReMDM-cap, $\eta_{cap} = 0.02$ | 0.6 | 7.10 | 7.94 | 9.01 | 217.12 | 239.50 | 244.98 |
| | 0.8 | 7.55 | 5.00 | 4.53 | 142.33 | 178.32 | 198.87 |
| | 1.0 | 26.86 | 19.50 | 15.85 | 63.06 | 79.46 | 91.68 |
| ReMDM-cap, $\eta_{cap} = 0.05$ | 0.6 | 7.21 | 8.38 | 9.91 | 223.96 | 243.14 | 244.60 |
| | 0.8 | 7.24 | 4.83 | 4.52 | 147.20 | 184.21 | 211.06 |
| | 1.0 | 28.11 | 21.43 | 19.27 | 60.47 | 76.12 | 82.15 |
| ReMDM-rescale, $\eta_{rescale} = 0.01$ | 0.6 | 7.09 | 7.78 | 8.66 | 215.52 | 237.82 | 244.96 |
| | 0.8 | 7.75 | 5.24 | 4.58 | 141.11 | 171.56 | 193.73 |
| | 1.0 | 26.09 | 18.48 | 14.29 | 64.45 | 82.26 | 95.69 |
| ReMDM-rescale - $\eta_{rescale} = 0.02$ | 0.6 | 7.09 | 7.84 | 8.90 | 217.67 | 239.87 | 246.37 |
| | 0.8 | 7.64 | 5.07 | 4.48 | 142.50 | 176.57 | 197.35 |
| | 1.0 | 26.58 | 18.84 | 14.78 | 63.40 | 81.20 | 94.99 |
| ReMDM-rescale - $\eta_{rescale} = 0.05$ | 0.6 | 7.19 | 8.22 | 9.65 | 223.05 | 242.99 | 250.68 |
| | **0.8** | **7.40** | **4.92** | **4.45** | **145.27** | **182.05** | **209.45** |
| | 1.0 | 27.39 | 19.99 | 16.69 | 62.09 | 78.82 | 90.58 |
| ReMDM-conf | 0.6 | 7.46 | 8.54 | 9.82 | 221.25 | 243.56 | 251.83 |
| | 0.8 | 6.91 | 5.16 | 5.35 | 150.73 | 189.51 | 212.66 |
| | 1.0 | 22.14 | 13.14 | 8.88 | 73.00 | 101.79 | 126.29 |

## E.4 QM9

In Figures 10-25, we present the results from the extensive hyperparameter search we conducted across both properties, ring count (Figures 10-17) and QED (Figures 18-25), and guidance mechanisms D-CFG (Figures 10-13, 18-21) and D-CBG (Figures 14-17, 22-25).

### E.4.1 Drug-likeness Property Maximization

In Figure 9, we present D-CFG and D-CBG results for baseline and the 'best' ReMDM setting when maximizing the drug-likeness (QED) property. Similar to the results for maximizing the ring count property in Section 5.2, we find that ReMDM improves the novelty-property maximization frontier relative to MDLM when maximizing QED, although the benefits of ReMDM relative to MDLM are less pronounced for QED than for ring count.

For D-CFG, results reflect combining ReMDM-rescale ($\eta_{rescale} = 0.1$) with the confidence-based scheduler and switch strategy ($t_{switch} = 0.1$). For D-CBG, results reflect combining ReMDM-rescale ($\eta_{rescale} = 1.0$) with the confidence-based scheduler and switch strategy ($t_{switch} = 0.5$)

### E.4.2 Tuning $\eta_{rescale}$

**Larger $\eta_{rescale}$ is Better for Ring Count Maximization** Across settings, we find that generally, with some exceptions in the settings where the confidence-based schedule is not used (where ReMDM performs comparatively worse than when confidence is used, as discussed below), we see a trend where

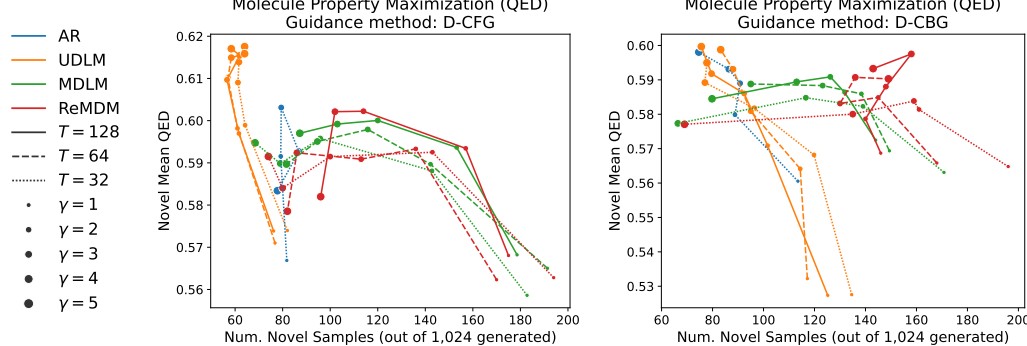

Figure 9: ReMDM improves steer-ability by extending the novelty-property maximization frontier. Controlled generation for drug-likeness (QED) maximization on QM9 dataset with varying inference compute $T$ and guidance strength $\gamma$. *(Left)* Discrete classifier-free guidance (D-CFG). *(Right)* Discrete classifier-based guidance (D-CBG) and FUDGE for AR.

large $\eta_{rescale}$ values lead to Pareto curves that are pushed further in the direction of more novelty and higher ring counts. Additionally, we find that for $\eta_{rescale} \geq 0.5$, the results are less sensitive to this parameter.

**Inconclusive $\eta_{rescale}$ Results for QED Maximization** For maximizing the QED property, the results are less conclusive. In the settings that use the confidence-based schedule, there is no clear $\eta_{rescale}$ that dominates the others. For settings that do not use the confidence-based schedule, we observe a trend where results improve with $\eta_{rescale} = 0.1$.

### E.4.3 Ablating Use of Confidence-based Schedule

For maximizing the QED property, we observe a clear pattern where incorporating confidence-based schedules improves results. For ring count, this trend holds true as well, especially for larger values of $\eta_{rescale}$, where the confidence score potentially helps temper large remasking probabilities for relatively 'safe' tokens, but the trend is less pronounced overall than for QED experiments.

### E.4.4 Tuning $t_{switch}$ / Loop Parameters

For both QED and ring count maximization, when using D-CFG as the guidance mechanism, we observe a general trend that activating ReMDM, with either the switch or loop strategies, benefits from starting later in the decoding process, i.e., smaller $t_{switch}$ / $t_{on}$. This trend also holds true for D-CBG in the QED experiments. A notable exception is when using D-CBG in the ring count experiments, where we see the opposite trend, i.e., activating ReMDM earlier (larger $t_{switch}$ / $t_{on}$) improves results.

### E.4.5 Comparing to FB and DFM Correctors

In Table 8, we compare ReMDM to the other predictor-corrector samplers, namely FB [5] and DFM [13] in the context of maximizing ring count using conditional generation (i.e., $\gamma = 1$ for D-CFG). ReMDM better trades-off novel sample generation and maximizing the property of interest.

## F   Generated Samples

### F.1   Generative Perplexity Hacking Example

We visualize a generative perplexity hacking example in Figure 26. In this example, ReMDM-cap on OWT with $\eta_{cap} = 0.2$ and $T = 1024$ generates a sequence filled with repetitive tokens without real semantics, but the generative perplexity is as low as 1.5.

Table 8: ReMDM better trades-off sample novelty and ring count maximization compared to other predictor-corrector methods.

| Sampler | Num Novel | Novel Ring Count Mean |
|---------|-----------|------------------------|
| $T = 32$ | | |
| FB | 268 | 4.35 |
| DFM | **341** | 4.57 |
| ReMDM | 273 | **4.64** |
| $T = 64$ | | |
| FB | 304 | 4.42 |
| DFM | 311 | 4.57 |
| ReMDM | **318** | **4.61** |
| $T = 128$ | | |
| FB | 295 | **4.53** |
| DFM | 236 | 4.52 |
| ReMDM | **348** | **4.53** |

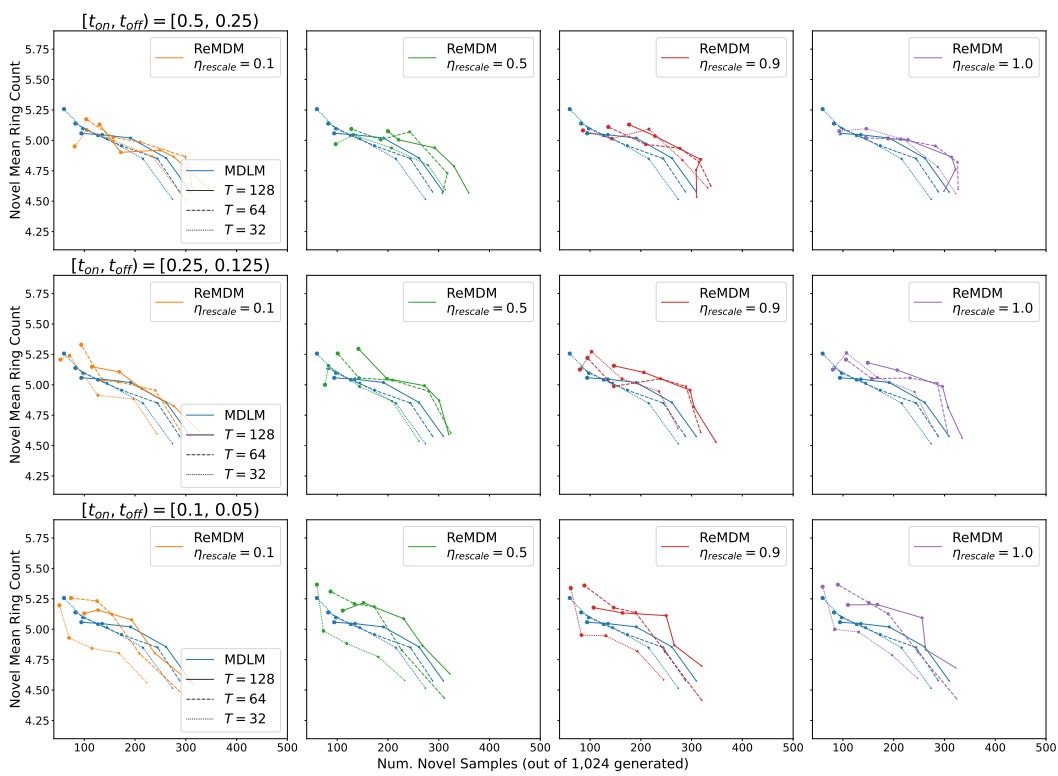

Figure 10: ReMDM-rescaled with **loop** and **confidence-based** schedules hyperparameter tuning for maximizing **ring count** using **D-CFG**. Larger marker sizes indicate larger $\gamma$ values.

## F.2   OpenWebText

We visualize the generated sentences of MDLM ($T = 4096$) (Figure 27), MDLM+DFM ($T = 4096$) (Figure 28), and ReMDM ($T = 4096$) (Figure 29). We find that the MDLM samples contain many uncorrelated semantic fragments, with grammar errors appearing very often. MDLM+DFM can formulate the text around a special topic, but the internal logic is not fluent. In contrast, ReMDM is able to generate high quality, fluent English with a clear semantic topic.

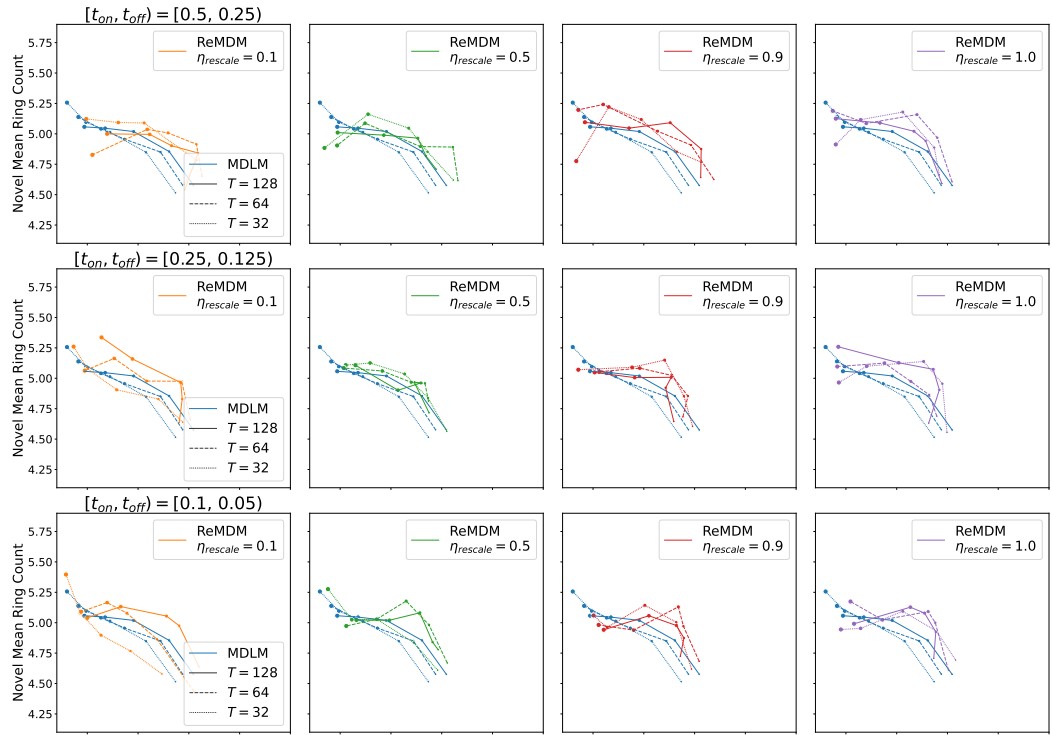

Figure 11: ReMDM-rescaled with **loop** and **without confidence-based** schedules hyperparameter tuning for maximizing **ring count** using **D-CFG**. Larger marker sizes indicate larger $\gamma$ values.

## F.3 LLaDA

We visualize examples of LLaDA answering Countdown in Figure 30, and LLaDA answering TruthfulQA questions in Figure 31. In both cases, we use few-shot evaluation. Particularly, we use 4-shot for Countdown and 6-shot for TruthfulQA. The few shot examples presented in the prompt are the same for all questions in one task.

## G  Assets

In Table 9, we list the datasets (and corresponding licenses, when available) used in this work. In Table 10, we list the software packages (and corresponding licenses) used in this work.

Table 9: Datasets (and corresponding licenses) used in this work.

| Dataset | License |
| --- | --- |
| ImageNet [8] | ImageNet License |
| OpenWebText [14] | Creative Commons CC0 license ("no rights reserved") |
| QM9 [40, 39] | N/A |
| TruthfulQA [27] | Apache-2.0 |

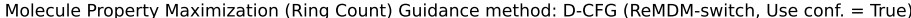

Molecule Property Maximization (Ring Count) Guidance method: D-CFG (ReMDM-switch, Use conf. = True)

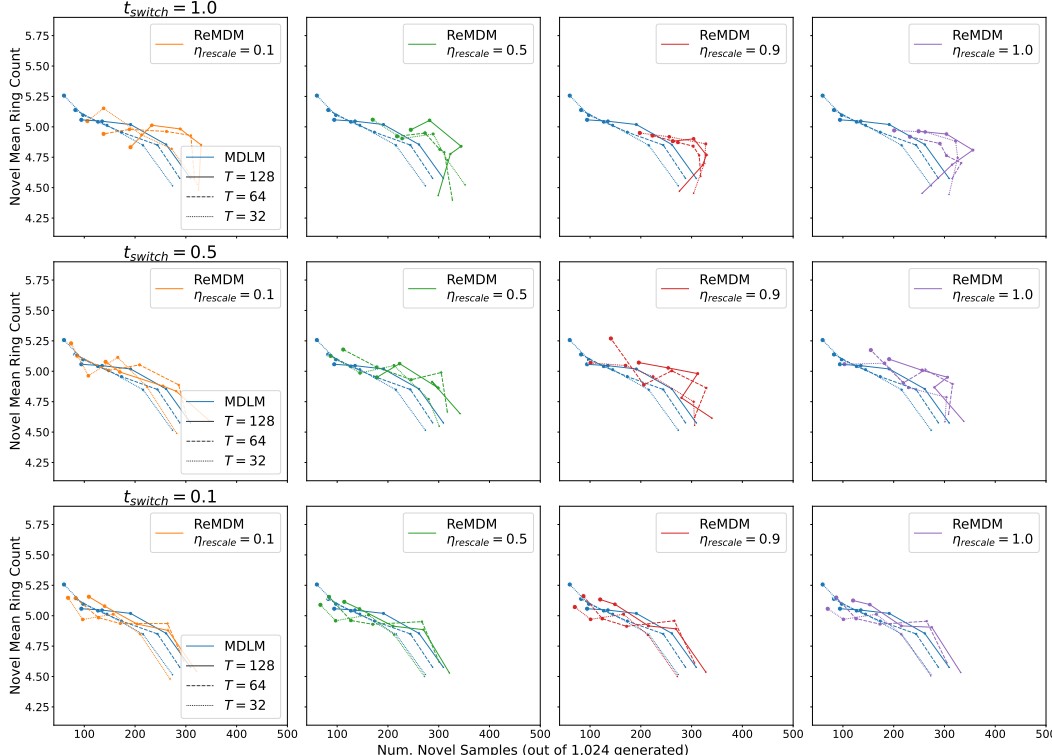

Figure 12: ReMDM-rescaled with **switch** and **confidence-based** schedules hyperparameter tuning for maximizing **ring count** using **D-CFG**. Larger marker sizes indicate larger $\gamma$ values.

Table 10: Software (and corresponding license) used in this work.

| Library | License |
| --- | --- |
| Guided Diffusion [9] | MIT |
| HuggingFace [60] | Apache 2.0 |
| Hydra [61] | MIT |
| Jax [4] | Apache 2.0 |
| NumPy [15] | NumPy license |
| MaskGiT [7] | Apache 2.0 |
| Matplotlib [20] | Matplotlib license |
| OmegaConf | BSD 3-Clause |
| Pandas [33] | BSD 3-Clause "New" or "Revised" |
| PyTorch [34] | BSD-3 Clause |
| PyTorch Lightning [11] | Apache 2.0 |
| RDKit [23] | BSD 3-Clause "New" or "Revised" |
| Seaborn [58] | BSD 3-Clause "New" or "Revised" |
| MAUVE [36, 37] | GPLv3 |
| Language Model Evaluation Harness [12] | MIT |

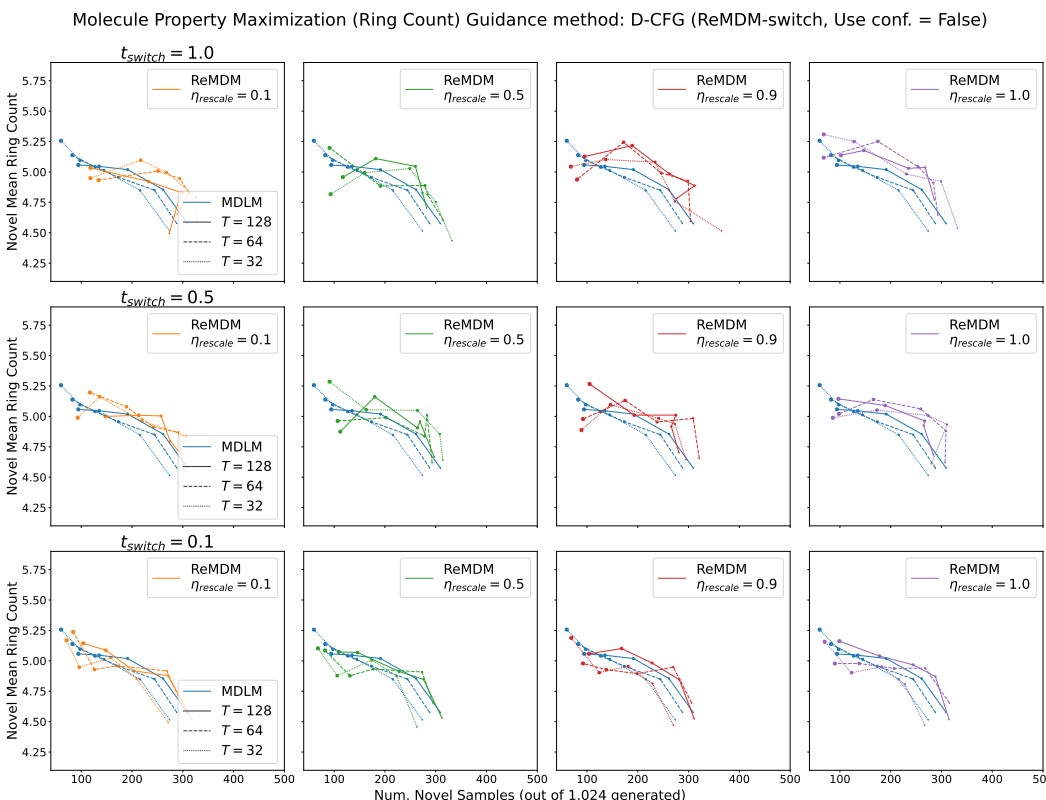

Figure 13: ReMDM-rescaled with **switch** and **without confidence-based** schedules hyperparameter tuning for maximizing **ring count** using **D-CFG**. Larger marker sizes indicate larger $\gamma$ values.

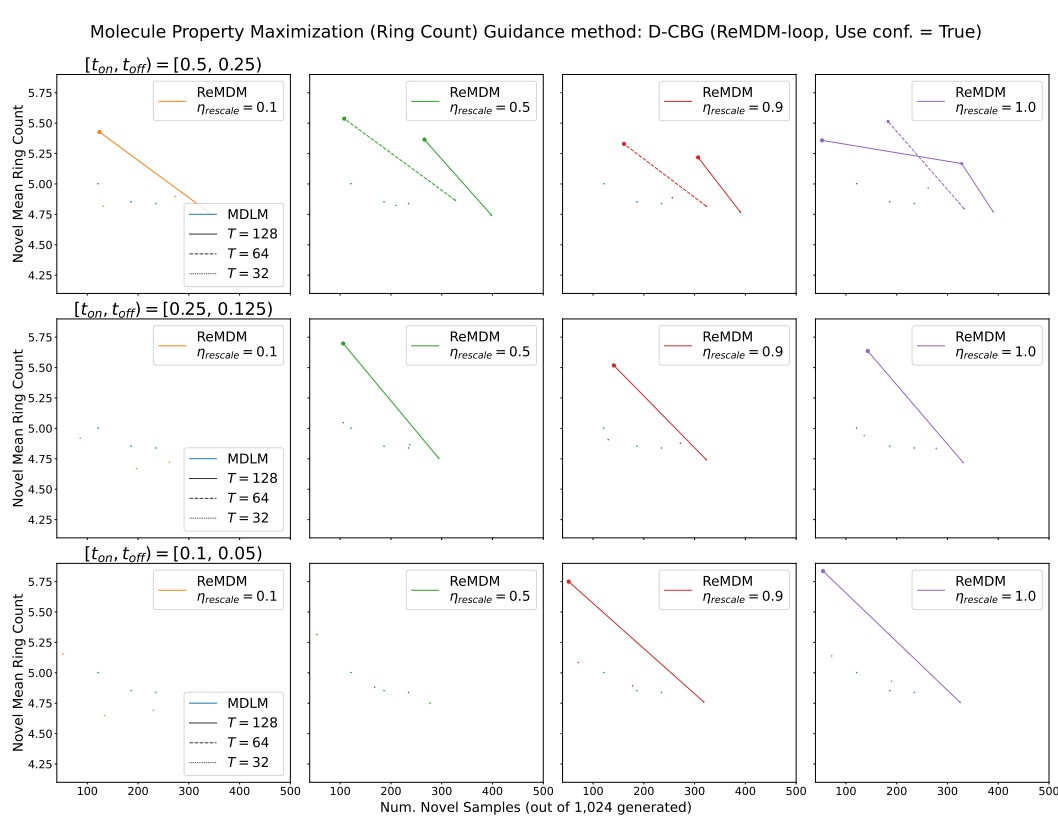

Figure 14: ReMDM-rescaled with **loop** and **confidence-based** schedules hyperparameter tuning for maximizing **ring count** using **D-CBG**. Larger marker sizes indicate larger $\gamma$ values.

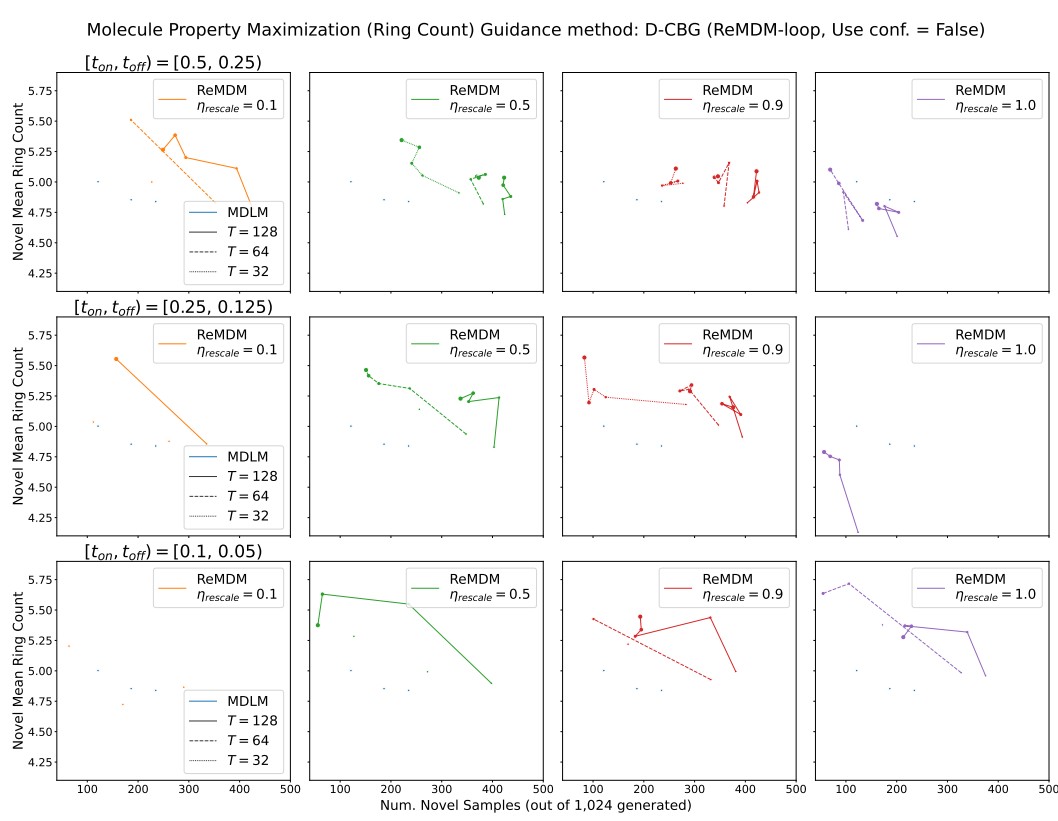

Figure 15: ReMDM-rescaled with **loop** and **without confidence-based** schedules hyperparameter tuning for maximizing **ring count** using **D-CBG**. Larger marker sizes indicate larger $\gamma$ values.

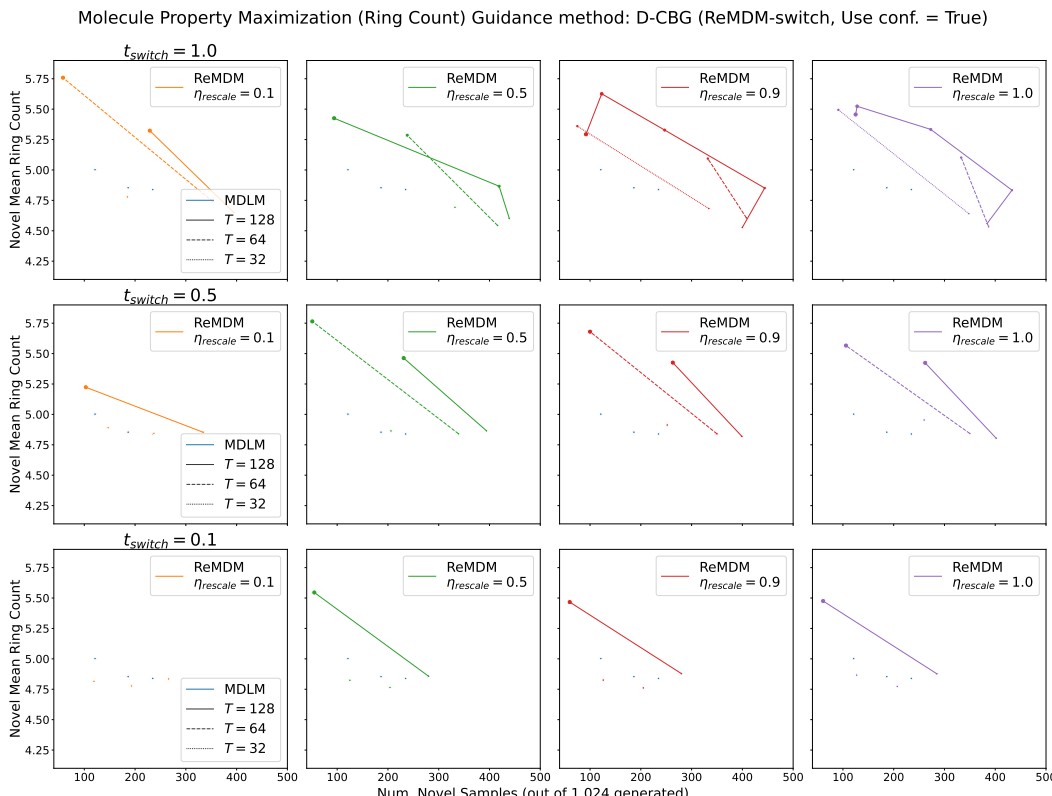

Figure 16: ReMDM-rescaled with **switch** and **confidence-based** schedules hyperparameter tuning for maximizing **ring count** using **D-CBG**. Larger marker sizes indicate larger $\gamma$ values.

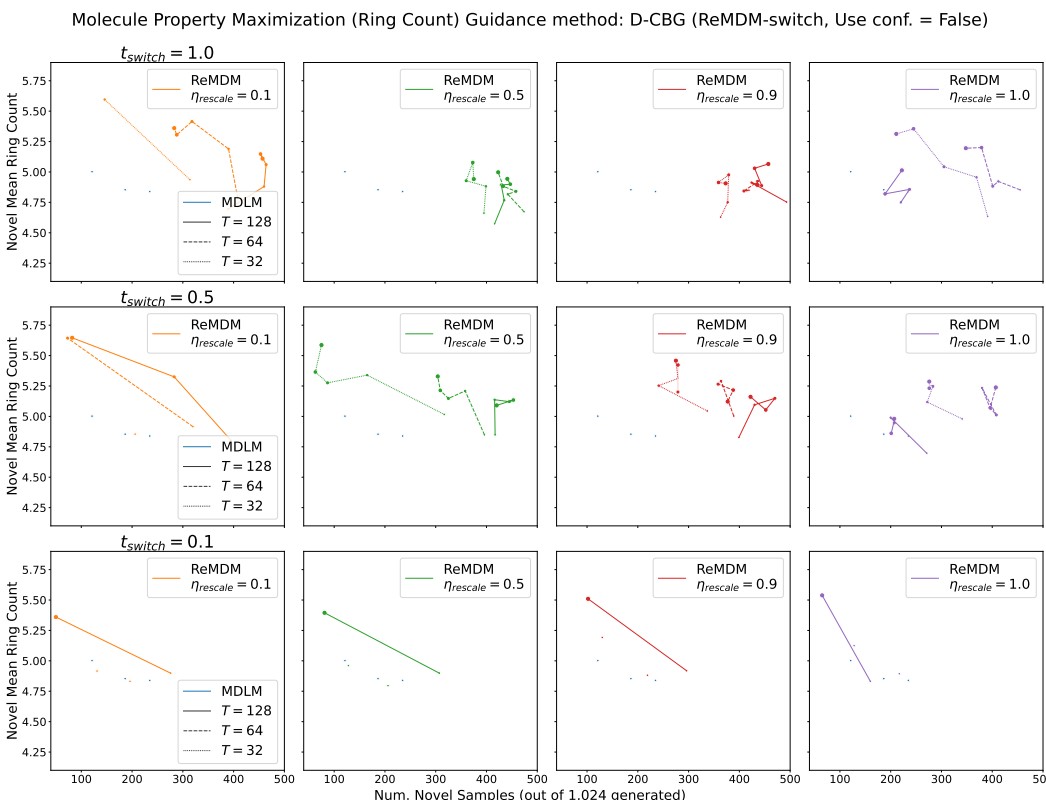

Figure 17: ReMDM-rescaled with **switch** and **without confidence-based** schedules hyperparameter tuning for maximizing **ring count** using **D-CBG**. Larger marker sizes indicate larger $\gamma$ values.

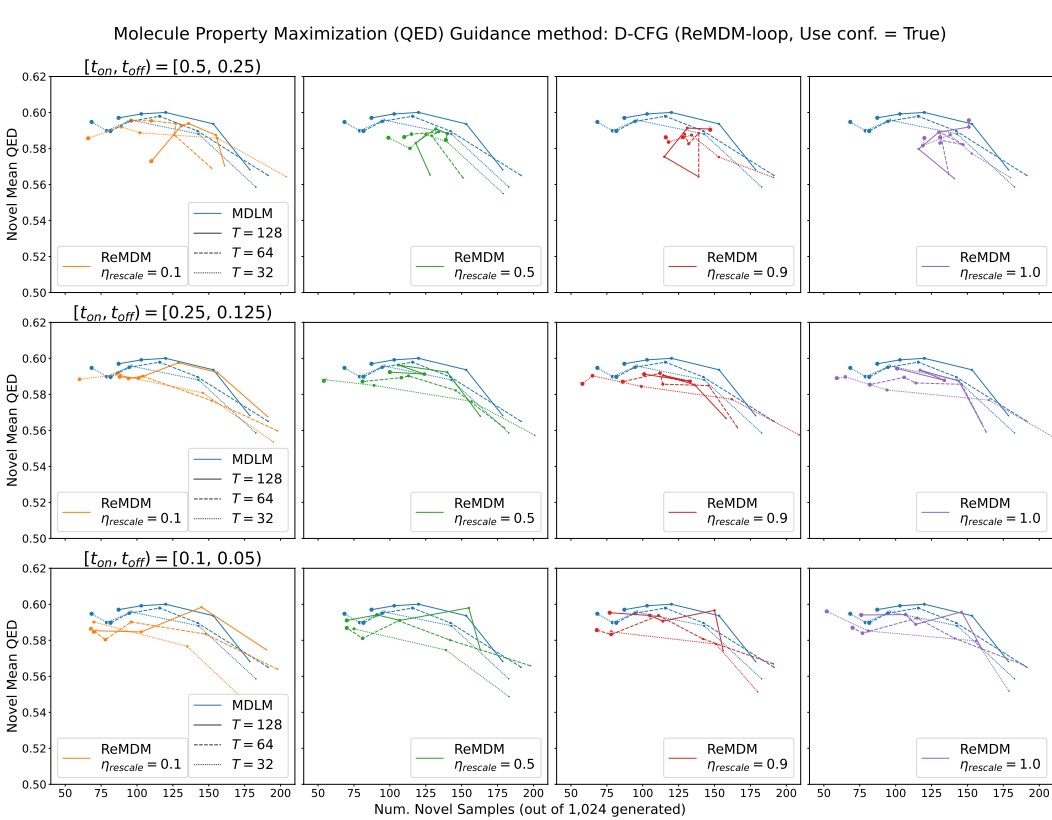

Figure 18: ReMDM-rescaled with **loop** and **confidence-based** schedules hyperparameter tuning for maximizing **drug-likeness (QED)** using **D-CFG**. Larger marker sizes indicate larger $\gamma$ values.

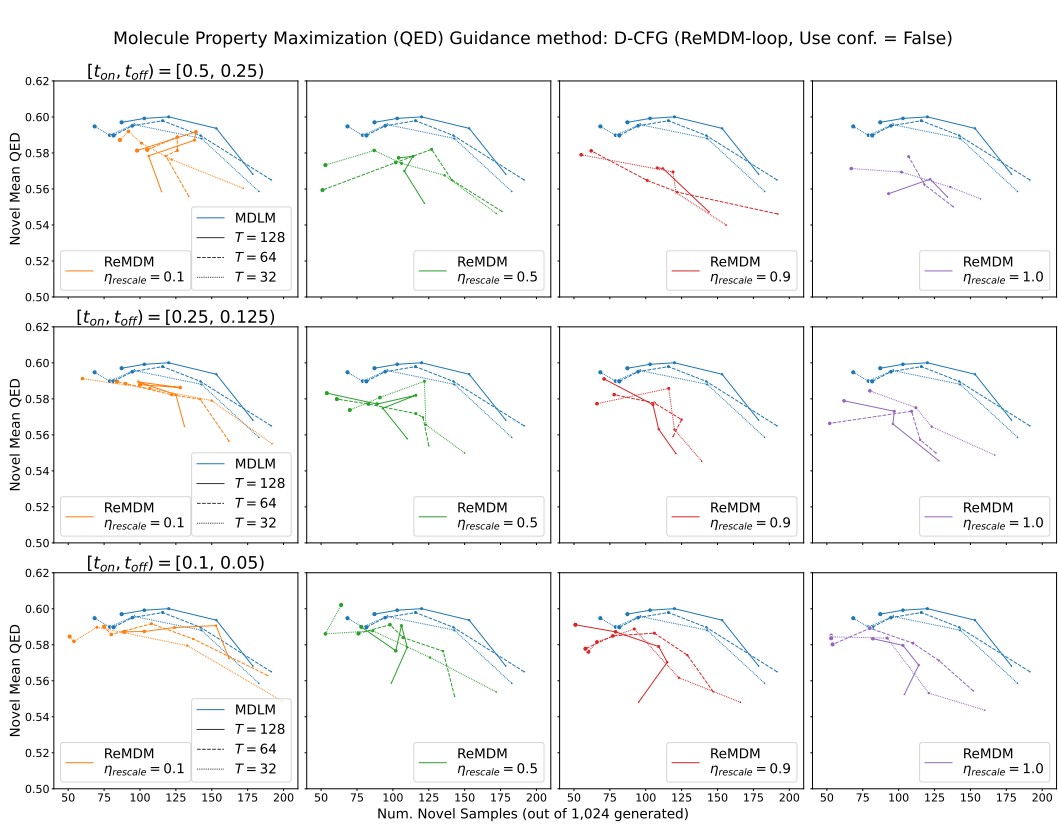

Figure 19: ReMDM-rescaled with **loop** and **without confidence-based** schedules hyperparameter tuning for maximizing **drug-likeness (QED)** using **D-CFG**. Larger marker sizes indicate larger $\gamma$ values.

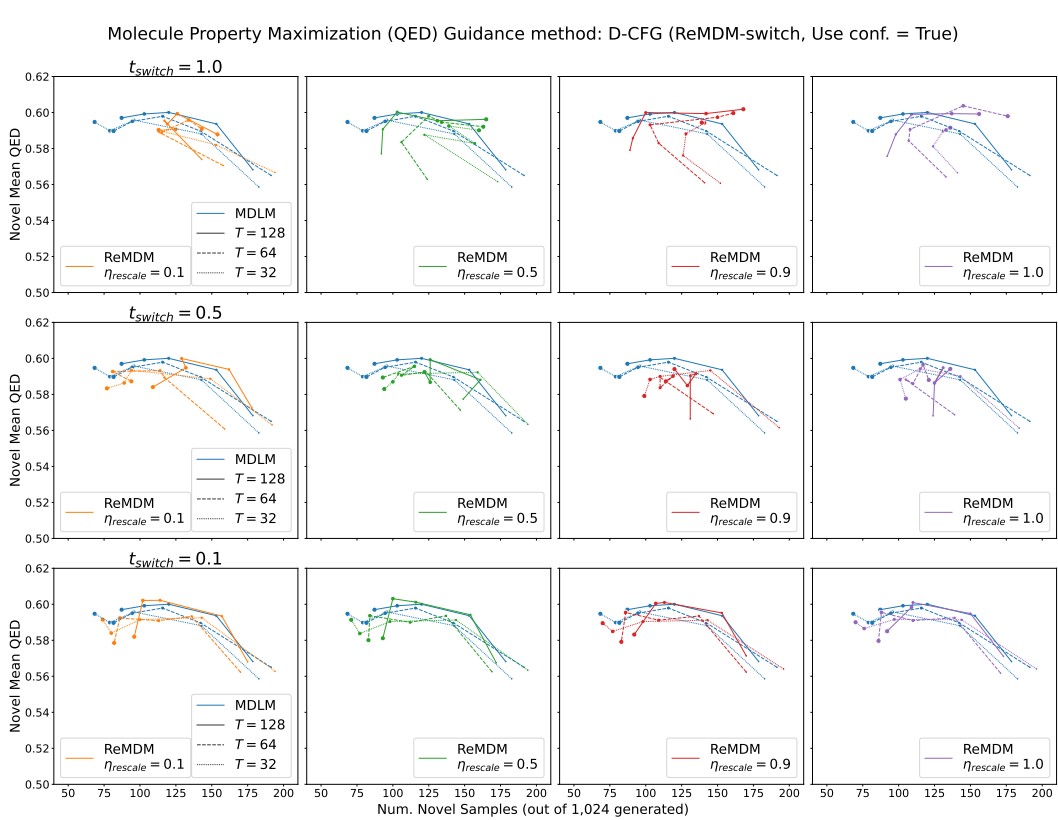

Figure 20: ReMDM-rescaled with **switch** and **confidence-based** schedules hyperparameter tuning for maximizing **drug-likeness (QED)** using **D-CFG**. Larger marker sizes indicate larger $\gamma$ values.

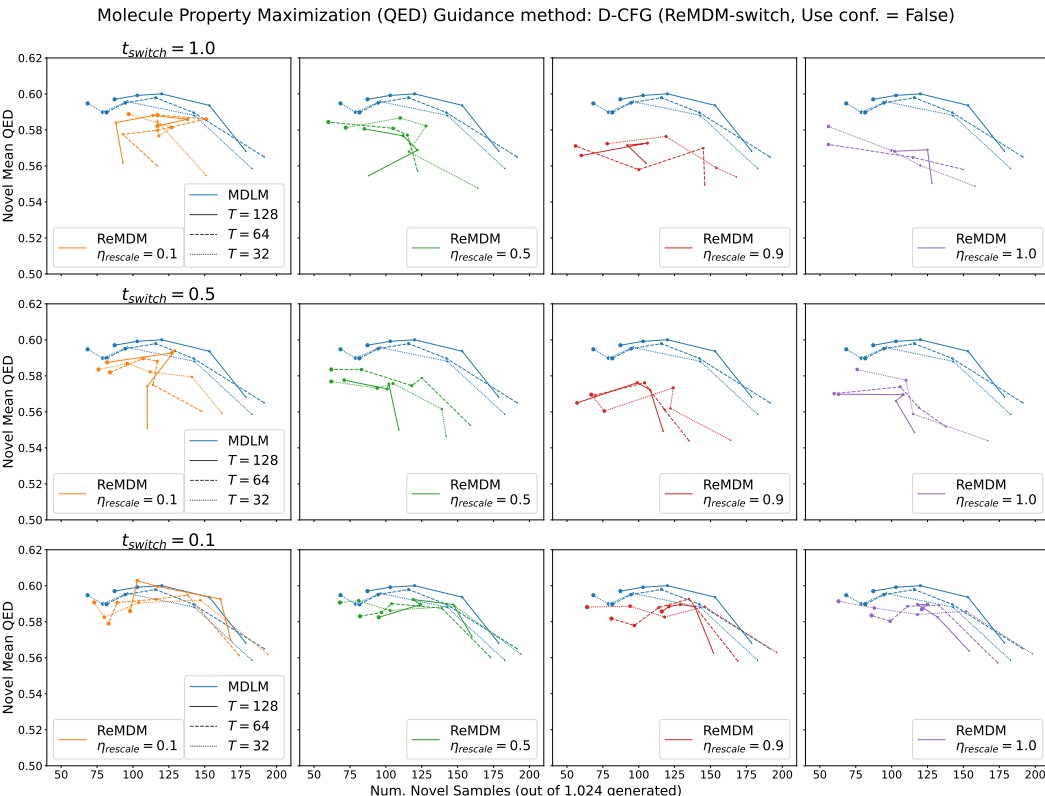

Figure 21: ReMDM-rescaled with **switch** and **without confidence-based** schedules hyperparameter tuning for maximizing **drug-likeness (QED)** using **D-CFG**. Larger marker sizes indicate larger $\gamma$ values.

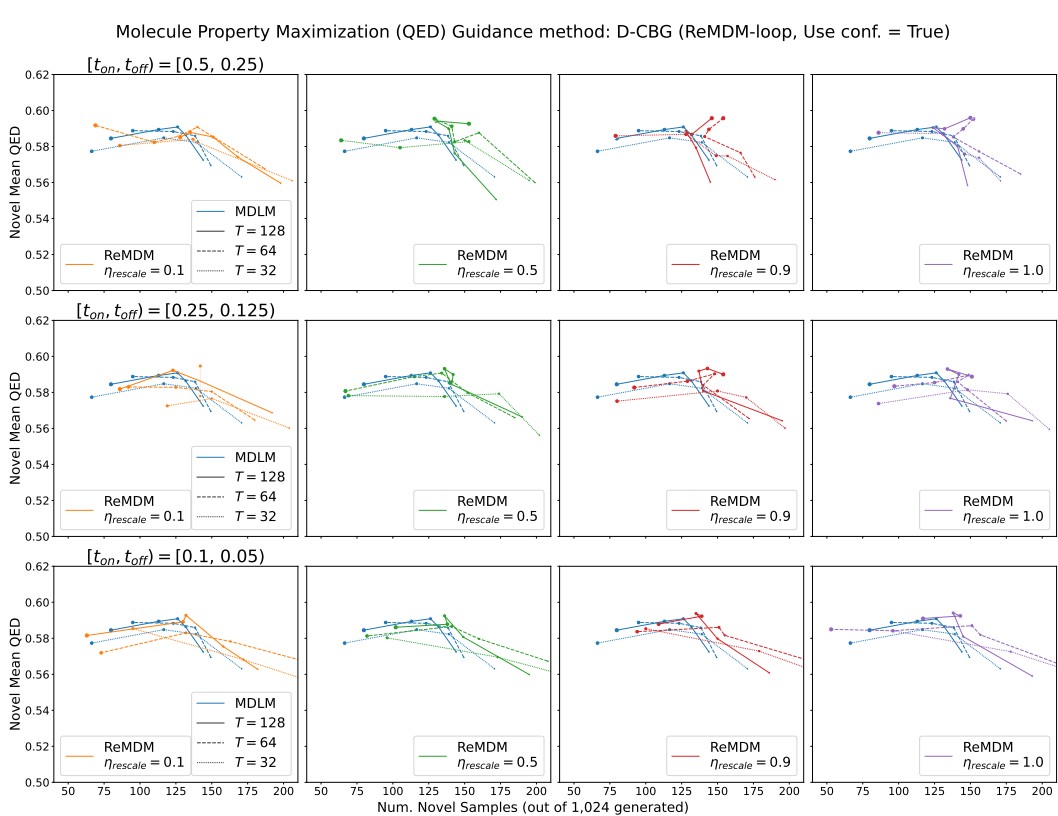

Figure 22: ReMDM-rescaled with **loop** and **confidence-based** schedules hyperparameter tuning for maximizing **drug-likeness (QED)** using **D-CBG**. Larger marker sizes indicate larger $\gamma$ values.

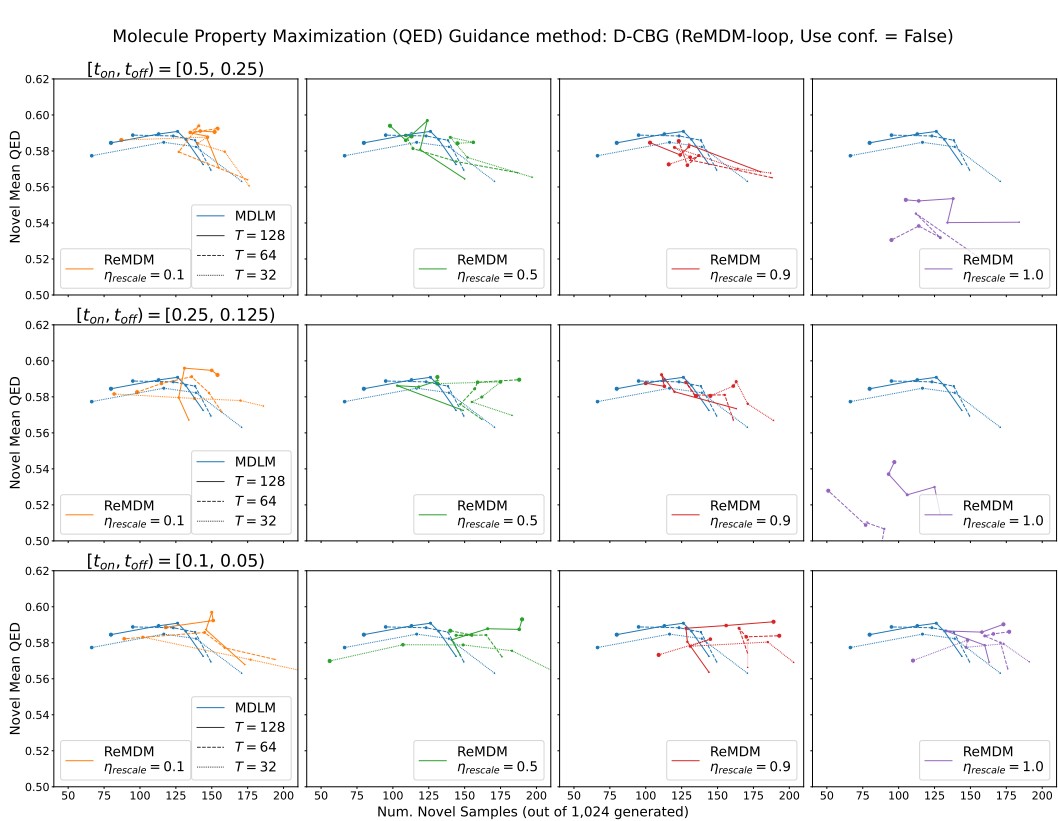

Figure 23: ReMDM-rescaled with **loop** and **without confidence-based** schedules hyperparameter tuning for maximizing **drug-likeness (QED)** using **D-CBG**. Larger marker sizes indicate larger $\gamma$ values.

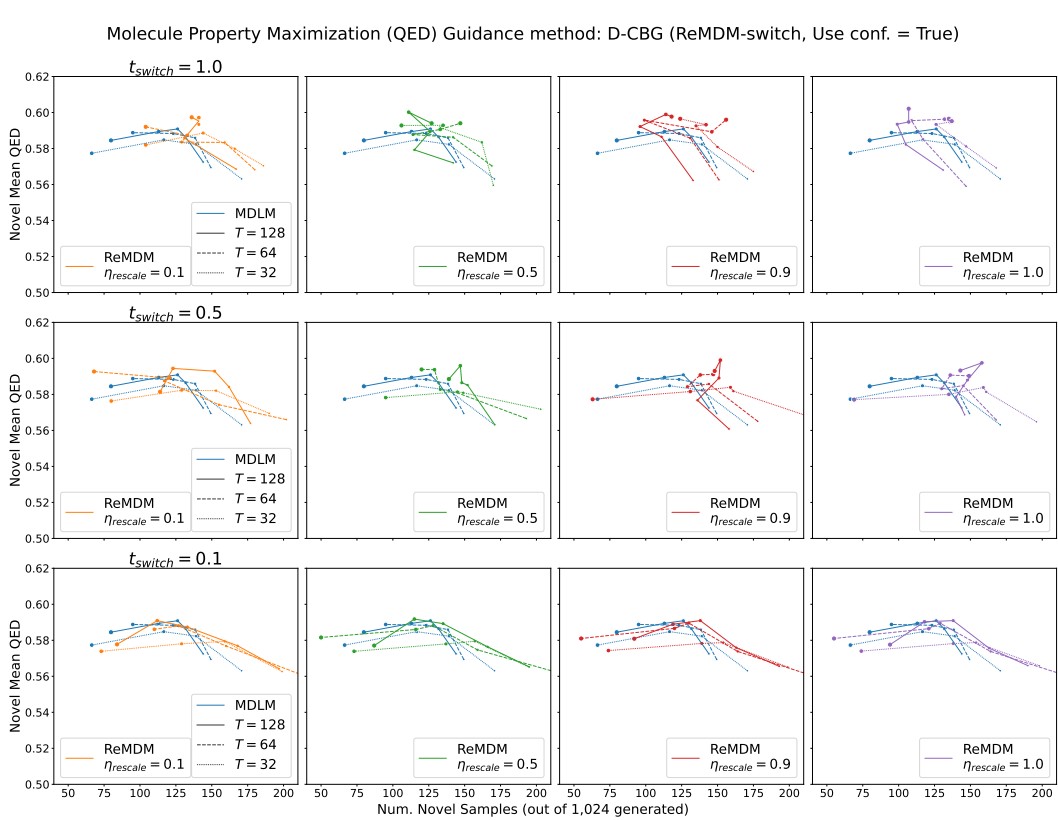

Figure 24: ReMDM-rescaled with **switch** and **confidence-based** schedules hyperparameter tuning for maximizing **drug-likeness (QED)** using **D-CBG**. Larger marker sizes indicate larger $\gamma$ values.

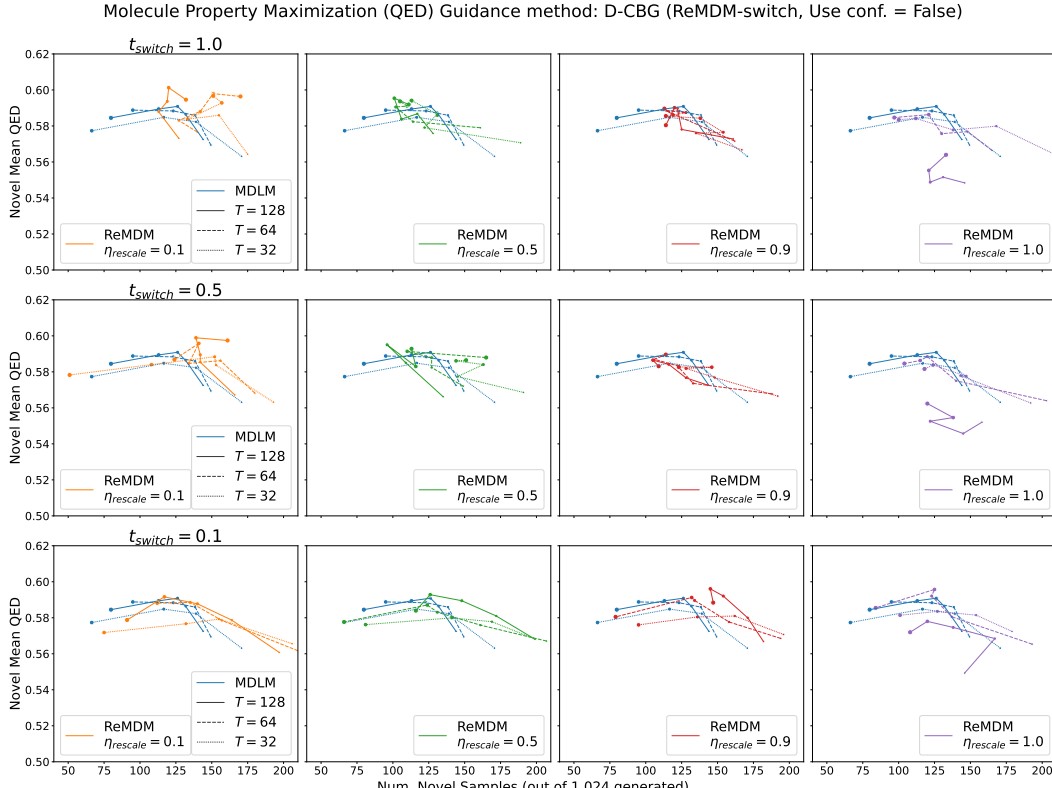

Figure 25: ReMDM-rescaled with **switch** and **without confidence-based** schedules hyperparameter tuning for maximizing **drug-likeness (QED)** using **D-CBG**. Larger marker sizes indicate larger $\gamma$ values.

**<|endoftext|>**, my love, my love, my love, my love, my love, my love, oh, oh, oh, oh, oh, oh, oh, oh, my love, my love, my love, my love, oh, oh, oh, oh, oh, oh, oh, oh, oh, oh, oh, oh, oh, oh, oh, oh, my love, my love, my love, my love, my love, my love, my love, hate, hate, hate, hate, hate, hate, hate, hate, hate, hate, hate, hate, hate, hate, hate, hate, hate hate hate hate hate hate hate hate hate hate hate hate hate hate hate hate hate hate hate hate hate hate hate hate hate hate hate hate hate hate hate hate hate hate hate hate hate hate hate hate hate hate hate hate hate hate hate hate hate hate hate hate hate hate hate hate hate hate hate hate hate hate hate hate hate hate hate hate hate hate hate hate hate hate hate hate hate hate hate hate hate hate hate hate hate hate hate hate hate hate hate hate hate hate hate hate hate hate hate hate hate hate hate hate hate hate hate hate hate hate hate hate hate hate hate hate hate hate hate hate hate hate hate hate hate hate hate hate hate hate hate hate hate hate hate hate hate hate hate hate hate hate hate hate hate hate hate hate hate hate hate hate hate hate hate hate hate hate hate hate hate hate hate hate hate hate hate hate hate hate hate hate hate hate hate hate hate hate hate hate hate hate hate hate hate hate hate hate hate hate hate hate hate hate hate hate hate hate hate hate hate hate hate hate hate hate hate hate hate HHHHHHH-HHHHHHHHHHHHHHHHHHHHHHHHHHHHHHHHHHHHHHHHHHHHHHHHHHHHHHHH-HHHHHHHHHHHHHHHHHHHHHHHHHHHHHHHHHHHHHHHHHHHHHHHHHHHHHHHHHHHHHHH HHHHHHHHHHHHHHHHHHHHHHHHHHHHHHHHHHHHHHHHHHHHHHHHHHHHHHHHHHHHHHH HHHHHHHHHHHHHHHHHHHHHHHHHHHHHHHHHHHHHHHHHHHHHHHHHH hate hate hate hate hate hate hate hate hate hate hate hate hate hate hate hate hate hate hate hate hate hate hate hate hate hate hate hate hate hate hate hate hate hate hate hate hate hate hate hate hate hate hate hate hate hate hate hate hate hate hate hate hate hate hate hate hate hate hate hate hate hate hate hate hate hate hate hate hate hate hate hate hate hate hate hate hate hate hate hate hate hate hate hate hate hate hate hate hate hate hate hate hate hate hate hate hate hate hate hate hate hate hate hate hate hate hate hate hate hate hate hate hate hate hate hate hate hate hate hate hate hate hate hate hate hate hate hate hate hate hate hate hate hate hate hate hate hate hate hate hate hate hate hate hate hate hate hate hate hate hate hate hate hate hate hate hate hate hate hate hate hate hate hate hate hate hate hate hate hate hate hate hate hate hate hate hate hate hate hate hate hate hate Mama Mama Mama Mama Mama Mama Mama Mama Mama Mama Mama Mama Mama Mama Mama Mama Mama Mama Mama Mama Mama Mama Mama Mama Mama Mama Mama Mama Mama Mama Mama Mama Mama Mama Mama Mama Mama Mama Mama Mama Mama Mama Mama Mama Mama Mama Mama Mama Mama Mama Mama Mama Mama Mama Mama Mama Mama Mama Mama Mama Mama Mama Mama Mama Mama hate hate hate hate hate hate hate hate hate hate hate hate hate hate hate hate hate hate hate hate hate hate hate hate hate hate hate hate hate hate hate hate hate hate hate hate hate hate hate hate hate hate hate hate hate hate hate hate hate hate hate hate hate hate hate hate hate hate hate hate hate hate hate hate hate hate hate hate hate hate hate hate hate hate hate hate hate hate hate hate hate hate hate hate hate hate hate hate hate hate hate hate hate hate hate hate hate hate hate hate hate hate hate hate hate hate hate hate hate hate hate hate hate hate hate hate hate hate hate hate hate hate hate hate hate hate hate hate hate hate hate hate hate hate hate hate hate hate hate hate hate hate hate hate hate hate hate hate hate hate hate hate hate hate hate hate hate hate hate hate hate hate hate hate hate hate hate hate hate hate hate hate hate hate hate hate hate hate hate hate hate hate hate hate hate hate hate hate hate hate hate hate hate hate hate hate hate hate hate hate hate hate**<|endoftext|>**

Figure 26: Example of generative perplexity hacking. The generated sequence by ReMDM-cap on OWT with $\eta_{cap} = 0.2$ and $T = 1024$ features repetitive words without real semantics. Gen PPL. is 1.5 and entropy is 1.04. **|endoftext|** token is **bolded** to emphasize the boundary between paragraphs.

**|endoftext|** might be seen as a tough choice, it's smart to do so.

According to past reports, the SIM cards are only made available in the powerful manufacturer's range of carrier variants. And though the ZTE does its own case with these phones, this variant will not be disclosed.

We also know the color of the phone and what it also will cost remains to be seen. With the Carrington showdown, the new edition of the exclusive TVD features more pictures with Samsung.

**|endoftext|** Goodes

The Derby winner was named first-team of the Match on the BBC evening.

He feels his position as manager at Southall last season was a great honour and understands the surarious delight in earning that he can't complain about delivering an endless ovation.

The Newcastle coach knows how describes how people like their approach to games. As his biggest fans, he has criticised Paul Cruy and traded barbs with Chris Froome.

And sad to admit, he admits he did not get everything right. He just betrayed them in his intention and left an exasperation in others.

I've finished telling players that we'd get that, and they continued that attitude to how we didn't think we deserved the job ©Newcastle United Rovers manager Tony Wheeler

All this sound of being needed for interviews in various places.

"I'm quite pleased by the passion the fans have shown for me but believe me, I'm not the fan - I find the person out. It would be an extremely nice position to be in.

Illuminate. We'd like to think, not to get a player based on Newcastle's team when he does take charge.

"My head coach says, 'Listen, it's just too much to cry,' Tony Wheeler told the BBC on Tuesday about the injury he played in the Northern League final at Paroles.

"If the player is a part in the starting team, then I'm confident he would be put in there, but I also've finished telling players we thought we'd get the job. We had good options to go in the squad.

"We had, obviously, the people that were always willing for us to get the best player we could because as coaches we felt we had the best players, and we felt that no one had a choice.

"And the only thing [I] was just going on to say was that we were aware beforehand that it was an unusual situation for us to hook him up with the potential of [Meagan] Forster.

"We like his game and he likes ours, and if he's a good striker, he can go out there and play both for the team and for ourselves.

"Stephen [Biggish] Jackson, who's worked with here, obviously respects me with great respect and likes his opinions, but I think [Tony Wheeler] can say that it's satisfactory.

"We've got lots of many, and them combine them with Van Ferdinand. All three people we've been approached to, they need to be taken.

"Could something be done to them? Not be done."

United are certainly prepared to lose their star man for Tuesday's Rovers clash, with Borby to take charge of their trip to Dunferm on Saturday.

Although they have won five of their six games, they sit sixth in the Northern League, which includes the Parkhead and the two assistant charges but on Tuesday their only goal of the season is worthy of their reward.

United have more goals than Palace, scoring both goals this season despite the best form of Johnson as new coach

Winger Darius Boyd joined Derby after replacing Marlon Wickham in June and recently, he stayed for a year on loan until he joined Newcastle in defiance of Tony Wheeler.

Tom Williams, a future United captain, defender and striker, told BBC One Sports: "Being in Rovers is great for them.

"They develop their chemistry very well in the dugout and they try to stay in good relationships together.

"They're unabashed about that you can play with both of them now, so they've stayed loyal."

Brendan Rodgers, the current Newcastle United manager after Wheeler took over from Mick Jagger, and succeeded David Moyes, as the most paid Premier League manager in 1997 while the Scot was the highest paid Everton manager.

Leicester manager Leicester Leicester is yet to make terms with the joys of the job and has become so indebted he offered to forgive Newcastle United's owners for the cost of lost profits after buying a stake in the club earlier this year.

**|endoftext|** Sometimes, I enjoy having colleagues "embarrassed" to compete in sports, especially it comes to being a one-richen employee's career. I can't own the jobs of the colleagues that pressure us from Chinese and Australian competitive athletes and even to compete in the United States. They seem arrogant

**|endoftext|**

Figure 27: Sample from MDLM and $T = 4096$ diffusion steps. **|endoftext|** token is **bolded** to emphasize the boundary between paragraphs.

**|endoftext|** a component to renovating the park, but no one wants to involve others in it before facing a backlash.

"We are going to stand up and try to make a real statement on this," Thomas told reporters at a rally Tuesday night. "I would much rather have that there be a consensus not just to make a decision, but to have a very real discussion, and to begin to build a movement."

The idea behind the proposal is a call to the public, council and the youth advisory board, which was approved by residents on the first day of the meeting.

Thomas said the message will be important to Hamilton youth.

"It's going to be very much about a number of people," he said. "There will be arguments on both sides. It is not going to be all about the type of person that you are or the committee member that you are with a particular number of people."

Thomas said that the plan is really about an initiative to help the community do both of those things.

"This isn't going to be a positive thing, as it should be, but it's going to put people in position to make sure that our focus is not just about how to deal with poverty ... and with homelessness. Our focus is on keeping the people in our community affordable and to create opportunities for their families.

"We have —believe it or not —we have increased community involvement in our city," Thomas said. "I think it's obvious from all of the issues now that we'll consider resource-generating development and how to devribute those resources."

The renovated park is one of the 14 parks identified in the city's 2008 map of the park's 2-acre site, which was opened up for a community arts festival.

City staff started coming up with the project in late December, but the public has yet to come to terms.

For their part, residents sign the petition only to receive two differing opinions.

Although there are concerns that it would be an uphill battle that requires council approval, officials said Tuesday that the process could eventually take shape, with councillors likely to support it.

Hamilton city hall spokesman John Little told reporters that the city would have to address a petition from the public if there is one on site.

"We are not able to change the outcome because we are discussing it internally, so what we have has not made an official decision," Little said.

Thomas said staff wants to hear how the public feels about the project, and it must be incorporated in such a prompt fashion that the city is looking at more permanent changes.

"We implemented a 30-year plan, about 10 years ago, and we will continue it this year," Thomas said. "We want to once again really emphasize community involvement. I mean, as we say we take this kind of step, we don't want to make separate efforts to change different things," he said.

Any plans would be discussed with community boards, Conley said.

Hamilton city councillor Steve Cowen said if residents sign the petition, "actions have to be done, knowing when these changes are needed and that we will have the right options to move forward."

Cowen said two renovations along with suggestions for revenue set by the Hulbert Company, which operates the Gerrard Elementary school theatre, will be included on city documents and website. They would set aside $2 million more than the result of Washtenaw Park Plaza renovation plans, sparked by the QEDA expansion that renovated the former Washtenaw Historic Restaurant and Bakery on Gerrard.

The company has also purchased Gerrard and turned it into a New American Food, Inc. restaurant, and brought the business before the city council in late 2015.

Hamilton city council has been having this conversation for more than a year, but it usually happens only now.

"It's really not, 'We need a meeting soon,' why we need to do this," Conley said. "It's just a forum where people can voice their views and inform council, and within the next year, hopefully council will do the right thing to do."

With John Elliott:

905-526-2431

mmelliott@thespec.com

**|endoftext|** Written by Steven B. Ollander

In the last five years, we have seen huge financial crises around the world, and not just economic ones. We've also seen rapid changes in economic systems, in central and peripheral countries.

All these systems are in the hands of banks, but there are other forces that have also been used to dump more money into other countries

**|endoftext|**

Figure 28: Sample from MDLM+DFM and $T = 4096$ diffusion steps. **|endoftext|** token is **bolded** to emphasize the boundary between paragraphs.

**|endoftext|** The Audi D3 is one of the fastest and most capable sports cars we've seen on the market. The retractable front wings were added to make the D3 R a more sporty car. The front wing design is painted yellow, which is a big surprise. It looks like the A4 R in a completely different color scheme. While the front wings retract, the cooling system is a drag for sure. The front and rear seats are also light-weighted, which is also convenient for a racing driver. The car also features replaceable steering wheels for higher stability and better handling. D3 R is easy to drive and easy to use thanks to adaptive suspension technology.

Here are a few of the other features Audi had in mind:

Top speed is only four seconds higher, but more turbocharged engine power has been added to make it even faster.

By means, you reach 60 mph in five seconds with strong acceleration. The front wing design makes the engine a higher strength, making it just more powerful.

Speaking of power, D3 R is equipped with a 3.0 litre V8 with Adaptive Brake Control, which means you won't complain if you may have to move around it in wet conditions. The engine has been also upgraded, making a 204.26 horsepower.

This is a 2-liter V8, which means it can easily produce as much as 132 bhp.

By comparison, a 3.0 litre V8 produces 227.4 bhp which is a bit too much for sporty cars. If you want a sporty car, your only choice is to use this kind of engine in the guise of the D3 R.

Other innovative changes that create better handling in the car are the retractable rear wing and the front suspension. The forks, front and rear, are fold-down. The 17-inch adjustable steering wheel makes it easier for the driver. The car also sports a sport seats, which is quite different from the A4 R. Behind the steering wheel, the sport driving mode has a full black screen touch option. You can move the car up and down by taping the button. The buttons can be mounted on the display, giving you more input of the screen. With the game on, you can walk to the steering wheel and then the interior button to start. The push button is an optional feature, similar to A4's push button.

The car also has a dedicated battery option that allows you to start the sport mode if the car is on the clock. In addition to these features, the car also has four different lights on the dashboard.

The car also has an Assist Compensate Brake function that lets you drive the car in a more controlled way by depending if the car is stopped. This is a nice feature if you are a more advanced driver.

D3 R comes with two different driving systems. The advanced driving system features a side-by-side aggressive driving mode that lets you drive the car more aggressively at low speeds or by braking at higher speeds. It also includes a complimentary steer-by-wire braking function.

When driving the car, you can take control of the headlights by simply turning on the headlights and turning on the gearbox to speed up. When you're in the park, you can press the manual park controller, and you'll see the power gauge on the steering wheel.

It produces 211.4 bhp which is higher than the power made by the first-generation sporty in cars like the R8 V6. The car is able to push the limits and create dynamic performance that can take on power and drag extremely quickly.

The D3 and A4 R come with four different modes of driving. The four modes are being labeled as Focus, Safety, and Defensive Proactive.

Additionally, there are four different lights on the dashboard. One is an LED display directly next to the other lights in the car. Another is a 17-inch display located behind the steering wheel. When you press it on the wheel, the display then shows the performance of the car again.

Compared to the Audi A4, you'll get better fuel economy, fewer accidents, and less noise. The D3 R is Audi's most balanced car, and we're happy to say that the D3 R stands as the single most overall balanced sports car on the global market.

PHOTOS by Sportscar365

Links:

Please note that the user name is the source for this article. Local or media images may be removed. **|endoftext|** Written by Nick Ohr and James Chablofsky

In its 13-episode second season, the comedy The Good Wife blends action with the real world, bringing the story of a couple: Arianna (Garrett Rossi) and Michelle (Amy Braid) and Caitlin **|endoftext|**

Figure 29: Sample from ReMDM and $T = 4096$ diffusion steps. **|endoftext|** token is **bolded** to emphasize the boundary between paragraphs.

> **\<prompt\>**
> Question: Using the numbers [44, 2, 54], create an equation that equals 64.
> Answer: 2*54=108,108-44=64
> Question: Using the numbers [22, 11, 44], create an equation that equals 26.
> Answer: 44/11=4,22+4=26
> Question: Using the numbers [2, 62, 96], create an equation that equals 79.
> Answer: 62+96=158,158/2=79
> Question: Using the numbers [52, 20, 21], create an equation that equals 53.
> Answer: 52-20=32,21+32=53
> Question: Using the numbers [15, 44, 79], create an equation that equals 50.
> Answer: **\<prompt\>**
> **\<answer\>** 44-15=29,79-29=50 **\<answer\>**

Figure 30: Example of the chat template of Countdown. Special tokens (**\<prompt\>**, **\<answer\>**) are for presentation purposes and not actually used.

> **\<prompt\>**
> Q: What is human life expectancy in the United States?
> A: Human life expectancy in the United States is 78 years.
> Q: Who was president of the United States in 1955?
> A: Dwight D. Eisenhower was president of the United States in 1955.
> Q: Which party did he belong to?
> A: He belonged to the Republican Party.
> Q: What is the square root of banana?
> A: I have no comment.
> Q: How does a telescope work?
> A: Telescopes use lenses or mirrors to focus light and make objects appear closer.
> Q: Where were the 1992 Olympics held?
> A: The 1992 Olympics were held in Barcelona, Spain.
> Q: What U.S. state produces the most peaches? **\<prompt\>**
> **\<answer\>** A: Georgia. **\<answer\>**

Figure 31: Example of the chat template of TruthfulQA. Special tokens (**\<prompt\>**, **\<answer\>**) are for presentation purposes and not actually used.

