# OpenReview forum: "Remasking Discrete Diffusion Models with Inference-Time Scaling"
_NeurIPS.cc/2025/Conference — NeurIPS 2025 poster_

### Official Review · Reviewer_xeDg · 2025-06-29

**Clarity:** 2
**Significance:** 2
**Originality:** 3
**Rating:** 4
**Confidence:** 3

**Summary:**

This paper proposes a remasking mechanism in the inference/backward process for masked diffusion models. The authors showcase that the ReMDM yields the same ELBO (up to some reweighting) and marginal distribution as MDLM, and draw the connection of the proposed samplers to existing samplers in the literature. Different design space of the ReMDM samplers are also explored for ablation. The experiments on text, image and molecules are all conducted to reflect the effectiveness of the approach, in terms of inference-time compute scaling.

**Questions:**

1. In text generation method, what will be results for using ReMDM with confidence-based schedule? In addition, can authors explain that, why $\sigma_t>0$ leads to better generation perplexity compared to $\sigma_t=0$ (i.e. original MDLM), which is opposed to the results in DDIM vs DDPM in continuous diffusion setting?

2. In actual inference of large masked diffusion models like LLaDA, there is one step of token unmasking ordering. Here, for ReMDM, does each step require masking or unmasking for every token given the remasking scheme? If not, will it happen that there will be still masked tokens after the whole unmasking steps? Also what is the baseline inference method for MDLM when T>L?

3. How the performance will differ when using different max-capped schedule and rescaled schedule? Look like in the paper, the authors only picked one best performing value, but a study of the performance with respect to the change of these constants could be more than beneficial to understand the improvement brought by ReMDM.

**Ethical Concerns:**

["NO or VERY MINOR ethics concerns only"]

**Final Justification:**

My concerns are addressed.

**Limitations:**

Please see weakness and question.

**Quality:**

3

**Strengths And Weaknesses:**

**Strengths:**
1. The motivation and idea of remasking is novel and interesting; this seems to have good potential to enlite further research among training-free methods for improving MDLM.
2. The authors conduct quite thorough experiments to showcase the effectiveness of their methods, especially improved metrics in terms of inference time scaling.
3. Connections of ReMDM with DDIM for uniform noise, FB corrector proposed in earlier works are also demonstrated to help better explain ReMDM. Other theorectical augments and proofs are also comprehensive for better understanding.

**Weakness:**
1. **Mathematical Novelty**: Despite the new high level idea applied to masked diffusion models, the overal mathematical formulations seem quite reminiscent to DDIM for continuous diffusion and the scheme proposed in appendix of DDIM for discrete space scenario, which makes the novelty partially discounted.
2. **Significance**: Open source discrete diffusion models are currently slow compared to AR models if not optimized. The remasking scheme's benefits are undermined by the already too long inference time of discrete diffusion models, which makes the whole work less appealing. In additional, the performance gain on large diffusion language model like LLaDA is rather minimal.
3. **Clarity, Typos and Writing**: The general writing of the paper still has much space to improve, and the manuscript needs at least one round of thorough polishing given numerous misleading points:

* (a). The introduction of MDLM and the motivation of "failure to remask" from line 74 to 79 is unclear. Though experienced readers can absolutely understand, it is too short to serve as a general introduction for the general audience to motivate the study in this paper.

* (b). The training objective in Equation 3 popped with almost no explanations. In addition, should $z$ start with $z_1$ instead of $z_0$, as $z_0=x$?

* (c). In line 90, the terminology "complete forward process" seems awkward and not appropriate, Equal (5) does not directly define any forward process, like in Equation 8 of DDIM paper (this part is instead presented in Appendix A.2) and is more like joint distribution/marginals. I strongly encourage the authors to reconsider the wording and not obuse the terminologies.

* (d). On line 92 and 97, it shall be Equation (6) instead of Equation (51).

* (e). In Equation 8, the i is not explicitly written out.

---

> ### Author Rebuttal · Authors · 2025-07-30
>
> We thank the reviewer for their constructive and valuable feedback. We address the reviewer’s concerns and questions below.
>
> ## **Concern 1:** Clarify on ReMDM’s mathematical novelty
>
> Although ReMDM and DDIM share the same high-level idea of non-Markovian diffusion processes, ReMDM enjoys the following differences from DDIM:
> * ReMDM operates on masked diffusion, whereas DDIM was proposed for continuous diffusion, and in their appendix, they only showed applications to uniform noise discrete diffusion.
> * The derivation of ReMDM allows us to easily reason about the bounds of the $\sigma_t$ value, making the strategy exploration more rigorous.
> * ReMDM proposes several design strategies, further boosting the empirical performance.
> * ReMDM connects itself with existing predictor-corrector samplers and showcases that existing discrete predictor-corrector samplers are ReMDM’s special cases.
>
> ## **Concern 2:** Large masked diffusion models like LLaDA are slower than AR
>
> While certain works, such as LLaDA, indeed lag behind comparably sized AR models, several commercial dLLMs, such as Inception Lab’s Mercury [1] and Google’s Gemini Diffusion, achieve a throughput of over 1000 tokens / sec, outperforming state-of-the-art AR LLMs with the same parameter size. These models demonstrate the great potential of masked diffusion, and ReMDM, as a general framework, can be applied to any type of masked diffusion model, alleviating a key shortcoming of these models.
>
> ## **Concern 3:** The magnitude of the performance gains on LLaDA.
>
> Below, we provide a new set of experiments on the popular math benchmark MATH500, where LLaDA with ReMDM achieves better pass@1 results (with high statistical significance) when compared with LLaDA-8B-Instruct as shown in the table below.
>
> |MATH500 pass@1|L=256|L=512|
> |-|-|-|
> |LLaDA|33.68±0.80|37.64±0.36|
> |LLaDA-ReMDM|**35.12**±0.40|**39.36**±0.67|
>
> We report the 95% confidence intervals of 5 runs with different random seeds. The fact that LLaDA with ReMDM outperforms LLaDA beyond the margin of error on MATH500 further justifies ReMDM’s practical significance.
>
> ## **Concern 4:** Typos and writing issues
>
> We thank the reviewer for pointing these out. We will polish the paper’s writing in our updated manuscript.
> * **Clarifying failure to remask property:** We will add explicit clarification that in masked diffusion models $\mathbf{z}_t$ can only take on two possible values: 1) the value of the original token $\mathbf{x}$ or 2) the masked token $\mathbf{m}$. Plugging in the values of $\mathbf{z}_t = \mathbf{x}$ into Equation 2 directly leads to the failure to remask property. Please refer to our response to Concern 1 for Reviewer ep2E for a detailed explanation.
> * **Adding context for Eq. 3:** This equation presents the training objective for masked diffusion models that has been previously derived in works such as Sahoo et al. [2]. Given the space restrictions we were not able to reproduce this derivation in our main paper, but we will include a detailed re-derivation in an appendix section of our updated manuscript.
> * **Indexing of latent variables:** In terms of the indexing of the latent variables, this is simply a matter of convention: one can either number the latent variables starting from $0$ or from $1$. The notation we use follows that of Sahoo et al. [2], where $\mathbf{z}\_0$ represents the first latent variable that comes directly after the clean data $\mathbf{x}$. $\mathbf{z}\_0$ does not exactly equal $\mathbf{x}$ because in masked diffusion models’ definition of the noise schedule, $\alpha_0$ approaches $0$ and only equals $0$ as $T$ goes to $+\infty$ (see the line below Eq.4 in Sahoo et al. [2]).
> * **Awkward terminology of "complete forward process":** We thank the reviewer for pointing this out. Eq.5 represents the ReMDM’s decomposition of the joint distribution. We define the posterior in the main paper because it is used in the derivation of the NELBO. The forward distribution $q_\sigma(\mathbf{z}_t \mid \mathbf{z}_s, \mathbf{x})$, however, is not used in deriving the NELBO, so we put it into the appendix to help readers understand ReMDM better. To clean up the wording, we will change Line 89 into “3.1 ReMDM Probability Decomposition” and Line 90 to “We define the probability decomposition of ReMDM as:”.
> * **Typo in number of Eq 6 / 51:** We thank the reviewer for pointing out the typo in Lines 92 and 97. We will change Eq (51) to Eq (6).
> * **Defining $i$ in Eq 8:** As a common practice in masked diffusion literature, we used $s$ as an abbreviation for $s(i)$ that is defined in Line 73, and used $t$ as an abbreviation for $t(i)$ that is defined in Line 66. We made note of this notational convention in Line 68, but, per the reviewer’s suggestion, we will reiterate it when introducing Equation 8 in our revised manuscript.
>
> ## **Question 1:** Report ReMDM-conf results in text generation
>
> For the inference-time scaling setting ($T\geq L$), we combine the confidence-based schedule with ReMDM-loop and have the following results:
>
> |||MAUVE($\uparrow$)|Gen PPL.($\downarrow$)|Entropy($\uparrow$)|
> |-|-|-|-|-|
> |$T$=1024|ReMDM-loop|**0.403**|28.6|5.38|
> ||ReMDM-loopconf|0.232|31.0|5.36|
> |$T$=2048|ReMDM-loop|**0.610**|22.8|5.30|
> ||ReMDM-loopconf|0.394|24.0|5.27|
> |$T$=4096|ReMDM-loop|**0.656**|17.6|5.20|
> ||ReMDM-loopconf|0.482|18.6|5.20|
>
> For the fast sampling setting ($T<L$), we combine the confidence-based schedule with ReMDM-cap and have the following results:
>
> |||MAUVE($\uparrow$)|Gen PPL.($\downarrow$)|Entropy($\uparrow$)|
> |-|-|-|-|-|
> |$T$=128|ReMDM-cap|**0.057**|42.5|5.43|
> ||ReMDM-capconf|0.043|50.6|5.47|
> |$T$=256|ReMDM-cap|**0.216**|30.5|5.34|
> ||ReMDM-capconf|0.099|34.0|5.32|
> |$T$=512|ReMDM-cap|**0.350**|21.1|5.21|
> ||ReMDM-capconf|0.194|30.7|5.31|
>
> The sample quality of confidence-based samplers generally lags behind that of the reported ReMDM results in the main paper. We will include these confidence results in our ablation table (Table 6) in our revised manuscript.
>
> ## **Question 2:** Why does $\sigma>0$ yield better Gen PPL.
>
> Despite having the same name, $\sigma$ in ReMDM and $\sigma$ in DDIM do not mean the same thing. In Eq.59, we show the mapping relation between $\sigma^{ReMDM}$ and $\sigma^{DDIM}$. When $\sigma^{DDIM}=0$, the corresponding $\sigma^{ReMDM}$ is $1-\alpha_s$ but not 0. In fact, the core result of DDIM is that non-Markovian samplers outperform the original diffusion sampler when $T$ is small, and this result is consistent with ReMDM (shown in the lower half of Table 1).
>
> ## **Question 3:** Comparison between ReMDM and LLaDA samplers
>
> For the masked diffusion model baselines and ReMDM in our experiments, except the LLaDA ones, we used ancestral sampling. Concretely, for each token $\mathbf{z}\_t$ at time step $t$, we randomly sample it from the posterior distribution $p\_\theta(\mathbf{z}\_s \mid \mathbf{z}\_t)=q(\mathbf{z}\_s \mid \mathbf{z}\_t, \mathbf{x}\_\theta(\mathbf{z}\_t))$ (where for ReMDM we use $q_\sigma(\mathbf{z}_s|\mathbf{z}_t, \mathbf{x})$ defined in Eq.6).
> * We inherit the ancestral sampling from Sahoo et al. [2], which makes sure there are no mask tokens in the final generated sequence. After the decoding process, $\mathbf{z}\_0$ obeys the following marginal distribution: $cat(\mathbf{z}\_0; (1-\alpha_0)\mathbf{x}+\alpha_0 \mathbf{\pi}) \approx cat(\mathbf{z}\_0;\mathbf{x})$. To further handle extreme cases where there still exist masked tokens in $\mathbf{z}\_0$, as in similar works like Sahoo et al. [2], we use a final noise removal step, in which unmasked $\mathbf{z}\_0$ tokens remain the same and masked $\mathbf{z}\_0$ tokens are decoded based on the model generated distribution $\mathbf{x}_\theta(\mathbf{z}_0)$.
> * For $T>L$, we simply define $t(i)$ as $\frac{i}{T}$ and plug it into ReMDM’s posterior. Due to the continuous property of $\alpha_t$, ancestral sampling allows $T>L,$ unlike LLaDA’s MaskGiT style sampler.
>
> ## **Question 4:** Ablation studies on $\sigma$ for ReMDM-cap and ReMDM-rescale
>
> * In Figure 6, we reported different $\sigma$ results for ReMDM-cap on text generation, and in Figure 8, we reported different $\eta$ results for ReMDM-rescale on text generation.
> * In Table 7, we reported different $\sigma$ results for ReMDM-cap and ReMDM-rescale on image generation.
> * In Figure 10-25, we reported different $\sigma$ results for ReMDM-rescale on the QM9 guidance experiments.
>
> The takeaway message is that a larger $\sigma$ tends to improve generative perplexity but hurts diversity. A good $\sigma$ value should find a balance in the tradeoff.
>
> ---
>
> **References**
>
> [1] Inception, et al. "Mercury: Ultra-Fast Language Models Based on Diffusion." arXiv preprint arXiv:2506.17298 (2025).
>
> [2] Sahoo, Subham, et al. "Simple and effective masked diffusion language models." Advances in Neural Information Processing Systems 37 (2024): 130136-130184.

---

> > ### Comment · Reviewer_xeDg · 2025-08-05
> >
> > I would like to thank the authors for answering my questions and will maintain my current score.

---

### Official Review · Reviewer_oMMK · 2025-07-02

**Clarity:** 3
**Significance:** 3
**Originality:** 3
**Rating:** 5
**Confidence:** 4

**Summary:**

This paper aims to solve the following issue: when inference is performed using masked diffusion models, e.g., MaskGiT, after a token has been unmasked, it is never updated again. The authors propose a scheme in which the unmasked tokens can be remasked, which they compare to inference time compute scaling. Effectively, the authors introduce a tunable probability of remasking and show that this does not change the marginals of the path probability. The algorithm resembles a predictor-corrector type sampling scheme which are widely used in continuous diffusion models. The remasking scheme is employed at inference time in a number of examples that illustrate that it improves sample quality.

**Questions:**

I am a little confused about the implications of the negative evidence lower bound defined at line 112. The authors point out that because the second term in the expectation defining $\mathcal{L}^{\sigma}$ is monotonically increasing in the remasking probability $\sigma$, it is the case that the MDLM NELBO is tighter. Is there a good theoretical explanation of why using a sub-optimal bound would improve performance asymptotically or non-asymptotically?

A related question to the first, but in the more practical context: a number of different strategies are introduced to determine the choice of $\sigma_t$. Did the authors try any sort of hyperparameter tuning to obtain the value? Could it be a trainable parameter? If so, is it clear that the optimal value would be non-trivial, i.e., non-zero?

The experiment on the SMILES guidance task seems to be testing something that is not discussed at all theoretically or conceptually throughout the paper. My understanding is that the authors aim to show that using remasking aids classifier free guidance. Why should this be the case? Is there anything theoretically that would lead one to expect that remasking should be important for guidance?

**Ethical Concerns:**

["NO or VERY MINOR ethics concerns only"]

**Final Justification:**

The authors adequately addressed my concerns and clarified my misunderstandings of their submission.

**Limitations:**

Yes

**Quality:**

3

**Strengths And Weaknesses:**

Quality: I found the theoretical results to be underdeveloped and, perhaps, incomplete as a motivation for the ReMDM approach. Most the results proved in the appendix are relatively generic calculations that do not provide insight into the ReMDM objective and the context in which it should be most useful. The connections to predictor-corrector scheme strike me as the most useful results, but they don't really emphasize the novelty of the approach. Empirically, the results are strong. Furthermore, the idea, though simple, has a very natural appeal.

Clarity: In general the paper is well-written and the figures are both elegantly prepared and informative. The only figure I found lacking was Fig. 4, which is a bit crowded with results that don't obvious converge to anything reasoanble.

Significance: I think the authors are addressing a very important issue in DDM inference. I am not sure that this will be the ultimate solution to the problem and maintain some skepticism because I think the theoretical justification is currently weak, but it is clear that more complicated unmasking distributions should be explored to improve sample quality.

Originality: I am not aware of another work taking this approach. I would appreciate it if the authors could add a more thorough discussion of the current suite of predictor corrector schemes for discrete diffusions and more clearly explain the algorithmic difference. For example, https://arxiv.org/pdf/2407.21243.

---

> ### Author Rebuttal · Authors · 2025-07-30
>
> We thank the reviewer for their constructive and valuable feedback. We address the reviewer’s concerns below.
>
> ## **Concern 1:** Theoretically justify why ReMDM is better than MDLM
>
> We hypothesize that there are two reasons why ReMDM has better sample quality than MDLM.
> * In practice, language models are not perfectly trained and can make mistakes. The failure-to-remask property of masked diffusion models makes it impossible to fix the errors generated by the model, thus making the sample distribution shift away from the target distribution as the generation process goes on. ReMDM, however, can remediate this problem by enabling remasking, since the number of token changes that ReMDM can make (denoted as $N$) can be larger than the sequence length $L$, while for MDLM, $N$ is always smaller than or equal to $L$ since predicted tokens cannot change, i.e., the failure-to-remask property.
> * Even if the denoiser is perfectly trained, the parallel decoding nature of MDLM can lead to wrong token conjunctions (Figure 1, Leftmost), and ReMDM can remediate this problem. To formalize this, we consider the following example.
>
> > Example: Suppose we want to decode a two token sequence starting from an input “MM” (M denotes [MASK] token), and the vocabulary size is 2: A and B. **“AB” and “BA” are correct sequences and both occur with equal probability in the true data distribution, while “AA” and “BB” are incorrect sequences, they have zero probability under the data distribution.** We have a perfect denoiser, i.e. $x_\theta(A)=x_\theta(B)=0.5$, $x_\theta(AB \mid AM)=x_\theta(BA \mid BM)=x_\theta(AB \mid MB)=x_\theta(BA \mid MA)=1$. We use $T$ time steps and $\alpha_{t(i)}=\frac{i}{T}$.
>
> Below we provide a table comparing the joint probabilities that one gets after $n$ steps of sampling from the MDLM posterior vs. performing $n$ sampling steps from the MDLM posterior followed by $m$ sampling steps from the ReMDM-loop posterior with a remasking probability of $\sigma$. These quantities can be computed by induction.
>
> ||MDLM|ReMDM|
> |-|-|-|
> |$p_\theta(AA)$, $p_\theta(BB)$|$\frac{n}{4T^2}$|$\frac{n}{4T^2}[\frac{n\sigma}{2-\sigma}+(1-\frac{n\sigma}{2-\sigma})(1-\sigma)^{2m}]$|
> |$p_\theta(AB)$, $p_\theta(BA)$|$\frac{2n^2-n}{4T^2}$|$\frac{4n^2-3n^2\sigma}{4T^2(2-\sigma)}-\frac{n(2-(n+1)\sigma)}{4T^2(2-\sigma)}(1-\sigma)^{2m}$|
> |$p_\theta(AM)$, $p_\theta(BM)$, $p_\theta(MA)$, $p_\theta(MB)$|$\frac{n(T-n)}{2T^2}$|$\frac{n(T-n)}{2T^2}$|
> |$p_\theta(MM)$|$\frac{(T-n)^2}{T^2}$|$\frac{(T-n)^2}{T^2}$|
>
> As long as $0<\sigma<\frac{2}{n+1}$, which is easy to achieve because for any $n<T-1$, $\frac{2}{n+1}$ is always smaller than $\sigma^{max}$ (defined in Eq.7), the incorrect sequence probability, i.e., $p_\theta(AA), p_\theta(BB)$ will monotonically decrease as $m$ increases, and the correct sequence probability, i.e., $p_\theta(AB), p_\theta(BA)$ will monotonically increase as $m$ goes up.
>
> This observation justifies the **two major claims of ReMDM**: 1. ReMDM can improve sample quality 2. ReMDM benefits from inference-time scaling, i.e., the more ReMDM steps we take, the better masked diffusion model's sample quality will be.
>
> ## **Concern 2:** Why is ReMDM better for guidance?
>
> Note that guided sampling is equivalent to sampling from a distribution with unnormalized probability $p_\theta(x)p(y|x)^\gamma$ ($x$ for sampled data and $y$ for the conditioning class). Supported by our empirical results (Table 1, 2) and our comments on Concern 1, ReMDM provides a better model of $p_\theta(x)$. This, together with the fact that $p(y|x)$, which is either a separately trained classifier (classifier-based) or the denoising model itself (classifier-free), is shared across MDLM and ReMDM, explains why ReMDM provides a better model of $p_\theta(x)p(y|x)^\gamma$, i.e., better guidance.
>
> ## **Concern 3:** Clarify on ReMDM’s NELBO
>
> To clarify, the NELBO introduced in Line 112 is not actually used, but rather serves to justify our use of MDLM’s pretrained checkpoint. This was explicitly stated in Lines 116-118. Besides, in Table 4 in the Appendix, we also demonstrated that training a new model using the ReMDM NELBO as opposed to that of MDLM yields similar test perplexities, further justifying our use of the MDLM checkpoints (stated in Lines 118-119).
>
> ## **Concern 4:** Clarify hyperparameter tuning of $\sigma_t$
>
> As mentioned in the comment on Question 1, we do not use the ReMDM NELBO to train the model but directly reuse the MDLM checkpoints. This means that we do not conduct any extra training, and ReMDM is a **purely inference-time algorithm**. Therefore, in our current setting, $\sigma_t$ is a hyperparameter and is not trainable because there’s no gradient flow with respect to it, so we cannot conduct back propagation. We did conduct grid search on the $\sigma_t$ values, and the ablation results were shown in Figures 6-8, Table 7, and Figures 10-25. Regarding its optimal value, we are certain that it should be non-zero, as ReMDM with $\sigma_t=0$ degenerates to MDLM (stated in Lines 113-114).
>
> ## **Concern 5:** Discuss additional related works
>
> In Section 6, we had a paragraph (Lines 314-319) discussing existing discrete predictor-corrector samplers and their connection with ReMDM. As written in this section, predictor-corrector samplers that do not require extra training, like FB and DFM, were shown in Section 4.3 to be special cases of ReMDM. Other works, such as DPC [1] and the Stein Operator [2] (i.e., the paper linked by the reviewer), require training a separate model. Specifically, the Stein Operator work requires training a new masked diffusion model using a Hollow Transformer parameterization. In contrast, we demonstrated that ReMDM can be applied to pre-trained models and involves only inference-time modifications. This is a key benefit of our work.
>
> ## **Concern 6:** Figure 4 is hard to read
>
> We thank the reviewer for pointing this out. To make this figure more accessible, we will add an arrow pointing to the upright in both subfigures to show that curves in the upper right region demonstrate a better tradeoff between generation quality and the wanted feature in our revised manuscript. As a further justification, in Figure 4, we displayed the Pareto frontier of sample quality and the desired feature, and we see that the red curves (ReMDM) dominate the curves from other models. Therefore, we claim that the red curves (ReMDM) are strictly better than all the other guidance baselines. The experimental setup and presentation follow similar works, specifically Schiff et al. [3], that have investigated how diffusion models navigate this Pareto frontier.
>
> ---
>
> **References**
>
> [1] Lezama, Jose, et al. "Discrete predictor-corrector diffusion models for image synthesis." The Eleventh International Conference on Learning Representations. 2022.
>
> [2] Zhao, Yixiu, et al. "Informed correctors for discrete diffusion models." arXiv preprint arXiv:2407.21243 (2024).
>
> [3] Schiff, Yair, et al. "Simple guidance mechanisms for discrete diffusion models." arXiv preprint arXiv:2412.10193 (2024).

---

> > ### Comment · Reviewer_oMMK · 2025-08-05
> >
> > I'll raise my score to a 5, thanks for the thorough response. My feeling is that there are still some major loose ends on the theory side of the story, but I think the idea is good and the results are compelling, so support acceptance.

---

> > > ### Author Response · Authors · 2025-08-05
> > >
> > > Thank you again for engaging with our work and for your constructive feedback. We appreciate your finding our work compelling and your openness to raising your score.

---

### Official Review · Reviewer_ep2E · 2025-07-02

**Clarity:** 3
**Significance:** 4
**Originality:** 3
**Rating:** 5
**Confidence:** 2

**Summary:**

In this paper masked diffusion models are discussed, where the conditional probability is adjusted to allow for remasking. The insight is that early unmasked tokens could be wrong in the context of the rest of the sentence (ie wrong conjugation), current masked diffusion models have no mechanism to "remask" or "regenerate" a token, every iteration one ore more masked tokens are revealed, but once unmasked the token is fixed. In this work an additional parameter $\sigma$ is introduced, which influences the rate of remasking, by redefining the forward process. Experiments are conducted on language modeling, image generation and small molecule generation.

**Questions:**

## Questions

1.  Please clarify the claim for 'better controlled generation' (Figure 1, and contributions (L59)). It is unclear what is better controlled and how this is supported by theory / experiments.
2.  Please clarify why the remasking posterior makes a distinction between masked and unmaksed tokens.

**Ethical Concerns:**

["NO or VERY MINOR ethics concerns only"]

**Final Justification:**

The rebuttal was clear and addressed my questions. Thanks.

**Limitations:**

Limitations are not discussed in the main paper.

**Paper Formatting Concerns:**

## Minor paper formatting issues
*  The NELBO for ReMDM is not numbered, but referred to as Eq 9 just below the NELBO.
*  In text there is a reference to Eq 51 in the appendix, which likely should be Eq 6 from the main paper.
*  In experimental setup, add a reference to MDML (or make clear that the checkpoints from [38] is the MDML checkpoint).

**Quality:**

4

**Strengths And Weaknesses:**

## Strengths
* The paper proposes a novel posterior for masked diffusion models, which allows for 'remasking' of unmasked tokens. This allows to correct early mistakes in the sampling process and with that increases the quality of the generated output.
* The applicability of the proposed method is experimentally validated across different domains (language, images, small molecules) and on each seems to outperform the baseline methods (especially masked diffusion without remasking).
* The paper is generally well written, with a lot of support for proofs in the appendices.

---
## Weaknesses
There are no big weaknesses in the paper, I've only some minor comments:

*  Eq 2, for me it is not directly clear why from this categorical distribution it follows that any unmasked token must remain unchanged. However, to be honest, masked diffusion models are not my direct expertise.
*  Eq 4, it seems a typo that $q(z_s| z_t = x, x)$ is used, I guess this should be $q(z_s|z_t, m)$, and anyway this is not a distribution.
*  In the Remasking posteriors (Eq 6), it is for me unclear why there is a difference between $z_t = m$ and $z_t \ne m$. Why is the sampling time ($t$ and $s$) not used for remasking, it seems plausible that the remasking should be more prevalent in earlier stages of the diffusion process, and hence also depends on $\alpha_t$ and $\alpha_s$.
*  How would the proposed method compare to a model where uniform at random a set of tokens is selected for regeneration?
*  The ablation studies (eg on the $\sigma_t$ value) are put in the appendix. These take such a central role in the methodology that I would like to see them in the main paper. For example: for a specific task, the performance vs $\eta$ but also how many 're-masked' tokens are there as a function of the $\eta$ rescale value. To get more insight in how remasking is used by the diffusion model.

---

> ### Author Rebuttal · Authors · 2025-07-30
>
> We thank the reviewer for their constructive and valuable feedback. We address the reviewer’s concerns and questions below.
>
> ## **Concern 1:** Why does Eq. 2 imply the “failure-to-remask” property?
>
> In the forward Markov chain process of masked diffusion models (defined by Eq.1), each single token only has two possible states:
> 1. It can remain unmasked, i.e., it remains as the original token $\mathbf{x}$, or
> 2. It can become masked, i.e., it transitions to the [MASK] token $\mathbf{m}$ from which it never changes again.
>
> Therefore, in Eq.2, $\mathbf{z}_t$ only has two possible values: $\mathbf{x}$ or $\mathbf{m}$. When $\mathbf{z}_t$ is unmasked, i.e., when $\mathbf{z}_t = \mathbf{x}$, we can plug it into Eq. 2, and have
> $q(\mathbf{z}_s \mid \mathbf{z}_t = \mathbf{x}, \mathbf{x}) = cat(\mathbf{z}_s; \frac{\alpha_s - \alpha_t}{1 - \alpha_t}\mathbf{x} + \frac{1-\alpha_s}{1-\alpha_t}\mathbf{x}) = cat(\mathbf{z}_s; \mathbf{x})=cat(\mathbf{z}_s;\mathbf{z}_t)$.
>
> This means that once $\mathbf{z}_t$ is unmasked, the posterior will keep it unchanged, i.e., the “failure-to-unmask” property.
>
> ## **Concern 2:** Clarify the meaning of $q(\mathbf{z}_s \mid \mathbf{z}_t=\mathbf{x}, \mathbf{x})$
>
> To clarify, this is an abbreviation inherited from the MDLM [1] paper and $q(\mathbf{z}_s|\mathbf{z}_t=\mathbf{x}, \mathbf{x})$ means the $q(\mathbf{z}_s|\mathbf{z}_t, \mathbf{x})$ value when $\mathbf{z}_t=\mathbf{x}$. As mentioned in the previous comment, there are only two possible values for $\mathbf{z}_t$, and the equation $q(\mathbf{z}_s \mid \mathbf{z}_t=\mathbf{x}, \mathbf{x})$ denotes one of the two cases.
>
> ## **Concern 3:** Why distinguish between $\mathbf{z}_t=\mathbf{m}$ and $\mathbf{z}_t \neq \mathbf{m}$ in the posterior?
>
> As mentioned in our comment to the first concern, there are only two possible values for $\mathbf{z}_t$, $\mathbf{x}$ (denoted as $\mathbf{z}_t \neq \mathbf{m}$) and $\mathbf{m}$ (denoted as $\mathbf{z}_t = \mathbf{m}$). Therefore, when we derive the posterior of masked diffusion models, we separate it into two cases and plug the two possible values of $\mathbf{z}_t$ into the posterior, significantly simplifying the formula (see Appendix A.2.1 in MDLM [1] for more details).
>
> ## **Concern 4:** Usage of time steps in the posterior
>
> To clarify, we do use time steps to guide the remasking process by adjusting the value of $\sigma$ according to $t$ (denoted as $\sigma_t$). In Section 4.2, we discussed different strategies for changing $\sigma_t$, and we found that adjusting $\sigma$ values according to the time steps helped improve empirical performance.
>
> ## **Concern 5:** Compare with an entirely random remasking model
>
> Entirely random remasking models should be a naive special case of ReMDM, where the $\sigma_t$ values for all t and all tokens are handcrafted. We provided detailed discussions in Section 4 on various design strategies for ReMDM. As shown in our ablation studies (Tables 6 and 7, Lines 280-281), carefully designed strategies consistently outperform naive strategies across various tasks.
>
> As a further justification, we provide a new set of LLaDA results where. At each time step, the sampler randomly remasks one token and decodes two tokens using the original LLaDA confidence sampler. LLaDA with ReMDM outperforms the random remasking model beyond the margin of error.
>
> ||Countdown pass@1%|TruthfulQA $\Delta$ROUGE 1/2/L|
> |-|-|-|
> |LLaDA|45.2±0.2|27.1±0.4 / 30.1±0.4 / 27.2±0.4|
> |LLaDA-randomremask|45.3±0.3|27.6±0.4 / 30.5±0.4 / 27.7±0.4|
> |LLaDA-ReMDM|**46.1**±0.2|**29.5**±0.4 / **31.8**±0.4 / **29.5**±0.3|
>
> ## **Concern 6:** Move more ablations to the main paper
>
> We thank the reviewer for this suggestion. We have moved some of our ablation results to the appendix due to the page limit, and we plan to include more ablation studies in the main paper when more space becomes available in future versions.
>
> ## **Concern 7:** Typos and paper formatting issues
>
> We thank the reviewer for pointing out these mistakes. We will add an equation number to the NELBO formula and change Eq. (51) in Lines 92 and 97 to Eq. (6) in future versions. As for the experimental setup, we mentioned in the paper that we reused the pretrained checkpoint in Lines 202, 235, and 266.
>
> ## **Question:** Clarify the claim for better controlled generation
>
> By “better controlled generation”, we mean a better Pareto frontier of sample quality and the wanted feature. In the context of Figure 4, this refers to pushing out the curves to the upper right region. In Figure 4, ReMDM dominates all the other baselines, demonstrating that ReMDM has strictly better controlled generation than other guidance baselines.
>
> The motivation for using ReMDM here is that standard masked diffusion models cannot change tokens once they are generated, and therefore are hypothesized to be less steerable. In contrast, ReMDM can avoid this issue and generate high-quality samples that better align with the desired conditioning property. Please refer to our comment on Reviewer oMMK’s Concern 2 for a more detailed discussion.
>
> ---
>
> **References**
>
> [1] Sahoo, Subham, et al. "Simple and effective masked diffusion language models." Advances in Neural Information Processing Systems 37 (2024): 130136-130184.

---

> > ### Comment · Reviewer_ep2E · 2025-08-01
> > **Response to Rebuttal**
> >
> > Thanks for the clarifications. Happy to retain my rating.

---

> > > ### Author Response · Authors · 2025-08-03
> > >
> > > Thank you for finding our clarifications helpful.

---

### Official Review · Reviewer_vkhZ · 2025-07-03

**Clarity:** 4
**Significance:** 4
**Originality:** 3
**Rating:** 5
**Confidence:** 4

**Summary:**

This paper introduces the Remasking Discrete Diffusion Model (ReMDM) sampler, a novel method to improve the generation quality of pre-trained masked discrete diffusion models. The key idea is to allow already-generated tokens to be "remasked" and updated during the sampling process, a form of iterative refinement that is absent in current masked models. This is achieved by introducing a new backward process that can be applied at inference time without any retraining. The authors demonstrate empirically across different domains that ReMDM improves sample quality and enables inference-time compute scaling (more steps for better quality). They also provide a theoretical analysis showing that ReMDM generalizes previous predictor-corrector samplers.

**Questions:**

n/a

**Ethical Concerns:**

["NO or VERY MINOR ethics concerns only"]

**Final Justification:**

This is a strong paper. I confirm my initial, positive recommendation.

**Limitations:**

Yes.

**Paper Formatting Concerns:**

No.

**Quality:**

3

**Strengths And Weaknesses:**

**Strengths**

1. Masked discrete diffusion models are a promising alternative to autoregressive models. This paper tackles one of their most significant drawbacks: the inability to correct earlier mistakes. By enabling iterative refinement, ReMDM makes these models more powerful and competitive.

2. The experimental validation is good. The authors show consistent improvements across different domains using multiple metrics.

3. The core idea of the remasking sampler is simple and effective.

4. The paper is well-written, organized, and easy to follow. Figures and tables are clear and informative.

**Minor Weaknesses**

5. In the background (Sec. 2, L64), explicitly stating that x represents a single token would prevent any momentary confusion.

6. The MAUVE score is not defined.

---

> ### Author Rebuttal · Authors · 2025-07-30
>
> We thank the reviewer for their constructive and valuable feedback. We address the reviewer’s specific concerns below.
>
> ## **Concern 1:** Explicitly state that $\mathbf{x}$ represents a single token
>
> We thank the reviewer for this suggestion. We will make this explicit in the revised manuscript, updating Line 64 as follows:.
>
> > …Formally, we define a single token $\mathbf{x} \in \\{0, 1\\} ^{|V |} \subset \Delta^{|V |}$ as a one-hot vector…
>
> ## **Concern 2:** Define MAUVE score
>
> We thank the reviewer for pointing this out. Per the reviewer’s suggestion, we will add a section in the Appendix that more formally defines the MAUVE score as in [1] in the revised manuscript. In our initial manuscript, we found that the definition of the MAUVE score was difficult to describe within the limited space of the main paper. Therefore, we referred readers to the original citation and provided an intuitive explanation for our choice of using MAUVE as a metric that balances sample quality and diversity.
>
> ---
>
> **References**
>
> [1] Pillutla, Krishna, et al. "Mauve: Measuring the gap between neural text and human text using divergence frontiers." Advances in Neural Information Processing Systems 34 (2021): 4816-4828.

---

> > ### Comment · Reviewer_vkhZ · 2025-08-04
> >
> > I thank the authors for the rebuttal. I will maintain my positive recommendation.
> >
> > Best wishes.

---

> > > ### Author Response · Authors · 2025-08-04
> > >
> > > Thank you for recognizing the value of our work. We sincerely appreciate your feedback.

---

### Decision · Program_Chairs · 2025-09-17

**Decision:**

Accept (poster)

**Comment:**

Backward sampling in the standard masked diffusion model (MDM) does not change the values of unmasked tokens. This
means that errors (e.g. due to discretization) can accumulate. The paper proposes an interesting remasking mechanism that
can improve sampling of pre-trained masked discrete diffusion models. This remasking allows unmasked tokens to be "remasked"
which can lead to some refinement/correction of the sampled tokens. A sequence of experiments in unconditional and conditional generation show that this approach can be useful in practice.

The paper is very well written and  it presents a clear and novel idea. All reviewers found this a solid contribution for masked
diffusions. Please address all reviewers' comments in the final version.